



# Circumarctic landcover diversity considering wetness gradients

Annett Bartsch[1], Aleksandra Efimova[1], Barbara Widhalm[1], Xaver Muri[1], Clemens von Baeckmann[1],
Helena Bergstedt[1], Ksenia Ermokhina[2], Gustaf Hugelius[3,4], Birgit Heim[5], and Marina Leibmann[6]

[1]b.geos, Industriestrasse 1, 2100 Korneuburg, Austria
[2]A.N. Severtsov Institute of Ecology and Evolution, Russian Academy of Sciences, Moscow, Russia
[3]Department of Physical Geography, Stockholm University, Stockholm, Sweden
[4]Bolin Centre for Climate Research, Stockholm University, Stockholm, Sweden
[5]Alfred Wegener Institute for Polar and Marine Research, Potsdam
[6]Earth Cryosphere Institute, Tyumen Scientific Centre SB RAS

**Correspondence:** Annett Bartsch (annett.bartsch@bgeos.com)

**Abstract.** Landcover heterogeneity information considering soil wetness across the entire Arctic tundra is of interest for a wide range of applications targeting climate change impacts and ecological research questions. Patterns potentially link to permafrost degradation and affect carbon fluxes. First a landcover unit retrieval scheme which provides unprecedented detail by fusion of satellite data using Sentinel-1 (synthetic aperture radar) and Sentinel-2 (multispectral) has been adapted. Patterns of lakes, wetlands, general soil moisture conditions and vegetation physiognomy are represented at 10 m nominal resolution. Units with similar patterns are identified with a k-means approach and documented through statistics derived from comprehensive in situ records for soils and vegetation (more than 3500 samples). The result goes beyond the capability of existing landcover maps which have deficiencies in spatial resolution, thematic content and accuracy. Wetness gradients have been eventually assessed and measures for landscape heterogeneity were derived north of the treeline. About 40% of the area north of the treeline falls into three units of dry types with limited shrub growth. Wetter regions have higher landcover diversity than drier regions. 45% of the Arctic landscape is highly heterogeneous with respect to wetness considering 1kmx1km units (representative scale of frequently used regional landcover and permafrost modelling products). Wetland areas cover on average 9% and moist tundra types 32%, what is potentially of relevance for methane flux upscaling.

## 1 Introduction

Landsurface hydrology, moisture gradients, wetting and drying processes play a major role in the context of Arctic biodiversity studies, carbon flux upscaling, carbon pools quantification, permafrost mapping and human impact assessment. In order to address such processes both open water fraction and soil moisture related information is essential. In the Arctic, the landscape heterogeneity and especially the occurrence of lakes, has so far been a major limiting factor for retrieval of near surface soil moisture time series using microwave satellite data as commonly applied on regional to global scale due to spatial resolution (Hogstrom and Bartsch, 2017; Högström et al., 2018; Wrona et al., 2017). Landcover properties are therefore often used as proxy for subground conditions. Multi-spectral satellite data, especially from Landsat (30m) have been regionally employed for characterizing typical tundra landscape types reflecting moisture regimes and vegetation physiognomy previously (Bartsch





et al., 2016b). Soil characteristics are for example required for parameterization of heat transfer modelling for permafrost stud-
ies (Westermann et al., 2017). Global landcover maps are currently used although deficiencies for the Arctic are known. This
relates to thematic content as well as high landcover heterogeneity not reflected by the comparably coarse spatial resolution.
For example, the accuracy of the Landcover CCI dataset (300 m) for high latitude wetlands has been determined to be only
19% (Palmtag et al., 2022). Nevertheless it has been used for permafrost modelling (Westermann et al., 2017) and wetland
delineation (Olefeldt et al., 2021; Albuhaisi et al., 2023) accepting the uncertainties in the absence of a better alternative.

The issue of spatial resolution has been extensively discussed for water bodies (Liljedahl et al., 2016; Muster et al., 2019)
and also for further landcover features on regional scale (e.g. fluxes (Virtanen and Ek, 2014), soils (Siewert et al., 2015),
carbon balance and landscape heterogeneity (Treat et al., 2018)). All studies call for very high resolution data, in the order of
few meters or sub-meter scale, for which availability and access is limited (Bartsch et al., 2023). A scheme with high thematic
content (with respect to tundra) has been previously implemented based on 1kmx1km data using multispectral data (AVHRR
- Advanced Very High Resolution Radiometer; Walker et al. (2005); Raynolds et al. (2019)). This widely used Circumpolar
Vegetation Map (CAVM) provides vegetation community information but does not provide a measure of the high spatial
heterogeneity of Arctic landscapes.

Recently, data from the multispectral Sentinel-2 mission which provides 10-20m detail came into focus. This provides
an advance compared to for example Landsat (30m) although still lacking some detail. Such data have been also shown of
added value in combination with C-band radar information from Sentinel-1 with similar resolution to obtain landcover related
information (Bartsch et al., 2019a, 2020; Scheer et al., 2023). For example the approach by Bartsch et al. (2019a, b) is based
on a combination of Sentinel-1 and Sentinel-2 using a k-means unsupervised classification. The application potential of the
obtained landcover units (21 classes) has been shown in regional studies (Bartsch et al., 2019a; Bergstedt et al., 2020; Kraev
et al., 2022; Spiegel et al., 2023). The approach targets use of landcover information as proxy for soil conditions, specifically
wetness gradients. This is achieved through focus on the use of selected bands of Sentinel-2 and the choice of frozen state
acquisitions of Sentinel-1.

Wetness patterns are known to drive the occurrence of certain vegetation communities in tundra environment (e.g. Silvertown
et al. (2014); Dvornikov et al. (2016); Ackerman et al. (2017)). A commonly used multispectral index is the Tasseled Cap
Wetness Index. This index has been demonstrated of value for longterm change studies targeting permafrost degradation in
tundra (Nitze and Grosse, 2016). Whereas the commonly used Normalized Vegetation Index NDVI utilizes the red and near
infrared information only, the Tasseled Cap Wetness index also considers green and short-wavelength infrared information
(Crist, 1985). Considering Sentinel-2, bands available at 10m as well as 20m nominal resolution are of interest.

The use of Sentinel-1 is usually confined to unfrozen conditions in order to use the moisture information reflected in the
backscatter measurements (e.g. for the Arctic Reschke et al. (2012); Ou et al. (2016)). High backscatter is associated with
high soil moisture. Other scattering mechanisms, such as surface roughness, however, also contribute to backscatter increase.
It has been shown that relevant information can be also derived from C-band SAR data acquired under frozen conditions
(Bartsch et al., 2016b; Widhalm et al., 2015). It has been for example demonstrated in Bartsch et al. (2016b) that C-band
frozen backscatter at HH polarization (horizontally sent and horizontally received) can be used as proxy for estimation of near



surface soil organic carbon in tundra regions. For tundra environments, this also coincides with specific landsurface wetness gradients what has been initially shown for ENVISAT ASAR (Advanced Synthtic Aperure Radar) Global monitoring mode

(1km) by Widhalm et al. (2015). The derived CAWASAR (CircumArctic Wetlands based on Advanced Aperture Radar) map has been previously applied for a permafrost equilibrium model soil parameterization (Obu et al., 2019) as well as for a recent estimation of the global methane budget (wetlands as input for landsurface modelling (Saunois et al., 2016), global wetland map compilation (Zhang et al., 2021)) and climate change vulnerability assessment (Kåresdotter et al., 2021).

Bartsch et al. (2019a) demonstrated that the different landscape units derived based on selected Sentinel-1/2 data reflect dif-

65 ferences in soil wetness as can be determined by seasonal subsidence patterns derived through SAR Interferometry. Ice in the soil pores melts and commonly leads to subsidence throughout the summer. This effect is less pronounced for dryer soils. The initial land-cover map covered Western Siberia with 20 m nominal resolution (Bartsch et al., 2019b). On regional scale, classes have been also matched to vegetation community descriptions. The classification accuracy ranged between 70 and 83.3% for central Yamal (Bartsch et al., 2019b). The approach does however consider bands of Sentinel-2 which have 10 m as well as

20m resolution. Adapted super-resolution processing can be, however, used for transformation to 10m nominal resolution. This has been shown applicable in case for Sentinel-2 for Arctic environments before (Bartsch et al., 2021b). Bartsch et al. (2021b) also demonstrate the feasibility of Sentinel-1/2 for circumpolar processing, but with focus on artificial objects. Circumpolar implementation of a landcover unit retrieval with high detail is, however, still lacking. A map for tundra regions based on Sentinel-1/2 and digital elevation information has been previously published, but with lower thematic content (ten classes,

CALC-2020, (Liu et al., 2023)). Topographic information was used in addition and shown to be the dominating factor in the random forest method based retrieval. In addition, shrub growth patterns which are a key characteristic of tundra landscapes (Raynolds et al., 2019) are not distinguished (Liu et al., 2023).

The purpose of this study was to provide an account of tundra land cover heterogeneity, considering wetness gradients and

80 diversity. (1) Landcover units at comparably high resolution (10m) and thematic content needed to be derived for the entire Arctic north of the treeline (Circumarctic Landcover Units - CALU). (2) Heterogeneity has been assessed with respect to the scale of current global permafrost modelling (1km). (3) The landcover units also have been documented with in situ data regarding their vegetation and soil properties to facilitate further use.

This requires an approach feasible to be implemented for the entire Arctic. A strategy is needed to extent the prototype landcover units as suggested in Bartsch et al. (2019a) which can be distinguished by combining the multispectral and C-band synthetic aperture radar data for representative regions (climatic gradients). The original approach considered only top of atmosphere radiance, 20m resampled data and flat to moderate terrain. The latter allowed the use of $\sigma^0$ for Sentinel-1 with a simplified normalization approach (Widhalm et al., 2018). Mountain regions are included on circumpolar scale, what requires

the use of $\gamma^0$ instead (Small, 2011). Overall, a retraining of the classifier is needed considering the enhanced pre-processing techniques.



## 2 Data

### 2.1 Satellite data

Data acquired from both the Copernicus Sentinel-1 and Sentinel-2 mission have been used for the retrieval. Both missions are
95 part of ESA's Copernicus program. Whereas Sentinel-1 carries synthetic aperture radar systems, Sentinel-2 provides multi-spectral data.

Two satellites were so far used for the Sentinel-1 mission, both with a near-polar, sun-synchronous orbit, 180 degrees apart from each other. The two earth observation satellites Sentinel-1A (launched in April 2014) and Sentinel-1B (launched in April 2016, operation stopped in December 2020) have an identical C-band SAR sensor on board (Schubert et al., 2017). The
100 Interferometric Wide Swath (IW) mode combines a swath width of 250 km with a relatively good ground resolution of $5 \times 20$ m. A pixel spacing of $10 \times 10$ m is commonly used for nominal resolution of derived products. This is also the case for Ground Range Detected (GRD) products as distributed by Copernicus. Information can be captured in dual polarization (HH+HV or VV+VH; H - horizontal, V - vertical). Mostly VV+VH is available for the Arctic land area for this mode and resolution. Greenland and several high Arctic islands are covered in HH+HV mode due to requirements of glacial monitoring.

GRD products are detected, multi-looked, and projected to ground range using an Earth ellipsoid model (ESA, 2012). As temporal variations of backscatter can occur with changes in liquid water content (Bergstedt et al., 2018; Bartsch et al., 2023), only winter data (December and/or January; frozen soil conditions) are used for cross-Arctic consistency and comparability.

The Sentinel-2 constellation has also two twin satellites, Sentinel-2A and -2B, in a sun-synchronous orbit, 180° apart from each other. Sentinel-2A was launched in June 2015 and Sentinel-2B in March 2017. The optical sensor samples 13 spectral
bands. The spatial resolution depends on the used band: four bands have a spatial resolution of 10 m, six bands of 20 m, and three bands of 60 m (ESA, 2015). Bands 3 (green, 10 m), 4 (red, 10 m), 8 (near infrared, 10 m), 11 (SWIR, 20 m), 12 (SWIR, 20 m) have been used for the classification following the prototype scheme (Bartsch et al., 2019a, b).

### 2.2 Landcover prototype

The original prototype covered a transect reaching from the Yamal peninsula into the northern part of the West Siberian
Lowlands (Figure 1). Four bioclimatic zones are covered. The processing is based on Sentinel-1 and Sentinel-2 images with bands sampled to 20m. Top of atmosphere radiance was originally used for Sentinel-2 and normalized $\sigma^0$ for Sentinel-1. 25 classes were considered, including three water classes, but excluding permanent snow/ice and shadows as the analyses region did not include steep mountain or high Arctic areas. The classes were determined with a k-means approach and labeled based on field data and expert knowledge (Bartsch et al., 2019a).



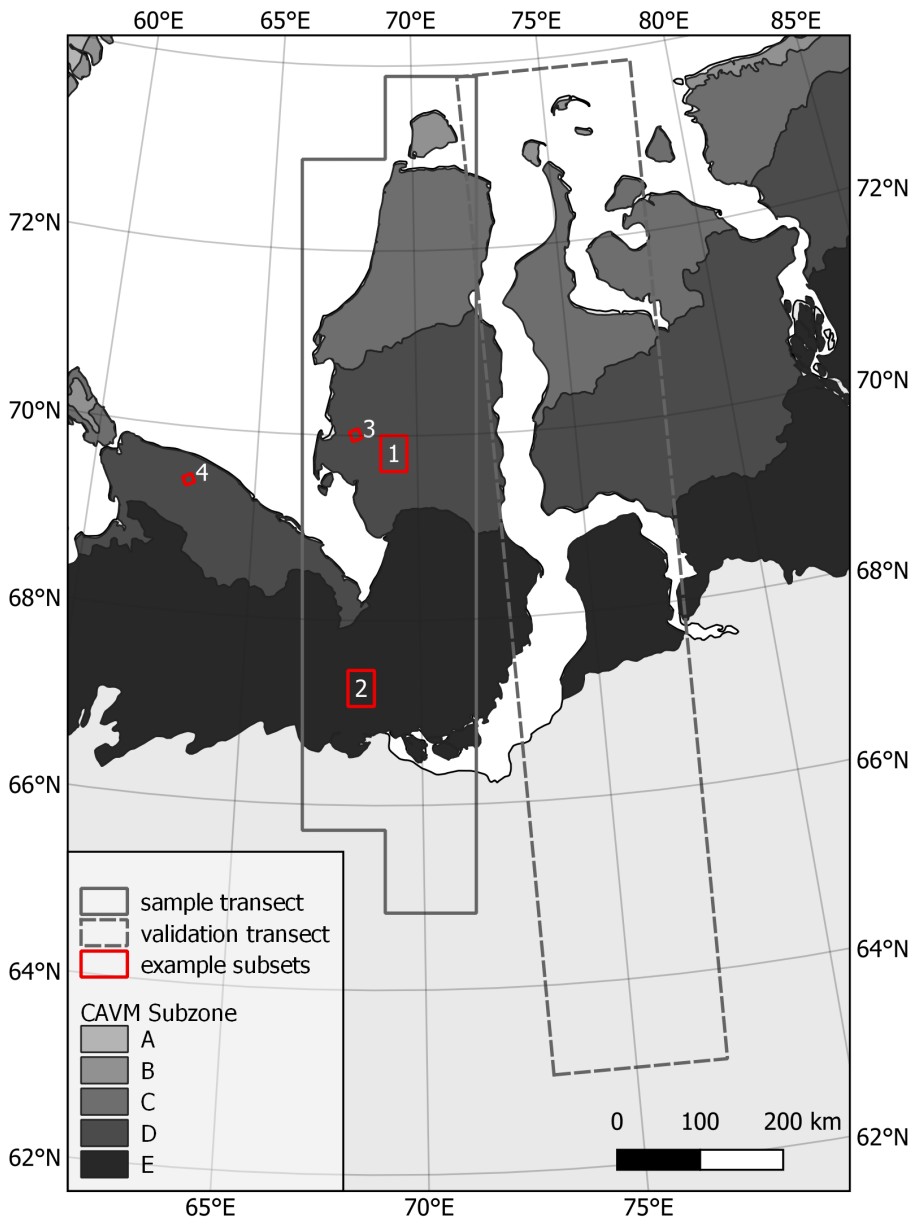

**Figure 1.** Location of the training region for the Maximum Likelihood approach used for transfer to the circumpolar domain based on the prototype (K-Means analyses; 'sample transect'), the evaluation region ('validation transect') and sites of examples of the results presented in Figure 15 (subset 1), Figure 16 (subset 2) and Figure 19 (subsets 3 and 4). CAVM - Circumarctic Vegetation Map (Raynolds et al., 2019) with subzones A) High Arctic tundra, B) Arctic tundra: northern variant, C) Arctic tundra southern variant, D) Northern hypo-Arctic tundra, E) Southern hypo-Arctic tundra.



## 2.3 In situ data

Three types of in situ records are available: (1) full pedon descriptions for key regions, (2) field data of soil surface organic horizon depth available from multiple sites across the northern hemisphere, (3) vegetation coverage records from the Arctic Vegetation Archive for Western Siberia. More than 3500 in situ data records have been used.

355 pedons were available for #1. The field soil sampling took place between 2006 and 2019. Protocols for field sampling, laboratory analyses and data is detailed in Palmtag et al. (2022). 788 non forest samples were available for #2. The data was extracted from several different sources, including (Hugelius et al., 2013, 2014, 2020) and (Palmtag et al., 2022). Soil horizons are defined as surface soil horizons when their organic carbon content is ≥17% (equivalent to ca. 30% organic matter content) (Hugelius et al., 2020). Many of the sites for (1) and (2) overlap.

We used the information from the Arctic Vegetation Archive (AVA standardized protocol, Zemlianskii et al. (2023)), which contains relevés (geobotanical plots) made in accordance with the Braun-Blanquet tradition (available for download at the AVA website, https://avarus.space). The relevés include species lists and species' cover estimations, as well as vegetation and habitat characteristics. They cover square plots with an area of 1-400 m$^2$ (74% cover 100 m$^2$, 12% cover 4 m$^2$, other variants comprise less than 10%). These plots are typically distributed across a 10x10 km site to ensure the representation of various plant communities with a statistically significant number of plots. The AVA data used in this study was obtained during fieldworks conducted from 2007 to 2020. 2705 relevé locations overlapped the analysis extent.

The following in situ measurements are used for the assessment and assignment unit descriptions:

1. sites with full pedon descriptions (Palmtag et al., 2022)

    – Volumetric Water content (%)

    – Organic volumetric content (%)

    – Mineral volumetric content (%)

    – Wet and dry bulk density

    – Soil organic carbon and total nitrogen density

2. sites with soil surface organic horizon depth (cm) (partial overlap with #1)

3. vegetation coverage (%) (from AVA https://avarus.space/profile/about/ , Zemlianskii et al. (2023))

    – Trees

    – Tall shrubs

    – Low shrubs

    – Erect dwarf shrubs

    – Prostrate dwarf shrub





– Graminoids

       – Tussok graminoids

       – Forbs

       – Seedless vascular plants

       – Mosses and liverworts

– Lichen

       – Crust

       – Algae

       – Bare soil

       – Bare rock

– Litter

The AVA dataset (#3) also includes a general wetness description (dry, moist, wet and aquatic).

## 2.4  Auxiliary data

The analyses extent also includes disturbances, specifically wild fire affected areas within the tundra-taiga transition zone, which need to be considered in the assessment. The Alaska Large Fire Database (https://www.frames.gov/catalog/10465)
contains current and historical reported fire locations and fire perimeters. It builds on Kasischke et al. (2002). The database covers all of Alaska and contains about 4600 polygons for fire extent dating back to breakouts in 1942. Polygons which overlap with the analyses extent date back to 2002. The latest fires represented are from October 2021.

Digital elevation data are required for pre- and post-processing of the satellite data. The Copernicus DEM GLO 90 was used. It represents the surface of the Earth including buildings, infrastructure and vegetation at 90m spatial resolution. It covers the
full global landmass of the time frame of data acquisition (2011-2015). The Copernicus DEM is based on the SAR-derived WorldDEM dataset provided by Airbus and acquired by the TanDEM-X mission. According to official statistics published by the Copernicus in DEM Product Handbook overall absolute vertical accuracy of the dataset is 90%, confidence level is > 4 m. For Svalbard the 20m spatial resolution DEM of the Norwegian Polar Institute was used Melvaer (2014). This DEM is based on stereo models of aerial photos. In areas where these were not available the DEM is built on elevation contours.

Air temperature has been derived from ERA5 reanalysis data which combines model data with observations to provide a globally consistent dataset (Hersbach et al., 2023). The data was available from the European Centre for Medium-Range Weather Forecasts (ECMWF) and accessed via the Climate Data Store (CDS). We used air temperature at 2m above the surface in a temporal resolution of 2 hours at 0.25° spatial resolution for selection of Sentinel-1 observations.



**Table 1.** Landcover datasets considered for cross-comparison. CCI - Climate Change Initiative, CAVM - Circumarctic Vegetation Map, CALC - Circumarctic Landcover, CALU - Circumarctic Landcover Units.

| Dataset | C3S/Landcover CCI | CAVM | CALC2020 | LCP | CALU |
|---|---|---|---|---|---|
| **Reference** | Defourny et al. (in preparation) | Raynolds et al. (2019) | Liu et al. (2023) | Bartsch et al. (2019a) | this study |
| **Spatial resolution** | 300m | 1000m | 10m | 20m | 10m |
| **Primary source** | Sentinel-3 OLCI 2019 | AVHRR 2000 | Sentinel-1/2, DEM | Sentinel-1/2 | Sentinel-1/2 |
| **Extent** | Global | North of treeline | As defined in CAVM | Western Siberia | As defined in CAVM |
| **Classes/Units** | 38 | 16 | 10 | 21 | 23 |

## 2.5 Existing land cover information for comparison

Two circumpolar and one global landcover dataset have been compared to the landcover units (Table 1). The spatial resolution ranges from 10m to 1000m. The raster version of the CAVM (Circumpolar Arctic Vegetation Map, Raynolds et al. (2019)) is considered a key benchmark dataset as it has been developed specifically for the Arctic by vegetation experts. It provides a similar level of detail regarding shrub types and soil wetness.

The CAVM also includes information on bioclimatic subzones. In total six subzones are distinguished for the Arctic of which

five can be found along the validation and calibration transect (Figure 1).

## 3   Methods

In a first step, a landcover unit retrieval scheme has been adapted in order to achieve 10 m nominal resolution, consider atmospheric correction, address issues in mountainous regions and in order to allow additional classes such as recent fire scars, snow and shadow. Results have been cross-compared to results of the original processing scheme and to external datasets

including classical landcover maps and soil and vegetation in situ records.

Statistics have been derived for 1km x 1km areas in a second step in order to quantify landcover diversity in general and specifically for wetland areas.

### 3.1   Landcover units retrieval scheme adaption

The retrieval schemes expands the prototype (Bartsch et al., 2019b) to the entire Arctic, incorporating advanced pre-processing

and a postprocessing step is introduced. The final classes are referred to as landcover units, the entire map as CALU - Circumarctic Landcover Units.

The processing has been based on Sentinel-2 granules as defined by the data provider. Granules have an extent of 100 km by 100 km. They partially overlap as they are aligned with respect to UTM projection zones. All Sentinel-1 images have been subset to the granules.

Sentinel-2 data are available in UTM projection and largely at Level 2A (orthorectified, top of atmosphere). Atmospheric correction is required in order to account for related differences between the dates. We therefore applied atmospheric correc-





tion using sen2cor on the Sentinel-2 data, which also generates a cloud mask during the process. Sentinel-2 provides spatial resolution of 10 m for some bands but not for all. Enhancement of spatial resolution of the coarser band therefore needs to be considered to exploit the multi-spectral capabilities offered by Sentinel-2. We therefore performed super-resolution based on the tool Dsen2 (Lanaras et al., 2018), which uses a convolutional neural network. Lanaras et al. (2018) showed that their approach clearly outperforms simpler upsampling methods and better preserves spectral characteristics. The original model was trained on Level-1 data, which have not been atmospherically corrected, and global sampling. Bartsch et al. (2021a) retrained and tested the model on Level-2 data (output of sen2cor) using the same published training and testing routines for selected granules from our study sites. After the super-resolution step, clouds were masked using the cloud mask output from sen2cor. In case of frequent fractional cloud cover also subsets of scenes have been used. Mostly mid-growing season acquisitions have been considered for Sentinel-2, this means mid July to mid August. This time frame has been slightly extended in some cases of lack of cloud free acquisitions. Up to eight granules have been considered. In a next step the median of three, if available, acquisition dates are calculated which further mitigates errors due to undetected clouds.

Acquisitions need to represent frozen conditions in case of Sentinel-1, but at maximum -10°C to minimize the effect of temperature on backscatter at C-band (Bergstedt et al., 2018; Bartsch et al., 2023). This requires the use of spatially consistent temperature data across all analysed areas. Reanalyses data (ERA5) was therefore used for scene selection. Processing steps of Sentinel-1 include border noise removal, based on the bidirectional all-samples method of Ali et al. (2018), calibration, thermal noise removal and orthorectification using the Copernicus 90m resolution DEM. These steps are carried out with the SNAP toolbox provided by ESA and $\gamma^0$ is derived. After normalization, data has been reprojected and subset to match the Sentinel-2 granules and temporal averaging has been performed.

A transect spanning nine degrees in latitude and representing different landscape gradients and thus all units (Figure 1) has been selected from the prototype landcover unit map in order to transfer the retrieval (re-training of the maximum likelihood classifier) to the entire Arctic as well as to the output of the enhanced preprocessing (super-resolution processing, atmosphere correction and use of $\gamma^0$ instead of $\sigma^0$). The units remained largely the same. Additional training data have been included in case of the disturbance unit. This unit originally combined different disturbance types which lead to vegetation removal, geomorphological processes as well as fires. Training data from recently burned areas were included into the training dataset for a separate class. The originally three water units have been merged into one water unit. Further on, a snow unit has been introduced based on training data from Svalbard.

Illumination conditions impact the reflectance in the Sentinel-2 bands in mountainous areas. This has been addressed in two steps. First a other/shadow unit was introduced based on training data obtained over Svalbard. Not all shadow areas can be however identified due to similarities in reflectance over water bodies and wetlands. Therefore a second post-processing step based on the slope derived from the Copernicus DEM data was introduced. 'Water', 'Permanent wetlands' or 'shallow water' on slopes larger than 7° have been set to 'other'. Shadow in regions with low slope (less than 3°) has been set to 'water'.

## 3.2 Evaluation and documentation of landcover units

Unit specific validation was carried out for the new disturbance unit which targets burned areasusing the Alaska Fire Database.





The description of the units is based on the comprehensive soil and a vegetation in situ dataset. The first has been previously used similarly in conjunction with a global landcover map (Palmtag et al., 2022). The second data set is part of a tundra specific international standardization and archive effort (Zemlianskii et al., 2023). Common statistics (mean, standard deviation) have been derived and are supplied with the dataset documentation.

Unit descriptions consider abundance of vegetation types including shrub growth form/height as well as moisture conditions. The following shrub types are considered (following the differentiation of the CAVM, Raynolds et al. (2019) and Walker et al. (2018)):

- Prostrate dwarf shrub – approximately 5 cm, also referred to as prostrate shrub

- Erect dwarf shrub – up to 40 cm, also referred to as dwarf shrub

- Low shrub – up to 2m

- Tall shrubs – tundra biome species taller than 2m

The comparison to Landcover CCI and the CAVM (see table 4) requires resampling. The coarser resolution maps have been resampled to 10m (as CALU). Classes have been grouped for comparison due to the large differences in thematic content. The translation tables are provided in the supplement. In addition, the new classification has been compared with the original
prototype (20m, 21 classes). A different transect has been used for the evaluation than for the re-training of the transfer samples (Figure 1).

A direct comparison can only be made for common or merged classes which include water, snow/ice, other, wetland, grassland, lichen/moss, shrub tundra, forest and barren. The grouping of CALU for the comparison has been primarily guided by the presence of shrubs. If more than 20% of erect dwarf to low shrubs are found on average the class is assigned to shrub tundra.
The assignment is referred to as grouping A (Table 2).

## 3.3 Heterogeneity assessment

Landscape heterogeneity is eventually addressed through assessment of (1) richness with respect to the CAVM classification specifications and (2) wetness diversity, both for 1km grid cells. The chosen cell size matches the grid of the permafrost model CryoGRID (Obu et al., 2021) as well as the rater version of the CAVM (Raynolds et al., 2019). The number of identified units
(minimum 1% fraction) within 1km x 1km cells has been counted for the CAVM classes. Permanent snow and shadow have been excluded from the richness assessment. In addition, units have have been grouped with respect to wetness (wet, moist, dry) for wetland and tundra classes, also excluding permanent snow and shadow as well as water, forest and recently burned areas. Groups (referred to as B, Table 2) have been summed up resulting in values of one (homogeneous) to three (heterogeneous).



## 4 Results

 ### 4.1 Coverage

1266 Sentinel-2 granules which overlap with the Arctic as identified in the CAVM (Raynolds et al., 2019) have been identified (Figure 2). The use of three Sentinel-2 acquisitions was targeted to account for anomalies due to undetected clouds and hydrological extremes (droughts, flooding) what could be achieved in 75% of the cases (Figure 3). Only two images were available for 4% of all granules. More than 3600 Sentinel-2 images at granule extent have been included in the processing.

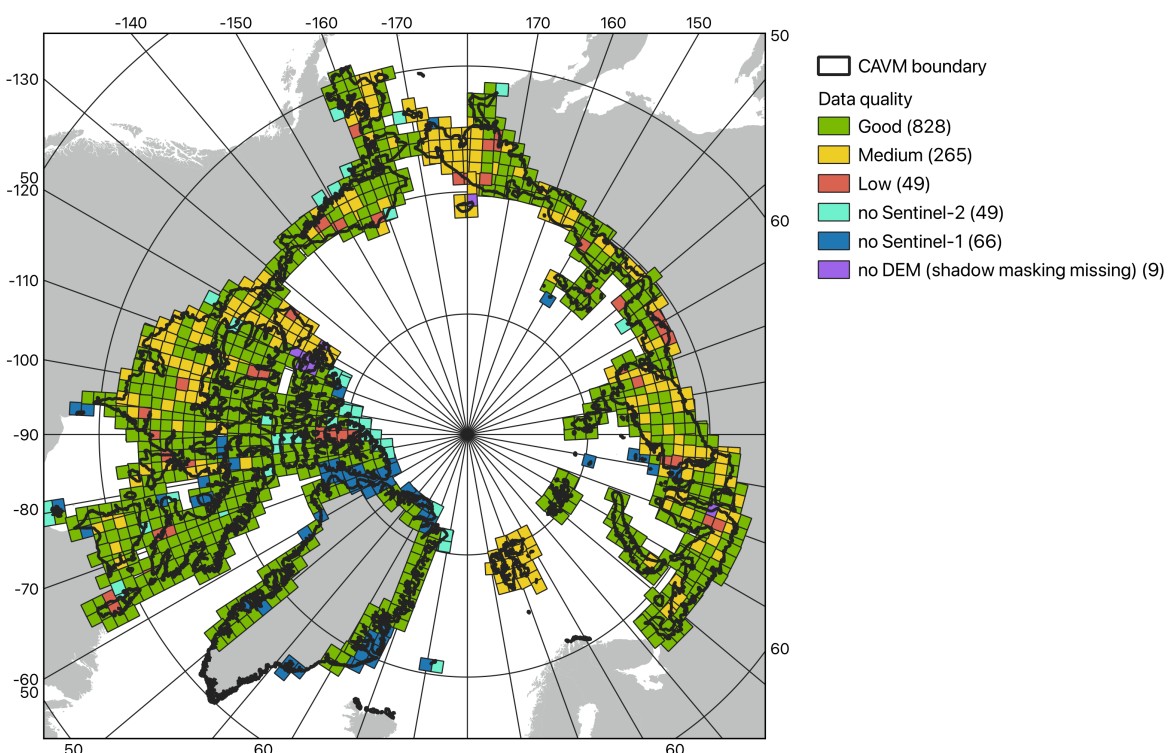

**Figure 2.** Data quality by granules within CAVM boundary

 No suitable Sentinel-2 acquisitions have been available in 52 (4%) cases and in addition 66 (5%) cases with no Sentinel-1 data (Figure 2). The latter is related to the general Sentinel-1 acquisition strategy for IW mode. Data are missing specifically for Greenland and the Canadian High Arctic islands. Nevertheless, 97% of the target area (CAVM extent) could be processed. In 13% of the area the availability of data was very limited and led to lower quality results or gaps (Figure 2 and 3). This results from acquisition date issues or inconsistencies in the used Copernicus DEM (no data: 23 times in case of some inland  water bodies and 2 times general gaps). 19% have been flagged as medium and 3% as low quality. Medium quality usually





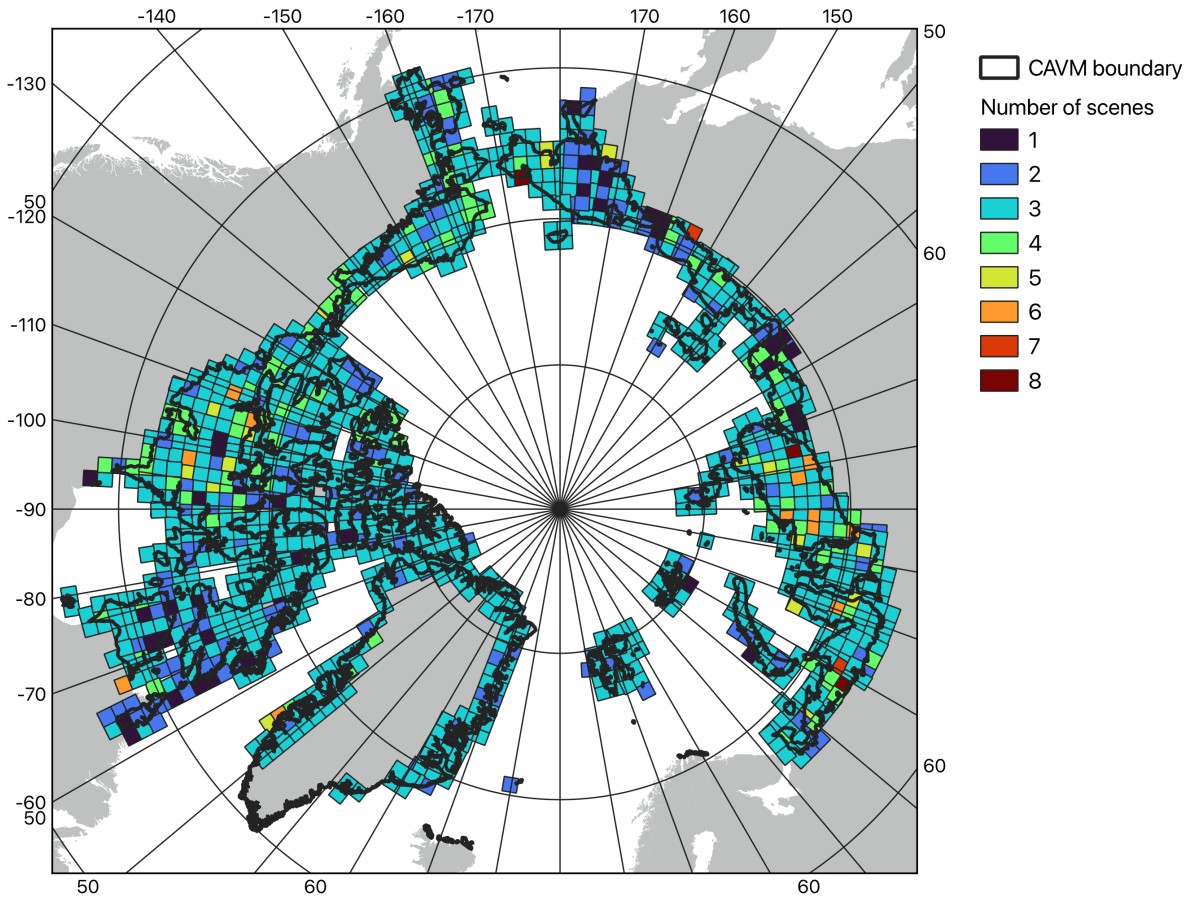

**Figure 3.** Number of used scenes by granules within CAVM boundary

results from anomalous meteorological/hydrological conditions in certain years, including flooding or droughts which impact landcover and vegetation optical properties. Low quality is usually caused by acquisitions too late or too early in the season.

## 4.2 Identified landcover units

23 unit types are eventually derived for the CALU (Circumarctic Landcover Units) map (Table 2). Descriptions are assigned
summarizing the statistics of the in situ data, referring to wetness, vegetation physiognomy and abundance. The most common unit within the extent of the CAVM is unit #6 'dry to moist tundra, partially barren, prostrate shrubs' with overall 22.5% (original 10x10m) and 28.5% in the majority (1x1km) respectively. This unit can be specifically found in the Canadian High Arctic and on the Taimyr peninsula (Russia). Units with shrubs of dwarf and higher growth form sum up to about 32% and wetlands to about 9%. A higher occurrence in the majority retrieval indicates relatively homogeneous patterns. This applies





**Table 2.** Classes of CALU and A) grouping for comparison, B) grouping for heterogeneity assessment, P) proportion within CAVM extent in % M) majority proportion (1km grid cells) in %. * sparse tree cover along treeline. Full documentation is available in Appendix A.

| ID | Description | Gr. A | Gr. B | P | M |
|----|-------------|-------|-------|---|---|
| 1 | water | Water | - | 5.9 | 7.1 |
| 2 | shallow water / abundant macrophytes | Wetland | wet | 2.9 | 0.8 |
| 3 | wetland, permanent | Wetland | wet | 2.6 | 1.5 |
| 4 | wet to aquatic tundra, abundant moss | Wetland | wet | 3.5 | 3.4 |
| 5 | moist to wet tundra, abundant moss, prostrate shrubs | Grassland | moist | 1.3 | 0.9 |
| 6 | dry to moist tundra, partially barren, prostrate shrubs | Lichen/Moss | dry | 22.5 | 28.5 |
| 7 | dry tundra, abundant lichen, prostrate shrubs | Lichen/Moss | dry | 3.5 | 2.4 |
| 8 | dry to aquatic tundra, dwarf shrubs* | Shrub tundra | moist | 0.9 | 0.2 |
| 9 | dry to moist tundra, prostrate to low shrubs | Shrub tundra | moist | 7.5 | 9.2 |
| 10 | moist tundra, abundant moss, prostrate to low shrubs | Shrub tundra | moist | 6.0 | 7.0 |
| 11 | moist tundra, abundant moss, dwarf and low shrubs | Shrub tundra | moist | 8.5 | 11.6 |
| 12 | moist tundra, dense dwarf and low shrubs* | Shrub tundra | moist | 1.3 | 0.7 |
| 13 | moist to wet tundra, dense dwarf and low shrubs* | Shrub tundra | moist | 0.2 | 0.02 |
| 14 | moist tundra, low shrubs | Shrub tundra | moist | 2.9 | 2.0 |
| 15 | dry to moist tundra, partially barren | Shrub tundra | moist | 2.6 | 1.3 |
| 16 | moist tundra, abundant forbs, dwarf to tall shrubs | Shrub tundra | moist | 1.6 | 1.5 |
| 17 | recently burned or flooded, partially barren | Shrub tundra | - | 0.6 | 0.1 |
| 18 | forest (deciduous) with dwarf to tall shrubs | Forest | - | 0.3 | 0.2 |
| 19 | forest (mixed) with dwarf to tall shrubs | Forest | - | 0.5 | 0.3 |
| 20 | forest (needle leave) with dwarf and low shrubs | Forest | - | 0.1 | 0.1 |
| 21 | partially barren | Barren | dry | 14.5 | 16.3 |
| 22 | snow/ice | Snow/ice | - | 2.1 | 3.4 |
| 23 | other (incl. shadow) | Other | - | 2.3 | 1.6 |
| nd | - | - | - | 6.1 | - |

specifically to class # 6 as well as #21 (partially barren). The contrary is the case for wetland classes #2 and #3 reflecting occurrence patches. Regarding wetness, 40% are assigned to the dry and 32% to the moist group.

### 4.3 Characterization through in situ records

All classes can be assigned distinct vegetation compositions and water/mineral volumetric content (Table 3, Figures 4 and 5). There are differences in in situ data availability between the landcover units which need to be considered. Soil data are largely
unavailable for forest and disturbed sites (Tables A2, A1). The dry tundra unit #7 which has the most abundant lichen coverage has only little pedon data but good vegetation description. Unit #13 lacks both pedon and vegetation description as it represents vegetation along incised creek channels (moist tundra, dense dwarf and low shrubs, Figure B40) which are rarely sampled. Soil samples for inundated areas (unit 2) are unavailable. Vegetation samples for this class are expected to be located at the boundary of shallow water bodies with macrophytes, but in the same pixel. They have been nevertheless included in the tables




and figures. The lack of soil wetness description for some units can be partially compensated by general wetness information contained in the vegetation records (Figure 6).

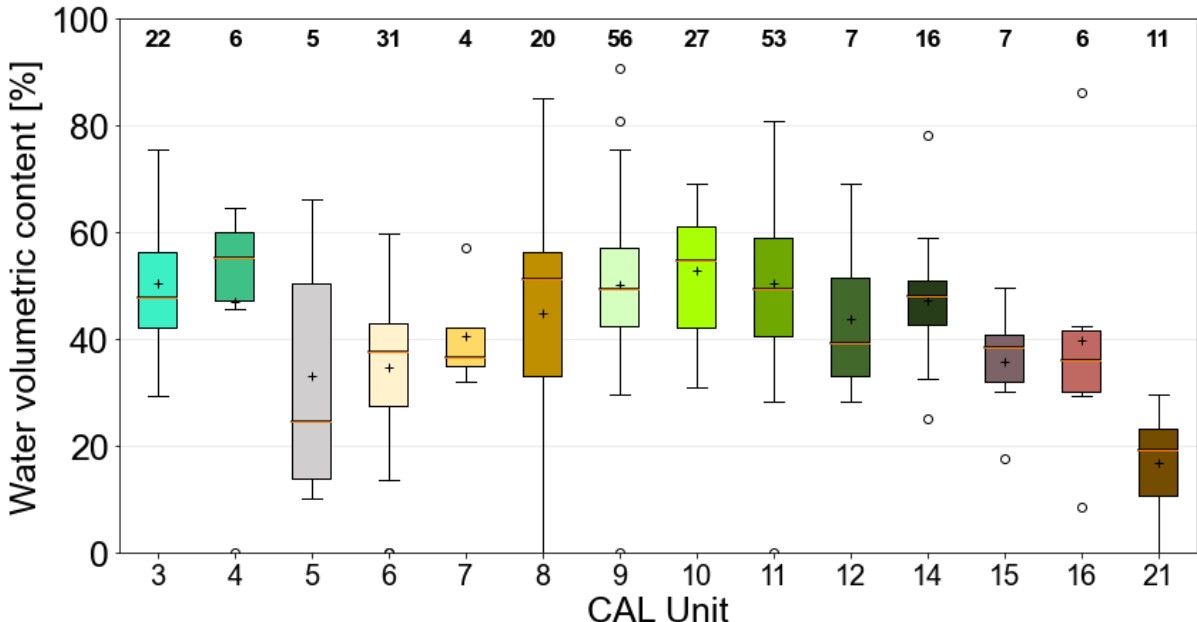

**Figure 4.** Volumetric water content statistics based on (Palmtag et al., 2022).

### 4.3.1 Vegetation

Mosses and graminoids are common across all units with moss being most dominant (48% overall average) and graminoids second (20% overall average) (Table 3). Lichens occur on average 12% and are reaching more than 20% only for units with
300 comparably dry parts. Forbs are less abundant except for one case, unit #16 for which the average is 41%. Mosses also show the highest standard deviation within the landcover units (Figure 7).

The 'Barren' unit #21 has the highest bare ground fraction with on average 32% for soil and 2% for bare rock. Bare rock is most common in unit #6 with 11%.

Unit #2 represents shallow water along lakes and seashores which is are not represented in the AVA or soil records. Macro-
305 phytes are abundant, emerging through the course of the summer season (see photograph B2).

Several classes with tall shrubs in proximity to the treeline also can include needle leave trees (8, 12 and 13, see Table 3). The latter are of limited height and also diameter due to the harsh environment and is common for the North American treeline zone (Pictures B23 and B41). Areas correspond to the 'woodlands' and 'open stands' definition level III according to Viereck et al. (1992) (10 to 24% crown canopy cover, and 24 to 60% respectively).



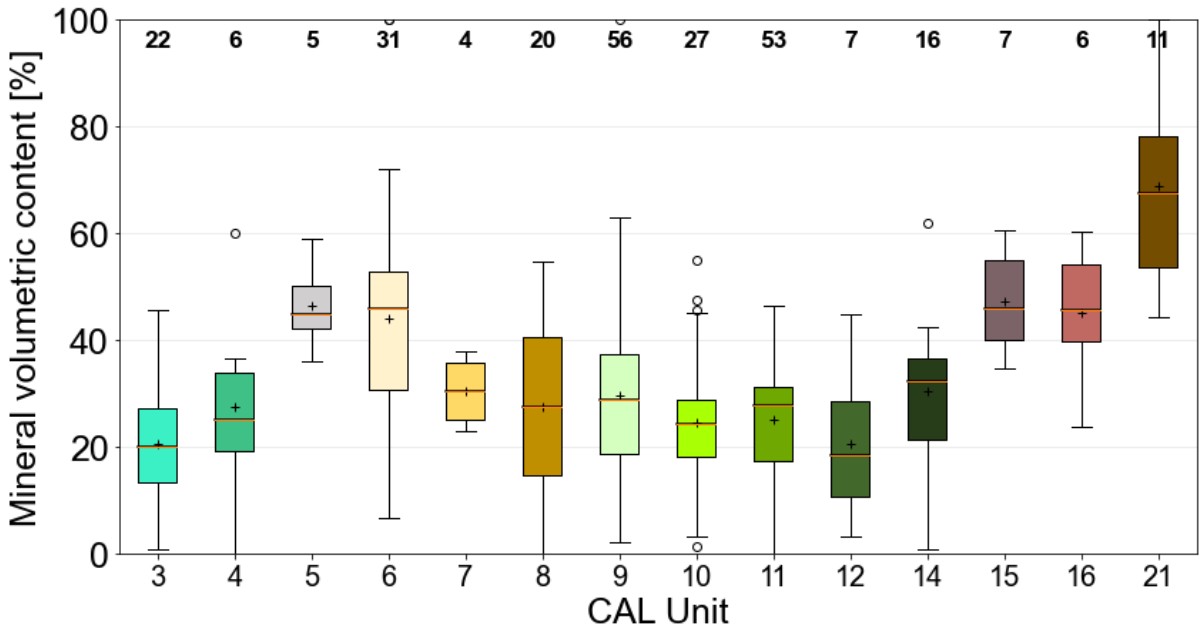

**Figure 5.** Mineral volumetric content statistics based on (Palmtag et al., 2022).

**Table 3.** Average coverage for each unit for Arctic Vegetation Archive types, Western Siberia.

| Unit | 2 | 3 | 4 | 5 | 6 | 7 | 8 | 9 | 10 | 11 | 12 | 13 | 14 | 15 | 16 | 17 | 18 | 19 | 20 | 21 | Mean |
|---|---|---|---|---|---|---|---|---|---|---|---|---|---|---|---|---|---|---|---|---|---|
| Samples | 51 | 56 | 47 | 68 | 412 | 142 | 85 | 354 | 348 | 293 | 78 | 20 | 193 | 95 | 181 | 7 | 29 | 18 | 45 | 95 | **131** |
| Tree layer | 0 | 1 | 0 | 0 | 0 | 0 | 8 | 0 | 0 | 0 | 2 | 4 | 1 | 2 | 1 | 4 | 28 | 7 | 43 | 0 | 5 |
| Tall shrubs | 3 | 3 | 0 | 0 | 0 | 0 | 2 | 1 | 0 | 1 | 2 | 5 | 1 | 0 | 10 | 0 | 11 | 26 | 2 | 0 | 3 |
| Low shrubs | 4 | 2 | 1 | 1 | 5 | 3 | 8 | 11 | 7 | 19 | 36 | 23 | 21 | 4 | 17 | 6 | 13 | 26 | 10 | 3 | 11 |
| Erect dwarf shrubs | 5 | 2 | 7 | 1 | 5 | 5 | 19 | 6 | 6 | 12 | 20 | 11 | 7 | 11 | 9 | 15 | 18 | 18 | 49 | 1 | 11 |
| Prostr. dwarf shrubs | 11 | 6 | 9 | 14 | 11 | 13 | 6 | 13 | 15 | 10 | 5 | 8 | 6 | 8 | 4 | 0 | 1 | 1 | 1 | 10 | 8 |
| Graminoids | 18 | 30 | 30 | 27 | 20 | 14 | 22 | 20 | 24 | 18 | 9 | 24 | 18 | 19 | 17 | 30 | 19 | 19 | 5 | 9 | 20 |
| Tussok graminioids | 3 | 1 | 2 | 1 | 3 | 2 | 1 | 4 | 6 | 6 | 2 | 2 | 2 | 4 | 2 | 0 | 2 | 2 | 0 | 1 | 2 |
| Forbs | 12 | 7 | 9 | 4 | 11 | 2 | 10 | 6 | 9 | 9 | 9 | 19 | 12 | 12 | 41 | 12 | 29 | 12 | 9 | 9 | 12 |
| Seedless vasc. pl. | 3 | 3 | 0 | 1 | 1 | 1 | 3 | 1 | 2 | 1 | 1 | 3 | 2 | 1 | 4 | 0 | 11 | 5 | 1 | 5 | 2 |
| Mosses and liverw. | 33 | 37 | 74 | 65 | 41 | 33 | 40 | 54 | 71 | 63 | 54 | 43 | 58 | 38 | 43 | 43 | 37 | 54 | 62 | 17 | 48 |
| Lichen | 8 | 15 | 5 | 11 | 16 | 27 | 23 | 14 | 9 | 10 | 13 | 11 | 12 | 26 | 3 | 0 | 3 | 5 | 17 | 7 | 12 |
| Crust | 3 | 9 | 2 | 0 | 2 | 6 | 1 | 2 | 1 | 1 | 1 | 1 | 1 | 1 | 0 | 0 | 1 | 1 | 0 | 2 | 2 |
| Algae | 0 | 2 | 0 | 0 | 1 | 1 | 0 | 0 | 0 | 0 | 0 | 0 | 1 | 0 | 0 | 0 | 0 | 0 | 0 | 0 | 0 |
| Bare soil | 9 | 4 | 2 | 1 | 3 | 11 | 3 | 3 | 1 | 1 | 4 | 1 | 2 | 4 | 2 | 21 | 0 | 3 | 0 | 32 | 5 |
| Bare rock | 3 | 1 | 0 | 0 | 11 | 1 | 2 | 2 | 1 | 1 | 2 | 3 | 3 | 6 | 2 | 0 | 0 | 0 | 0 | 2 | 2 |
| Litter | 9 | 22 | 14 | 24 | 8 | 10 | 22 | 13 | 18 | 11 | 14 | 15 | 11 | 9 | 17 | 25 | 43 | 15 | 22 | 3 | 16 |





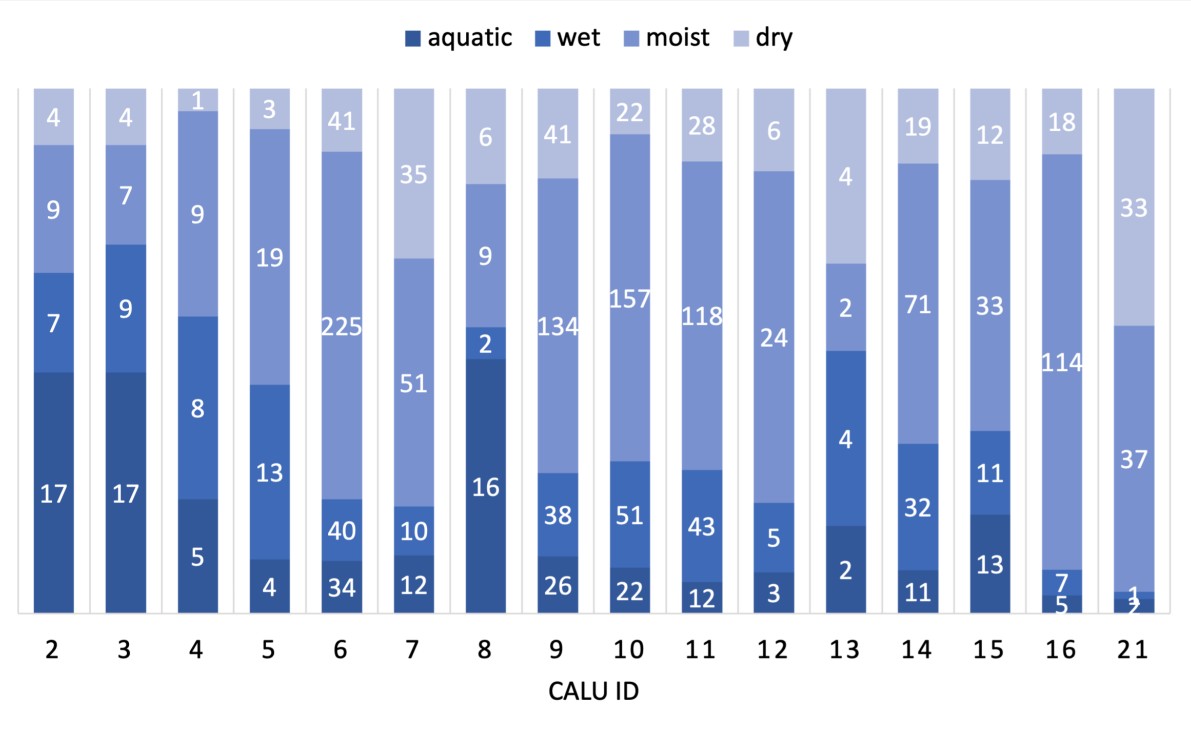

**Figure 6.** Western Siberia Arctic Vegetation Archive wetness categories (only units with at least 10 data points).

### 4.3.2 Soils

The units with a certain barren fraction (#6, #15 and #21, see Table 3 ) show all comparably high mineral volumetric content. Higher wetness (>40% volumetric content on average) can be found in permanent wetlands and several moist tundra units with differing types of shrub physiognomy. The permanent wetland class has one of the lowest standard deviations, the seasonal wetland class one of the highest representing homogeneous and heterogeneous conditions respectively. The number of available samples is, however, too low for generalization. The 'barren' unit is most dry. Unit #16 shows the highest total nitrogen density values (Table A2 with at the same time comparably high mean organic layer thickness (Figure 8, Table A1) but only 6 pedon samples are available.

SOC and TN are lowest for the 'Barren' unit (Table A2). Unit 10 and 11 (moist tundra, abundant moss and shrubs) show the highest SOC values with more than $35 kgCm^2$.

AVA records which include wetness descriptions show in sum more than 50% in the categories wet and aquatic in units 3, 4 and 8 (Figure 6). Unit 15 (dry to moist tundra, partially barren) also has a comparably high fraction of moist and aquatic samples. The AVA records complement the soil description for specifically unit #6 (only four pedon records). The AVA records confirm that it is a relatively dry landcover type.



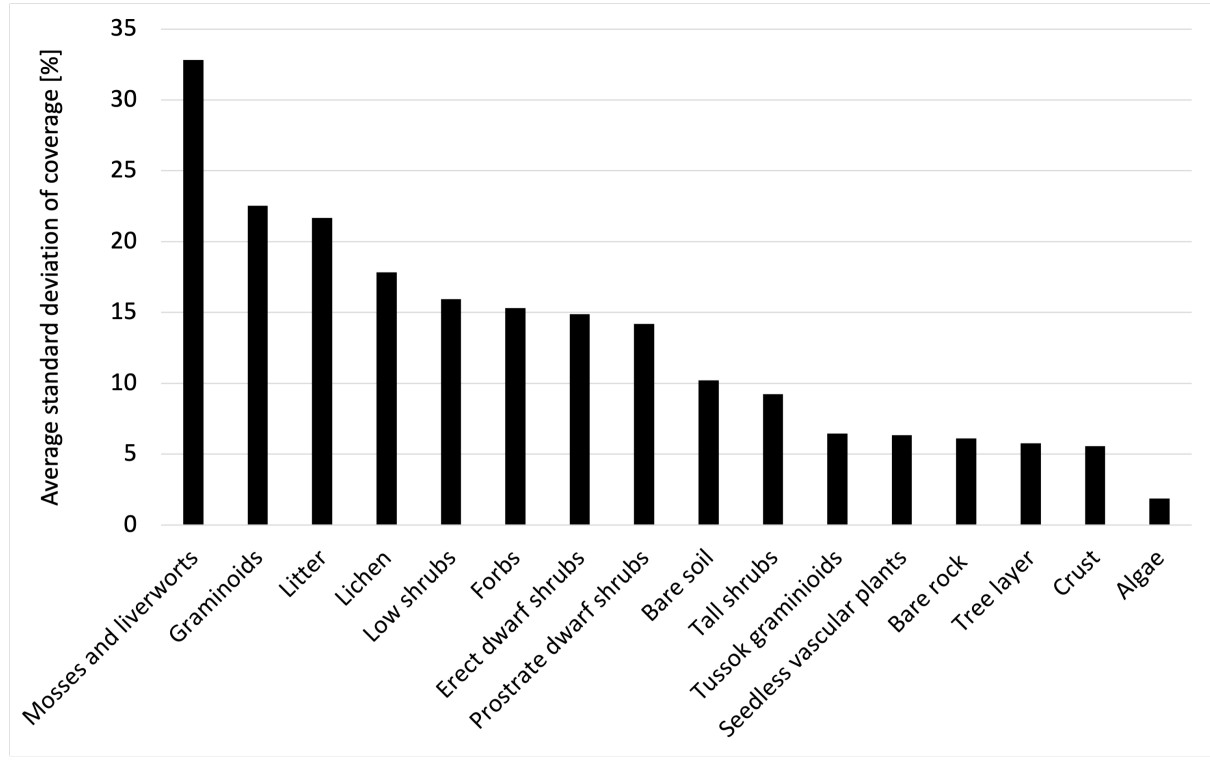

**Figure 7.** Western Siberia Arctic Vegetation Archive average standard deviation of coverage within CAL units.

## 4.4 Fire disturbance assessment

The majority of input data for the landcover classification has been acquired between 2017 and 2020. The assignment to the disturbance unit #17 does occur up to four years after a burn event (Figure 9). A specific year cannot be determined in most cases due to averaging over up to three years. Burned areas from events before 2014 are represented through other classes (vegetation recovery).

The disturbance unit #17 occurs within recently burned areas in approximately 72 % of all cases. The majority of remaining 330 unit #17 areas occurs along shorelines of lakes with varying extent. This is specifically the case on the Alaskan North Slope (Figure 10).

## 4.5 Heterogeneity

Landcover diversity (richness based on fractions - at least 1 percent - within 1x1km areas) is lowest in the higher Arctic and higher towards south (Figure 11). The average number of landcover units ranges from 4 to 9, depending on the majority unit 335 type. The barren unit dominated regions have the lowest richness (Figure 12).



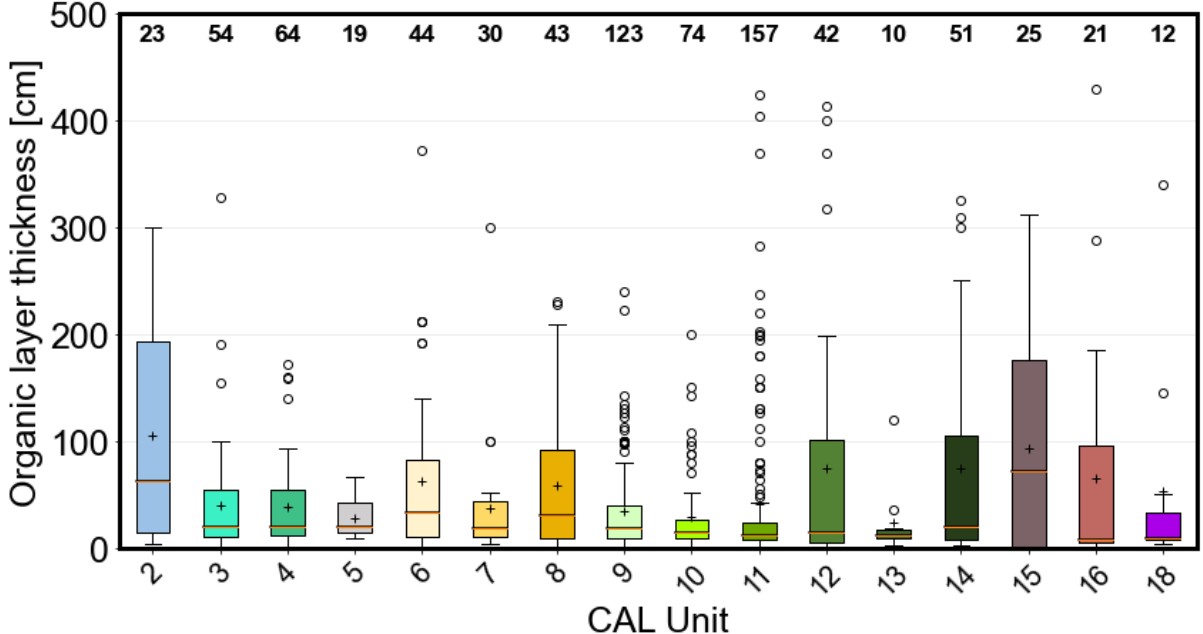

**Figure 8.** Soil surface organic horizon depth statistics for classes with at least 10 data points.

The CAVM class W3 (Sedge, moss, low-shrub wetland complex) shows the highest diversity with almost 9 units on average (Figure 13). In general, wetter regions have a higher diversity than drier ones. This can be also confirmed through the wetness group assessment. In 59% of cases when wet types are present, dry and moist can be found at the same time (Figure 14 right). 45% of the CAVM extent is heterogeneous regarding wetness (group sum three) and only 10% homogeneous (group sum one) (Figure 14 left). 81% of the CAVM extent has at least 1% wetland within the 1x1km cells (considering the sum of all types), but only 0.7% are cells with pure wetlands. The average wetland fraction across the entire CAVM region is 8.8%.

## 4.6 Comparison with existing datasets

Distinct difference can be found between CALU and other landcover maps which do not only relate to the spatial resolution difference (Figures 15 and 16). Wetland areas are considerably more extensive in the CALC-2020 (which is the only classification largely driven by terrain information) than in all other maps, including CALU.

### 4.6.1 Prototype

Results well demonstrate the resolution difference between the prototype with 20m and the CALU with 10m (Figure 17). The patchiness of shrub tundra leads to a reassignment to other vegetation classes. About 15% of the shrub tundra group have been reassigned to other groups, mostly to wetlands (5%) and grassland/lichen/moss (5%). The new disturbance classes result in




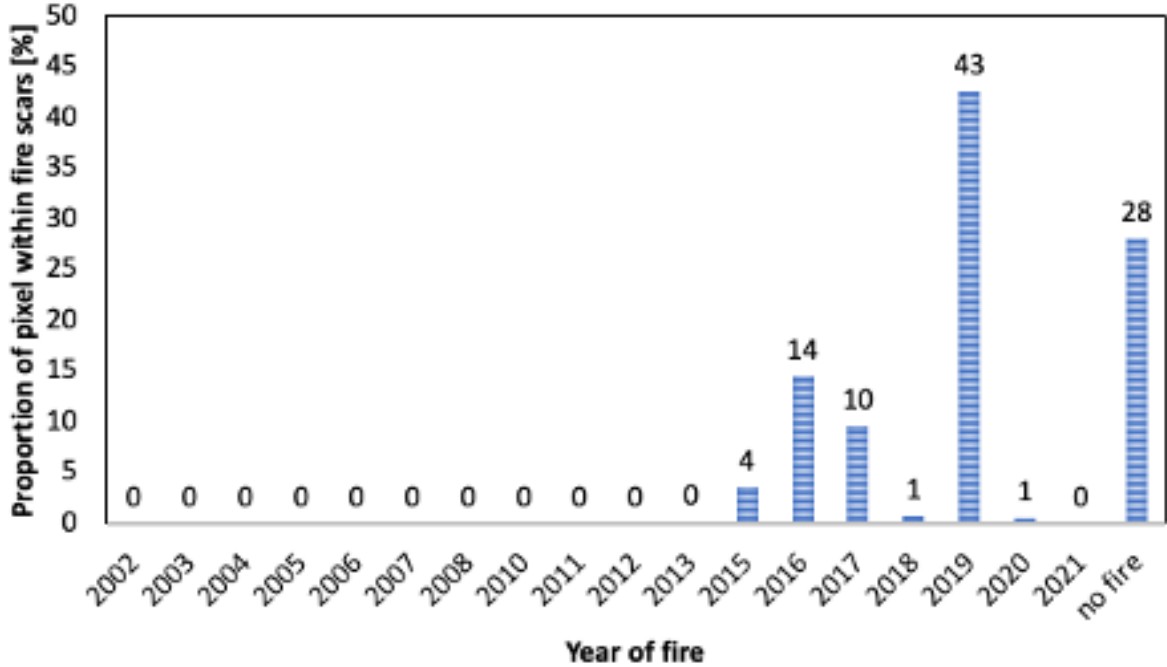

**Figure 9.** Year of fire for pixels in unit 17 (recently burned or flooded, partially barren) for Alaska (source: Alaska Fire database)

a shift of area from lichen/moss to the group barren (which is largely only partially barren according to description of CAL units). Pixels originally classified as disturbed have been assigned to vegetation classes.

### 4.6.2    Landcover CCI

The barren group of CCI Landcover is predominant in all tundra vegetation groups of CALU (Figure 17). In case of the lichen/moss group of CALU, less than 30% are in the same group in CCI Landcover. Also the shrub and forest groups are

most abundant in shrub tundra and forest respectively. More than 88% percent of the CCI Landcover group barren are actually characterized by vegetation (example maps in Figures 15 and 16). 63% are shrub tundra and about 9% grassland and 16% lichen/moss (Table 11).

About 80% of the CCI Landcover class forest are in the shrub tundra group of CALU. The overlap applies specifically to unit 12 'dry to moist tundra, dense dwarf and low shrubs'. Low shrubs are abundant and very dense in the proximity to the

treeline (see picture in B38).

### 4.6.3    CAVM

The shrub tundra group of the CAVM is present with more than 40% in all CALU groups (Figure 17). This can be attributed largely to the spatial resolution difference. The spatial patterns are, however, similar (Figures 15 and 16)). Overlapping areas





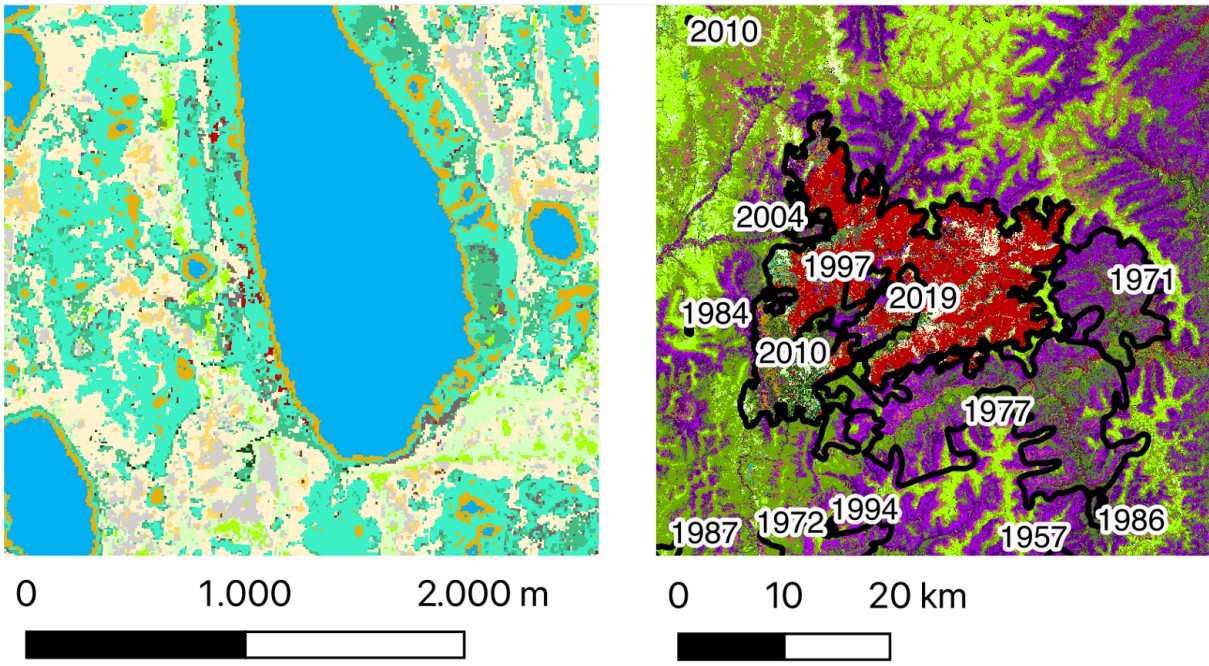

**Figure 10.** Examples for class 17 'recently burned or flooded' (shown in red). Left: Thaw lake on the Alaskan North Slope; right: Burned areas in southwestern Alaska (outlines and years from Alaska Fire Database).

**Table 4.** Comparison matrix of grouped CCI Landcover classes and grouped CAL Units. Values in percent. For grouping schemes see Tables 2 and C1.

| | | CCI Landcover | | | | | | | | |
|---|---|---|---|---|---|---|---|---|---|---|
| | | water | snow/ice | other | wetland | grassland | lichen/ moss | shrub tundra | forest | barren |
| | water | 65.50 | 0.45 | 28.74 | 7.65 | 0.18 | 2.07 | 0.68 | 1.35 | 1.24 |
| | snow/ice | 0.50 | 9.68 | 0.53 | 0.10 | 0.01 | 0.03 | 0.01 | 0.02 | 0.09 |
| | other | 0.65 | 13.69 | 0.24 | 0.65 | 0.64 | 0.67 | 0.51 | 0.59 | 1.14 |
| CALU | wetland | 15.26 | 1.14 | 13.55 | 20.65 | 2.12 | 16.43 | 4.90 | 4.77 | 8.76 |
| | grassland | 0.32 | 0.00 | 0.03 | 0.24 | 1.44 | 4.20 | 0.52 | 0.56 | 9.47 |
| | lichen/moss | 2.60 | 47.97 | 0.10 | 2.57 | 1.28 | 58.38 | 2.04 | 1.64 | 15.97 |
| | shrub tundra | 13.53 | 1.21 | 40.25 | 49.63 | 93.09 | 13.30 | 82.50 | 79.40 | 60.09 |
| | forest | 0.95 | 0.85 | 16.51 | 18.33 | 1.13 | 0.02 | 8.71 | 11.54 | 0.46 |
| | barren | 0.69 | 25.01 | 0.05 | 0.18 | 0.11 | 4.90 | 0.12 | 0.14 | 2.80 |

with the group forest correspond mostly to shrub tundra as the CAVM does not have a forest class. Less than 1% of CAL within
365 the CAVM boundary has been identified as mixed or needle leaf forest.



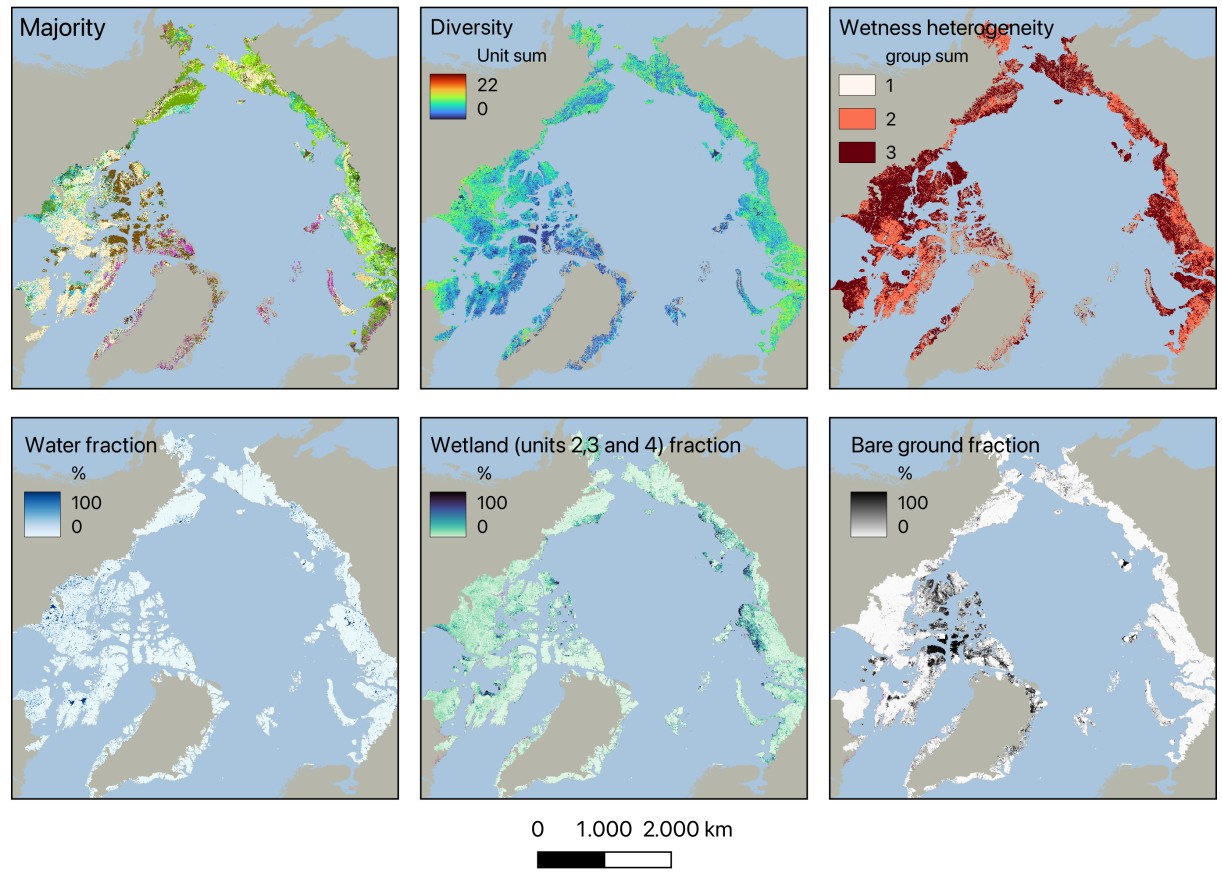

**Figure 11.** Statistics for 1x1km areas. For legend of majority, groups and unit IDs see Table 2.

### 4.6.4 CALC-2020

There is a substantial mismatch between the two datasets for the group 'shrub tundra' and 'wetland' (Figure 17). About 80% of this group has been labeled as grassland and most of the remaining part to wetland in the CALC-2020. Large proportions of wetlands are also found in the CALU groups lichen/moss as well as forest as a larger proportion of the landscape.

A comparison of CALC-2020 with the CAVM has been made for further investigation of the differences (Figure 18). Similar to CALU, CALC-2020 grassland is found most abundant in the CAVM group shrub tundra. Also the proportion of the wetland class in the groups grassland and shrub tundra are high. The analyses by CALC-2020 group (Figure 18b) shows a high proportion of CAVM shrub tundra in all groups and especially grassland. The spread of tundra shrubs across all classes is similar to the comparison with CALU (see Figure 17) as this can be attributed partially to the resolution difference. But in case of

grassland it is much higher (about 90% for CALC-2020 (Figure 18) compared to about 50% for CALU.

     Since dwarf and low shrubs are very common over the evaluation region, there is a widespread mismatch (Figure 16).



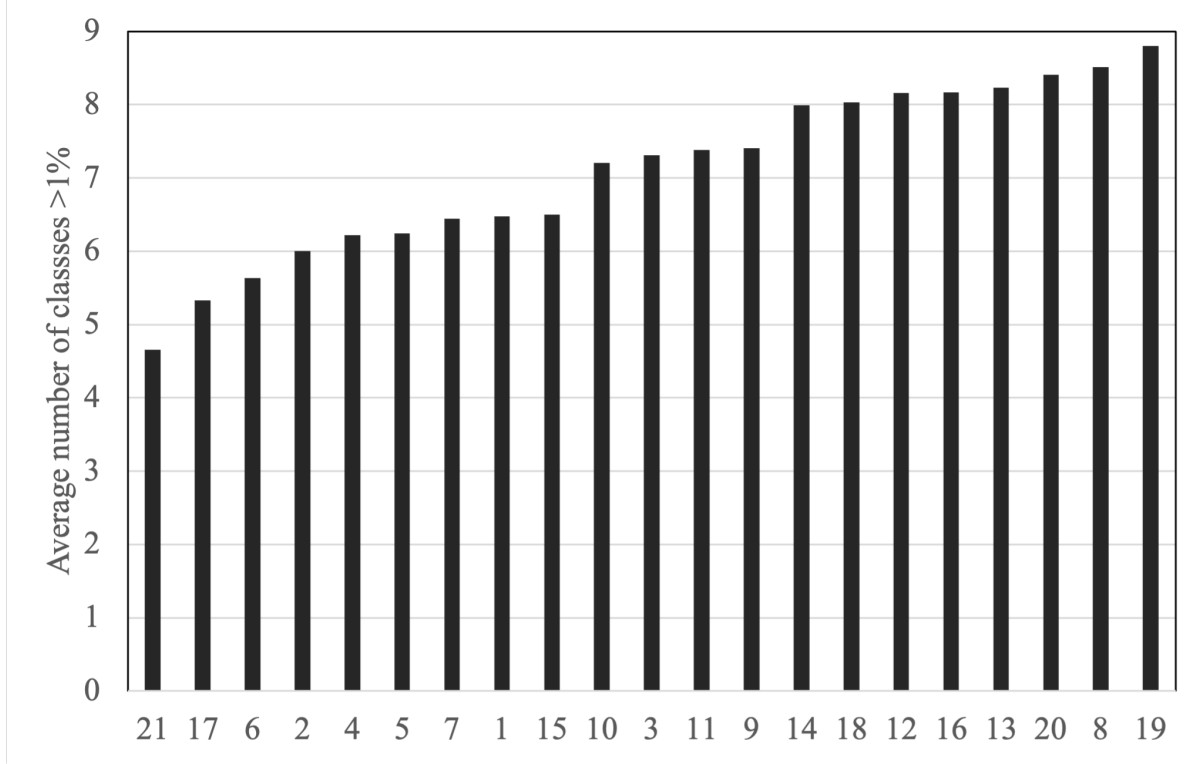

**Figure 12.** Diversity (average number of CAL Units within 1km x 1km cells) versus majority, for unit IDs see Table 2. Sorted by diversity.

## 5   Discussion

The high richness for shrub tundra and wetland complexes (as defined in the CAVM) underlines the importance of spatial resolution and need for thematic content. The largest differences in class assignment between the prototype with 20m and the
new version with 10m occurred also for the shrub tundra groups what reflects the high heterogeneity. Especially wetlands (in addition to lakes from lakes (Matthews et al., 2020)) are of high relevance for permafrost studies as dry and wet gradients determine carbon fluxes, specifically potential release of methane. The three wetland types represent distinct environments with respect to hydrology and nutrient status. The TN density of #3 (permanent) is lower than for #4 (seasonal) (Table A2). Unit #2 represents permanent inundation. In addition, landcover unit composition and spatial patterns may allow for separation
of different wetland types. Especially unit #8 is expected to be very heterogeneous and to include inundated parts. Figure 19 provides examples for different patterns of wetland classes. Drained lake basin composition is distinct from wet sites in areas dominated by peat bogs. The high spatial and thematic detail that CALU provides potentially allows for space for time approaches of impact assessment of permafrost related landcover change in addition to classical vegetation index trend analyses, e.g. Nitze et al. (2018); Foster et al. (2022).



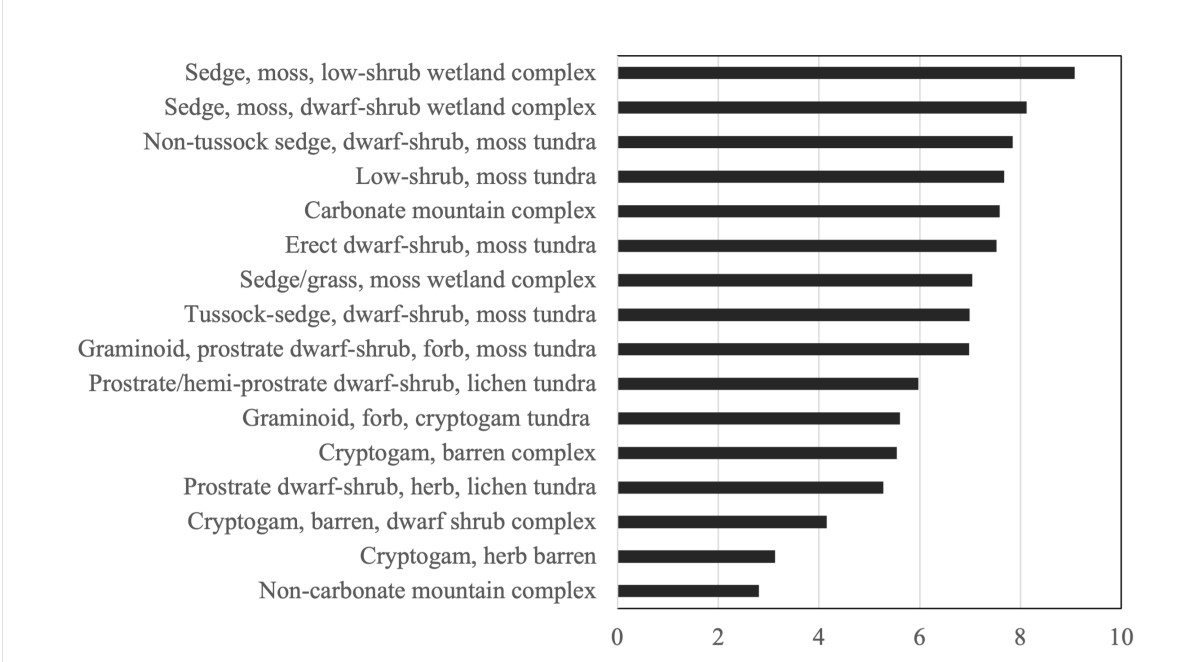

**Figure 13.** Landcover diversity for CAVM (Circum Arctic vegetation map (Raynolds et al., 2019)) classes: average number of CAL Units within 1km x 1km cells.

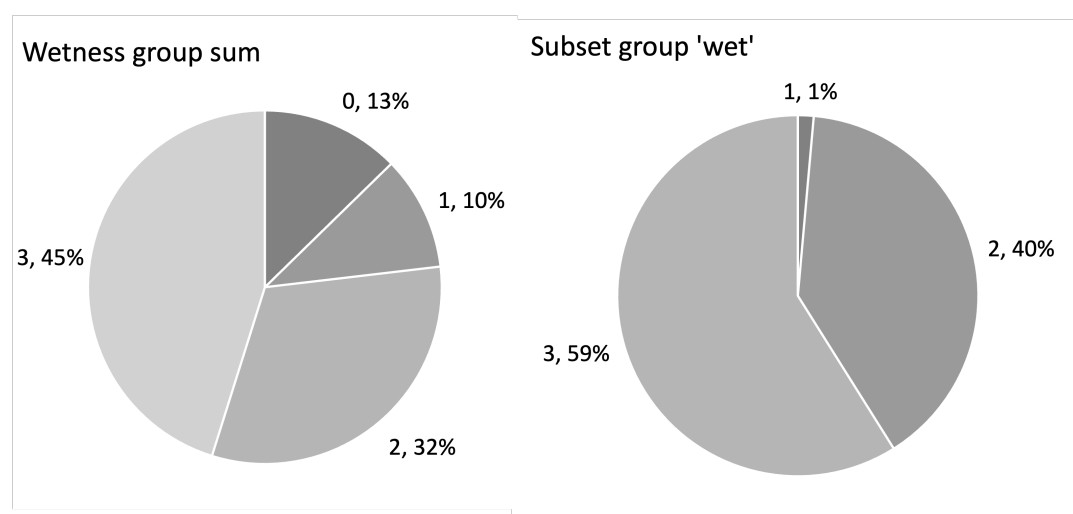

**Figure 14.** Wetness diversity within 1km x 1km areas (sum of groups B: dry, moist, wet; see Table 2; minimum 1% fraction for each type), (left) all CAVM (Circum Arctic vegetation map (Raynolds et al., 2019)) region, (right) spatial subset for group B 'wet' (see Table 2).



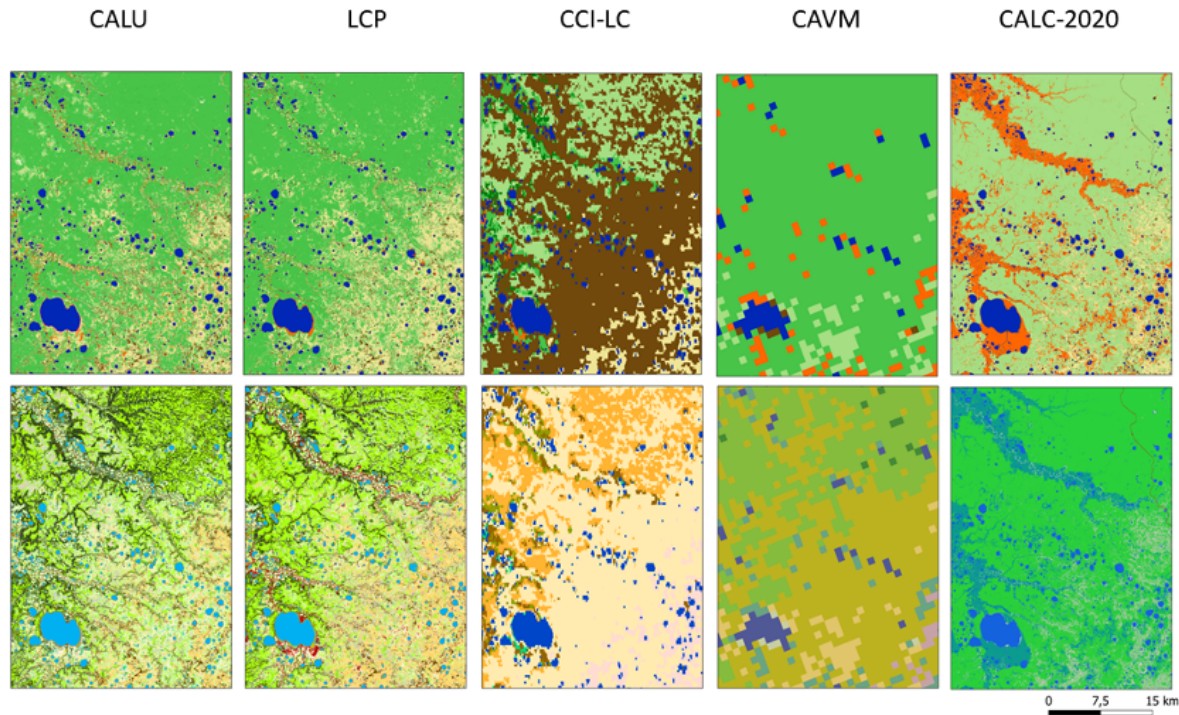

**Figure 15.** Comparison across all evaluated products for a region dominated by shrub tundra with patches of wetland. LCP refers to the prototype. Upper row- grouped classes, lower row – original classes (for detailed legend see Tables 2, C1,C2 and C3 )

The comparison to Landcover CCI confirms previous assessments based Landsat derived landcover maps (Bartsch et al., 2016a). The thematic content does not match tundra specific landscape types. The barren fraction is in general too large. A large proportion is covered by low vegetation. Palmtag et al. (2022) calculated an average of more than 9 kg C m-2 over the top meter for the barren classes, which is much higher than in the CALU assessment (2 kg C m-2) due to the inclusion of areas with organic soils in the Landcover CCI barren class. The agreement for the wetland classes is similar as was found in (Palmtag et al., 2022), with 20.65% (Table 4) and 19% respectively. This has implication for further use as for example in case of wetland map creation such as in Olefeldt et al. (2021). Palmtag et al. (2022), with reference to Hugelius et al. (2020) suggest a four times larger permafrost wetland area. Just within the extent of the CAVM, our estimate is larger with 1.16 Mio km$^2$ than the Landcover CCI fraction for the entire high latitude permafrost domain with 1.01 Mio km$^2$.

The use of terrain information in the retrieval as in the CALC-2020 dataset leads to overestimation of wetland areas in valleys (Figures 15 and 16) and drained lake basins (Figure 19). The heterogeneity in those areas is lost in this case. In total, more area is, however, characterized by wetland in CALU. This might be attributed to the inclusion of shallow water bodies. The classification of areas with shrubs (80% of shrub tundra as defined in CALU) as graminoid tundra leads to differences in dominating class. The most common class of CALU is 'moist tundra, abundant moss, prostrate shrubs' instead.



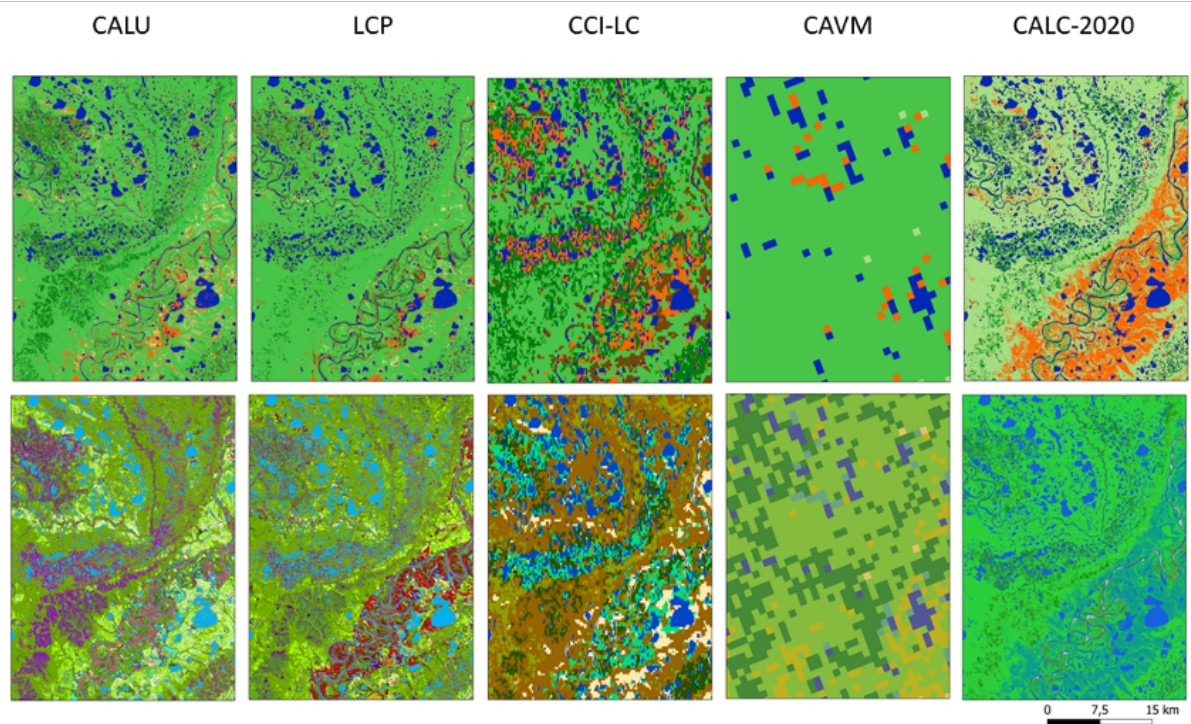

**Figure 16.** Comparison across all evaluated products for a region dominated by shrub tundra with patches of forest and wetland. LCP refers to the prototype. Upper row- grouped classes, lower row – original classes (for detailed legend see Tables 2, 2, C1,C2 and C3).

The high number of classes requires cloud free Sentinel-2 at the peak vegetation season. Several years are therefore needed
to assemble the circumpolar mosaic. Ingested images span a wide range of acquisition years which may limit application of landcover change studies. Such analyses should be carried out only locally and the documentation of acquisition dates consulted. In addition, years can differ regarding landsurface hydrology and vegetation state. Wet years with flooding and dry years leading to drought are expected to impact the detectability of specifically the wetland classes. Haze caused by forest fires in general cannot be corrected with the used tools. Affected scenes can not be used. This reduces the number of usable scenes
considerably, as fires are abundant and smoke is transported to the tundra area.

Although that a very high number of in situ data was available (almost 40 times more vegetation sites than used for the CALC-2020 assessment (Liu et al., 2023)), more is needed for full characterization of Arctic landscapes. There is a lack of soil data (organic layer thickness) for barren areas, and unit #13 (dense shrubs) as it is less common and more difficult to access. The latter also applies to the shallow water wetland class as it is aquatic and thus usually not part of terrestrial vegetation
surveys. This class might be, however, important for upscaling of methane fluxes. Soil data availability largely represents unit abundance. The most soil data are available for the second most common unit (#11), but much less for the most abundant class (#6) what is most likely due to the comparably low organic carbon content and occurrence in high Arctic regions. Soil sampling usually targets soil carbon quantification. Shrub tundra types are in general well sampled (more than 50% of records).

 

**Figure 17.** Proportion of grouped Landcover CCI, CAVM, CALC-2020 and prototype classes within grouped CAL Units. For grouping schemes see Tables 2 and C3.

Issues with wrong assignment are common for lakes in the northern parts. Late floating ice fragments are identified ice/snow
instead of open water. Lakes may also be missing in regions close to the coastline where the Copernicus DEM contained zero
or no data. Tree species typical for forest occur with reduced size, crown diameter and sparse distribution in the transition zone.
Several units which include low shrubs may therefore also include trees.

The current CALU version and presented analyses does not distinguish between natural and artificial barren areas, but a
dataset which can be used for separation is available Bartsch et al. (2021a, b). It was derived from Sentinel-1/2 at 10m nominal
resolution. The fraction of artificial areas over the analyses region is, however, comparably small, < 0.02 % versus an overall
barren fraction of 14.5% (Table 2).





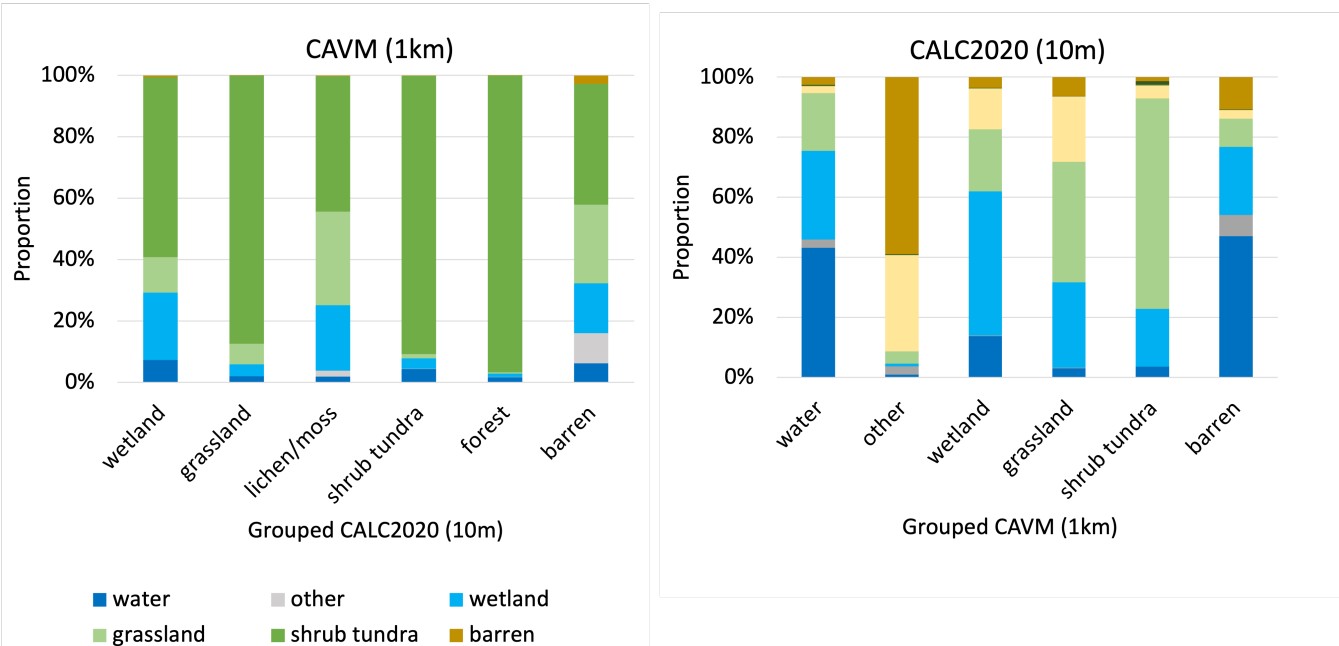

**Figure 18.** Comparison for landcover groups – CAVM versus CALC 2020. For grouping schemes see Tables C2 and C3.

## 6 Conclusions

Tundra regions with wetlands are characterized by high landcover diversity, in addition to the high number of small lakes. Wet, moist and dry landscape types co-occur close to each other in most of the area north of the treeline. Pure wetland coverage over

430 an extent of 1x1km are rare. This is expected to be of relevance for any type of analyses related to carbon fluxes, including up-scaling, inversions or landsurface modelling.

Permafrost degradation features are known to lead to changes in landcover type and specifically wetness. For example drained lake basins and also fire scars represent fast changing environments in the years after events. The new circumpolar landcover unit map provides three types of wetland classes in addition to 14 terrestrial tundra units covering barren area as well

as different types of shrub physiognomy and soil moisture gradients. This level of detail is unprecedented. Both, vegetation and soil type variations can be documented with extensive in situ datasets. Some landcover units with less extensive coverage, specifically disturbed sites (fires) are under-represented in available records.

The high thematic content requires good quality of input data. Although data availability from Sentinel-1 and -2 is good since 2016, the seven years of records have not been sufficient for complete coverage (1%). Cloud coverage in case of Sentinel-2

and inconsistent acquisition strategies for Sentinel-1 lead to quality issues. Several years of Sentinel-2 have been combined for robustness where sufficient acquisitions have been available. This needs to be considered when combined with other records specifically with respect to impact of assessment of cryosphere change impacts on landsurface hydrology.



**Figure 19.** Examples for different wetland types: left - drained lake basins on central Yamal in the tundra zone, right - peat bogs in the hypo-Arctic tundra zone of Western Siberia. top: CALU - for legend see Table 2, bottom: CALC-2020 ((Liu et al., 2023), blue - open water, turquoise - wetland, grey - lichen/moss, light green - graminoid tundra).

*Data availability.* The CALU dataset will be openly available following under the following URL: https://zenodo.org/record/ 8399018; DOI: 10.5281/zenodo.8399018



*Author contributions.* AB developed the concept for the study, analysed the results and wrote the first draft of the manuscript. KE and GH contributed to the in situ surveys, their compilation and analyses and to the writing of the manuscript. AE, BW, XM, CB processed the satellite data and contributed to in situ and other reference data statistical analyses. HB supported the postprocessing and analyses of results. BH and ML supported the interpretation and documentation of the results as well as writing of the manuscript.

*Competing interests.* The authors declare no competing interests.

*Acknowledgements.* This work was supported by the European Space Agency CCI+ Permafrost and AMPAC-Net projects, the European Research Council project No. 951288 (Q-Arctic), and has received funding under the European Union's Horizon 2020 Research and Innovation Programme under Grant Agreement No. 869471 (CHARTER).

We acknowledge all AVA data collectors for the western Siberia region, specifically Olga Khitun and Elena Troeva for photographs. We further acknowledge Mareike Wieczorek for provision of photographs from the Kolyma region.



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



# Appendix A:  Class statistics

**Table A1.** Soil surface organic horizon depth statistics

| Unit | Samples | Mean | Median | Std |
|------|---------|------|--------|-----|
| 2    | 23      | 105  | 62     | 103 |
| 15   | 25      | 93   | 72     | 99  |
| 12   | 42      | 74   | 15     | 112 |
| 14   | 51      | 74   | 20     | 98  |
| 16   | 21      | 64   | 8      | 109 |
| 6    | 44      | 62   | 33     | 78  |
| 8    | 43      | 59   | 30     | 65  |
| 11   | 157     | 41   | 12     | 74  |
| 17   | 5       | 40   | 18     | 36  |
| 3    | 54      | 40   | 20     | 54  |
| 4    | 64      | 39   | 20     | 39  |
| 7    | 30      | 37   | 19     | 55  |
| 9    | 123     | 34   | 18     | 42  |
| 10   | 74      | 29   | 15     | 37  |
| 5    | 19      | 28   | 20     | 20  |
| 13   | 10      | 23   | 11     | 33  |
| 21   | 3       | 13   | 0      | 19  |





**Table A2.** Average soil characteristics based on Palmtag et al. (2022)

| Type | Dry bulk density (g/cm3) | Wet bulk density (g/cm3) | Organic volumetric content | Water volumetric content | Mineral volumetric content | SOC density (kg C m-2) | TN density (kg N m-2) | Samples |
|---|---|---|---|---|---|---|---|---|
| 3 | 0.61 | 1.11 | 4.53 | 50.38 | 20.61 | 29.41 | 1.47 | 22 |
| 4 | 0.92 | 1.48 | 3.97 | 46.87 | 27.34 | 30.97 | 2.07 | 6 |
| 5 | 1.27 | 1.62 | 3.36 | 33.03 | 46.42 | 22.19 | 1.61 | 5 |
| 6 | 1.18 | 1.53 | 3.03 | 34.64 | 44.02 | 21.19 | 1.04 | 31 |
| 7 | 0.86 | 1.24 | 4.04 | 40.51 | 30.42 | 26.64 | 1.16 | 4 |
| 8 | 0.86 | 1.38 | 3.72 | 44.70 | 27.35 | 26.77 | 1.49 | 20 |
| 9 | 0.85 | 1.35 | 4.60 | 50.25 | 29.48 | 30.79 | 1.43 | 56 |
| 10 | 0.74 | 1.27 | 5.90 | 52.82 | 24.60 | 38.20 | 1.63 | 27 |
| 11 | 0.75 | 1.28 | 5.54 | 50.55 | 25.06 | 36.90 | 1.63 | 52 |
| 12 | 0.63 | 1.06 | 3.56 | 43.48 | 22.82 | 23.05 | 1.28 | 8 |
| 13 | 0.74 | 1.13 | 3.69 | 38.62 | 26.08 | 23.83 | 1.58 | 1 |
| 14 | 0.86 | 1.34 | 4.69 | 47.33 | 30.43 | 30.31 | 1.56 | 16 |
| 15 | 1.27 | 1.62 | 2.26 | 35.84 | 47.31 | 15.14 | 1.03 | 7 |
| 16 | 1.30 | 1.73 | 3.50 | 39.72 | 44.99 | 22.77 | 1.91 | 6 |
| 19 | 1.47 | 1.90 | 2.13 | 38.01 | 58.59 | 14.69 | 0.65 | 3 |
| 20 | 0.76 | 1.17 | 4.12 | 41.00 | 27.35 | 26.89 | 1.55 | 2 |
| 21 | 1.83 | 2.03 | 0.30 | 16.77 | 68.82 | 2.98 | 0.39 | 11 |



# Appendix B: Class descriptions

## B1 Class 1

**Description:** Open water

## B2 Class 2

**Description:** Shallow water/abundant macrophytes

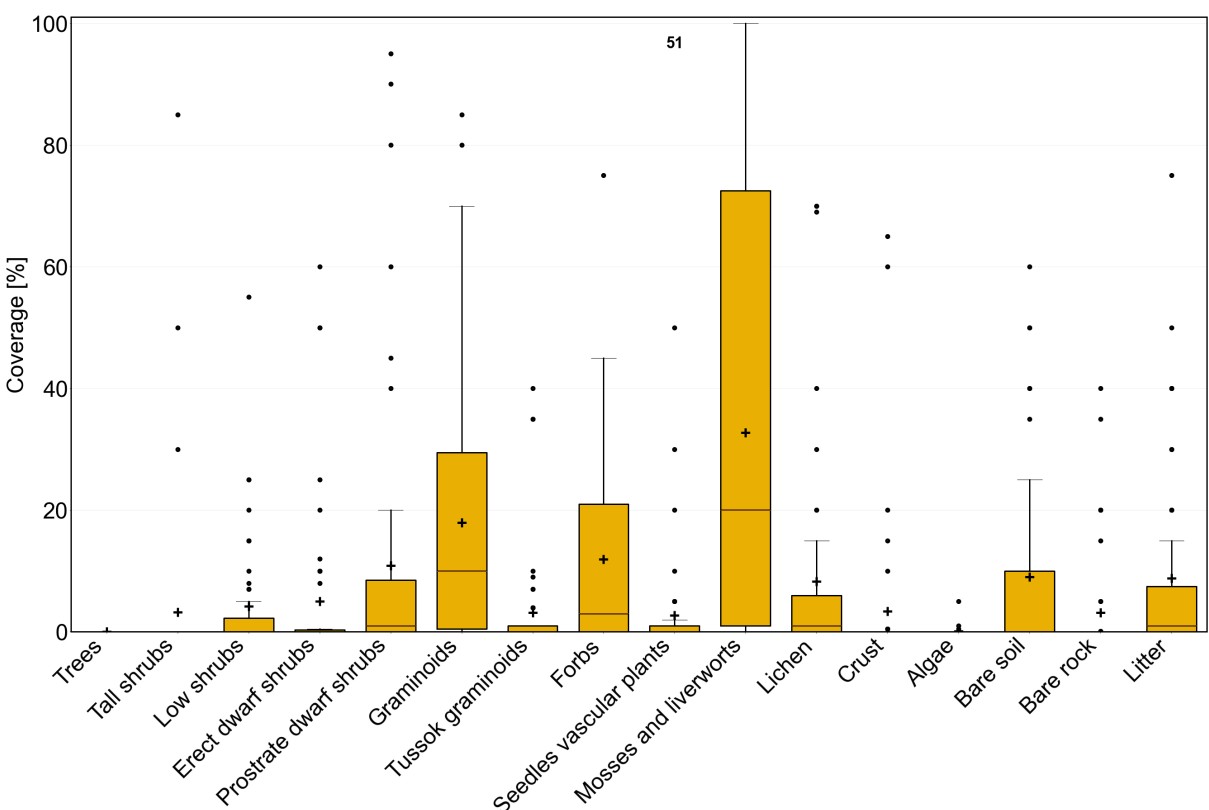

**Figure B1.** Class 2 vegetation properties based on AVA (Zemlianskii et al., 2023).





**Figure B2.** Example photograph class 2, lake margin with patches of macrophytes, water depth approximately 22 cm (Annett Bartsch, 21.07.2023, Inuvik region)



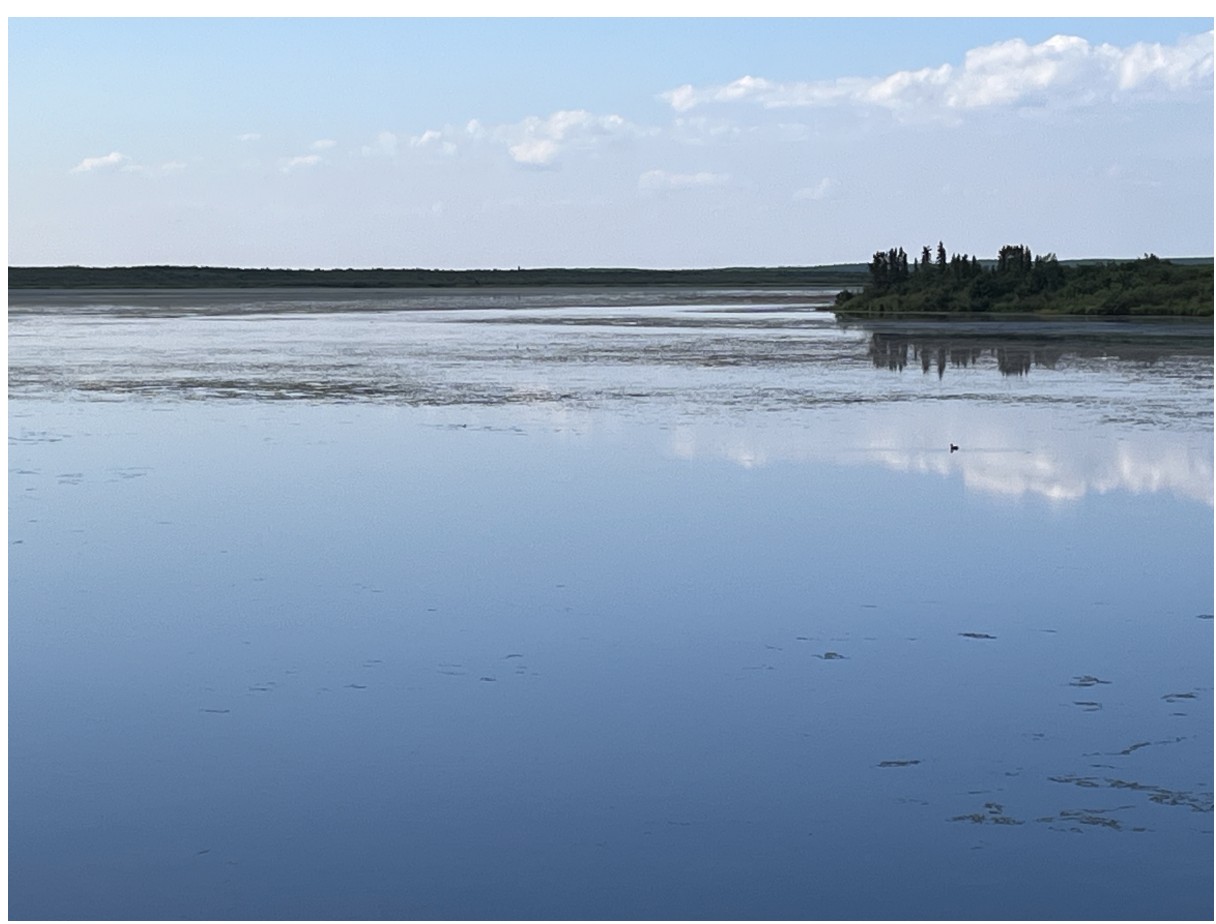

**Figure B3.** Example photograph class 2, lake with patches of macrophytes in the centre (Annett Bartsch, 2023, Inuvik region)



**B3    Class 3**

**Description:** permanent wetland, aquatic, low to medium organic layer thickness, medium mineral volumetric content.

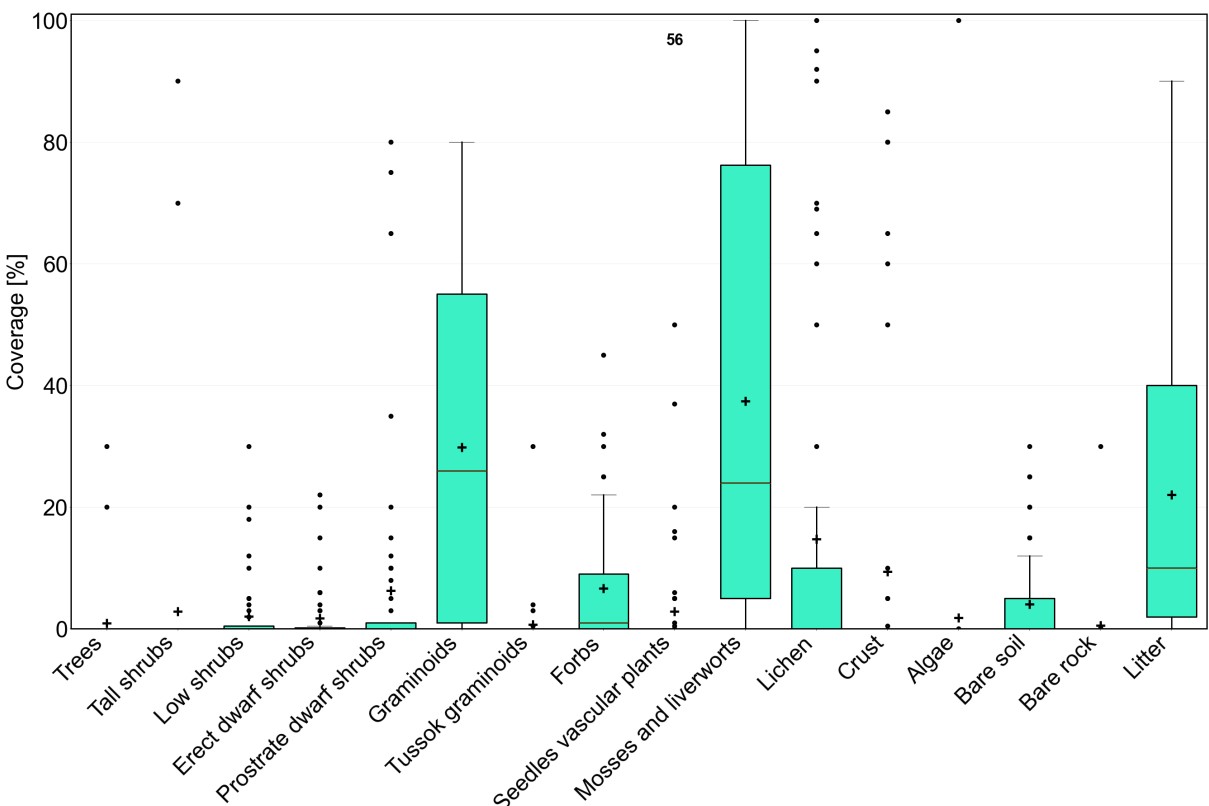

**Figure B4.** Class 3 vegetation properties based on AVA (Zemlianskii et al., 2023).



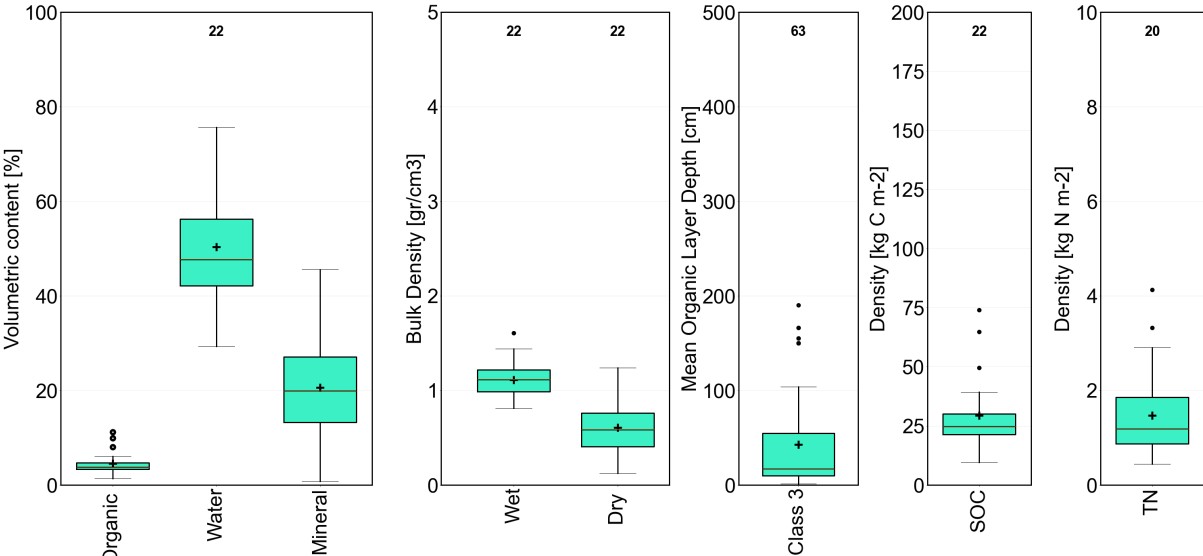

**Figure B5.** Class 3 soil properties based on Palmtag et al. (2022).



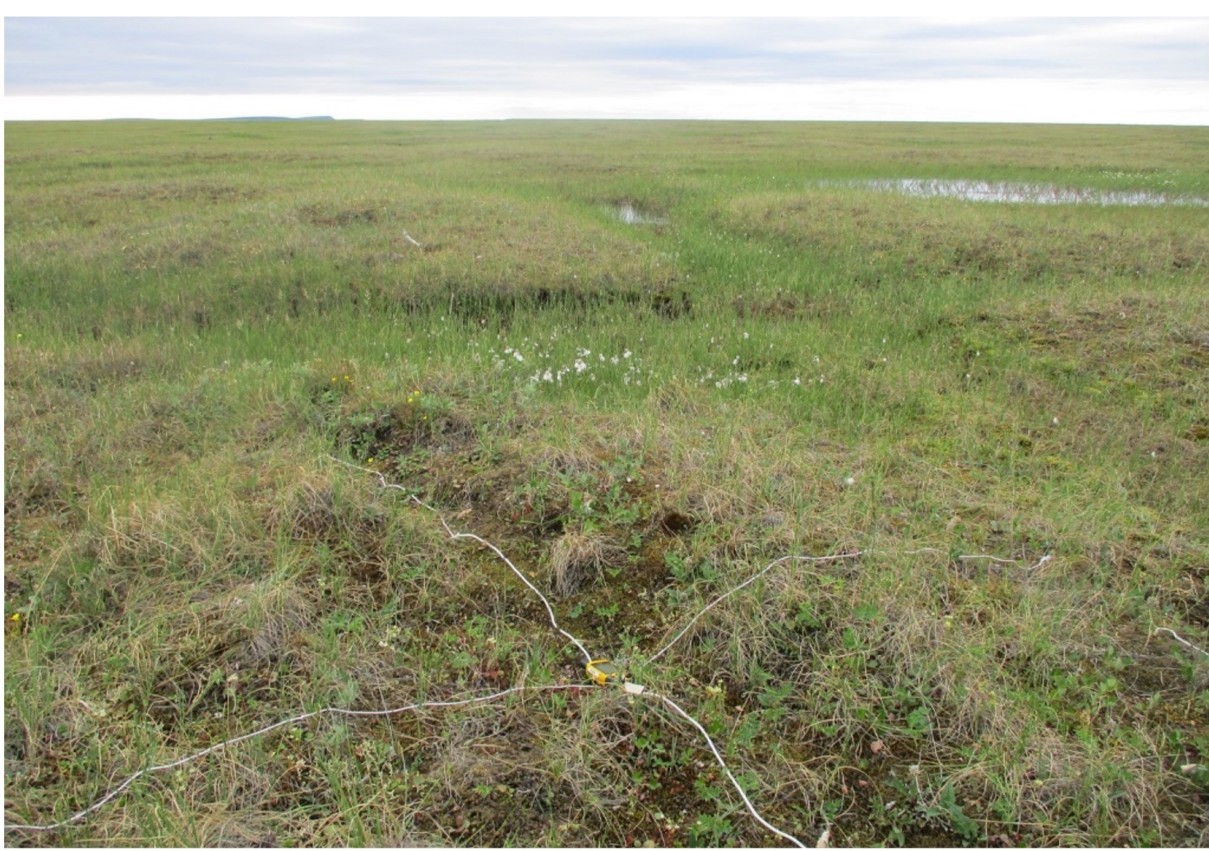

**Figure B6.** Example photograph class 3 (Birgit Heim, Lena Delta, Samoylov polygonal tundra)



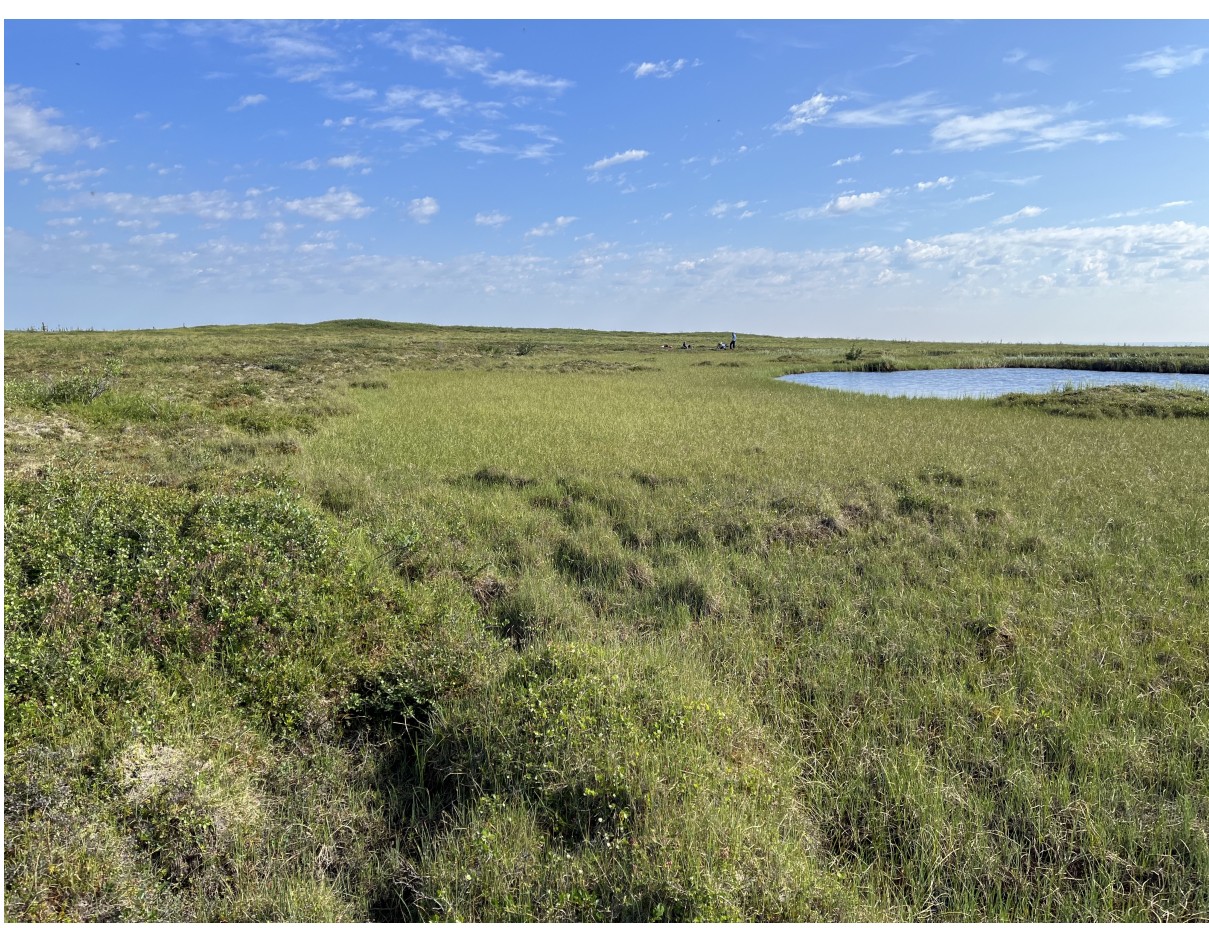

**Figure B7.** Example photograph class 3 (Annett Bartsch, 2023, Inuvik region)



## B4 Class 4

**Description:** wet to aquatic (seasonal wetland), abundant moss, low to medium organic layer thickness, low mineral volumetric content.

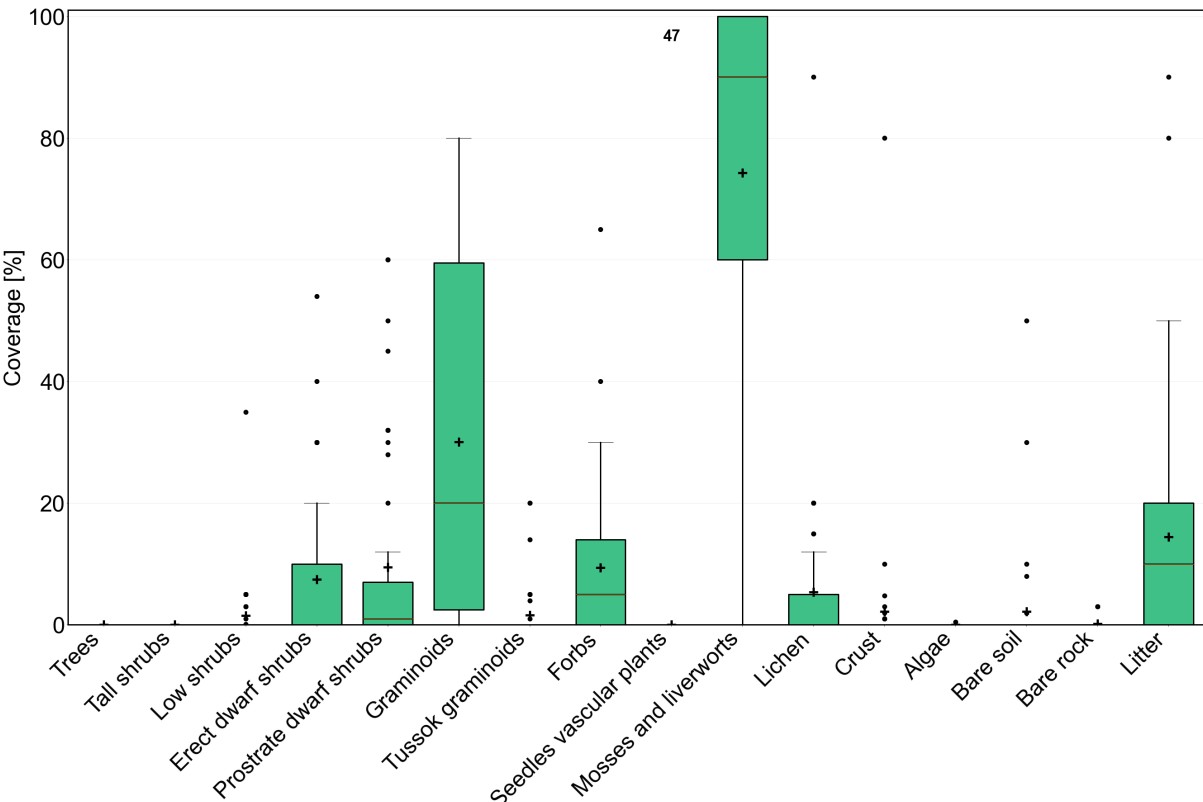

**Figure B8.** Class 4 vegetation properties based on AVA (Zemlianskii et al., 2023).



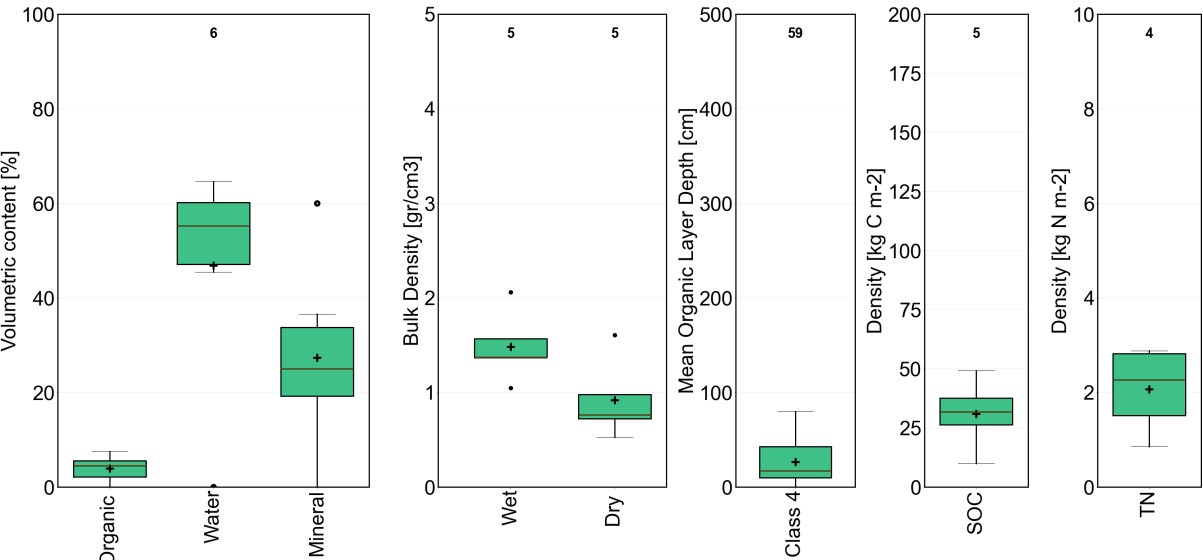

**Figure B9.** Class 4 soil properties based on Palmtag et al. (2022).



**B5   Class 5**

**Description:** moist to wet tundra, abundant moss, prostrate shrubs; low to medium organic layer thickness, medium mineral volumetric content.

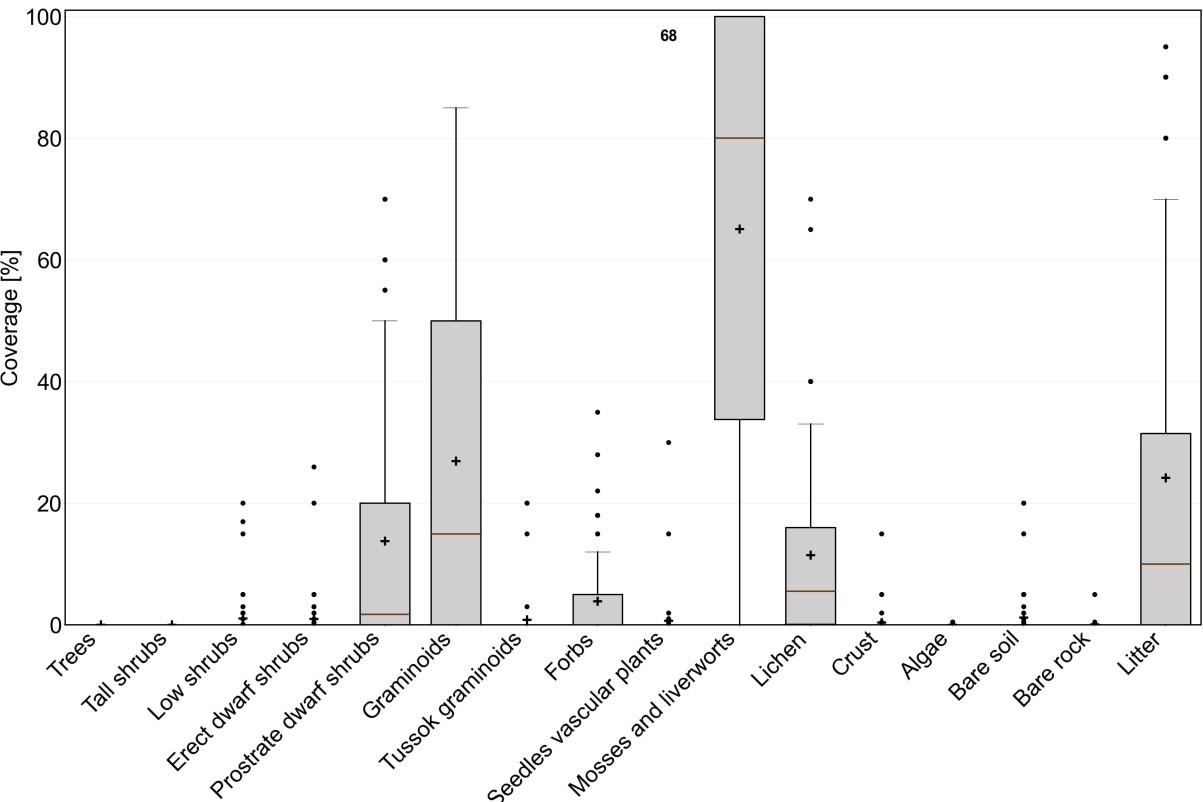

**Figure B10.** Class 5 vegetation properties based on AVA (Zemlianskii et al., 2023).



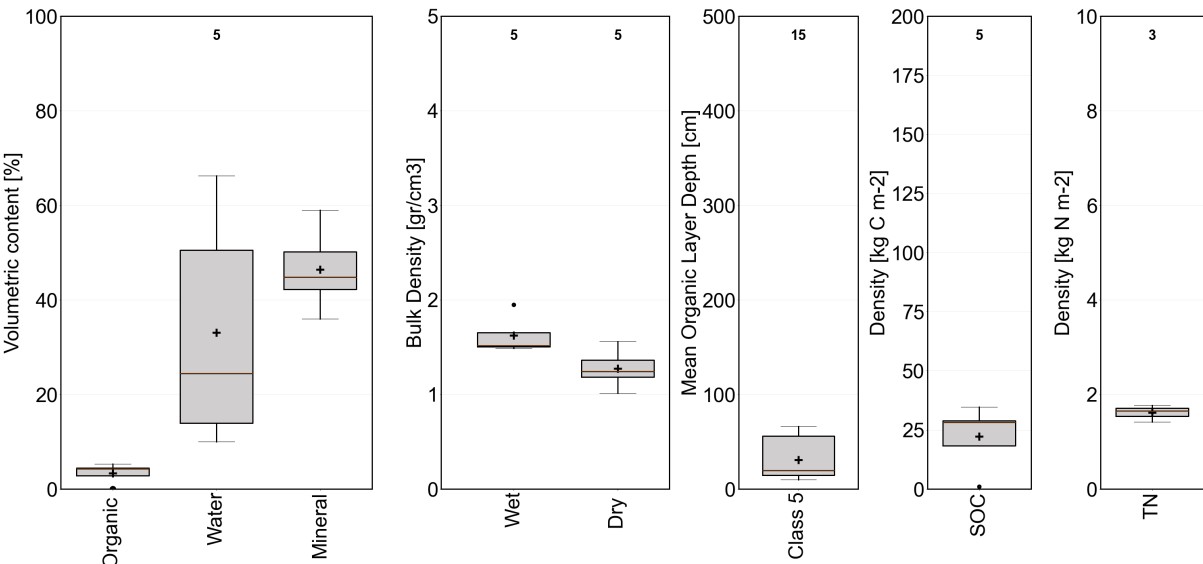

**Figure B11.** Class 5 soil properties based on Palmtag et al. (2022).



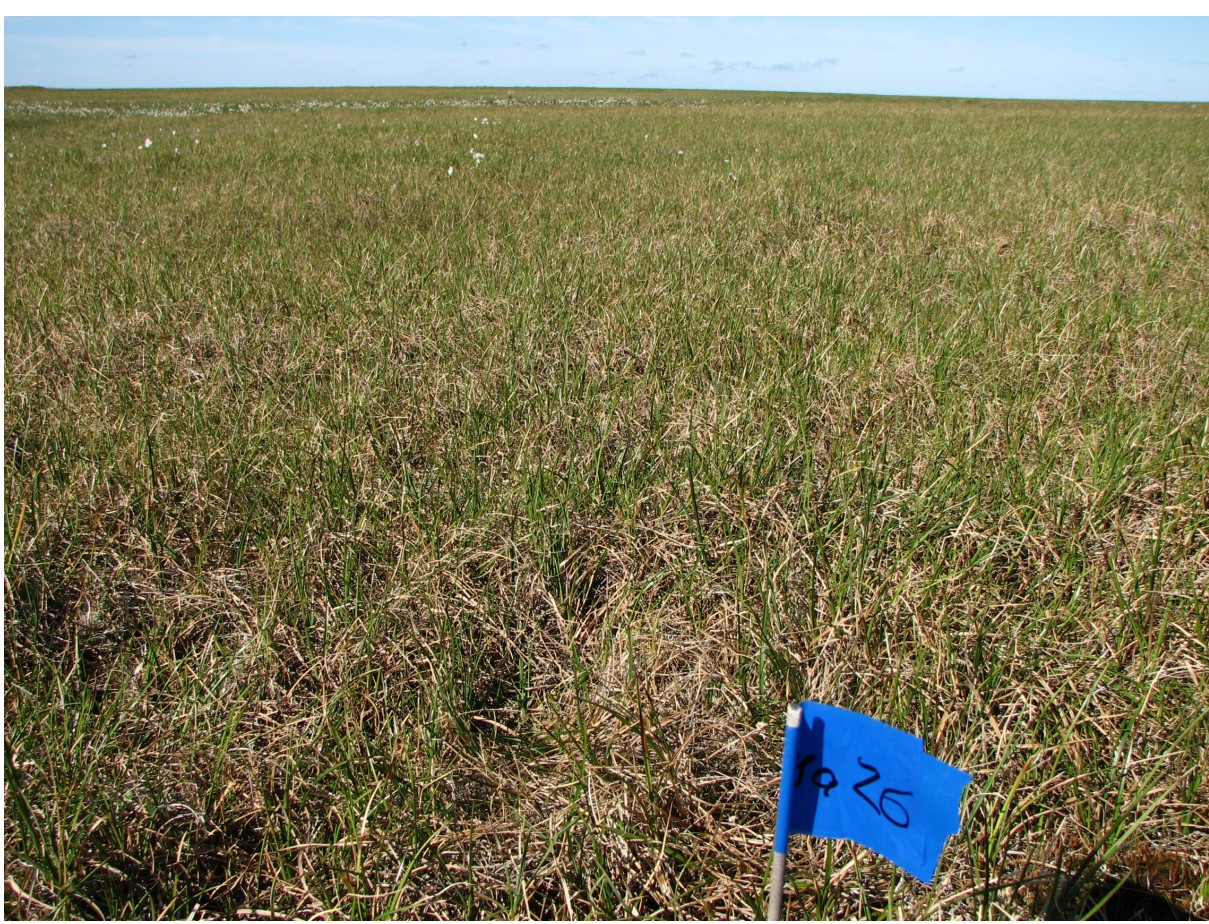

**Figure B12.** Example photograph class 5 (Elena Troeva, 2017, Northern Yamal)



## B6    Class 6

**Description:** dry to moist tundra, partially barren, prostrate shrubs; medium organic layer thickness, high mineral volumetric content.

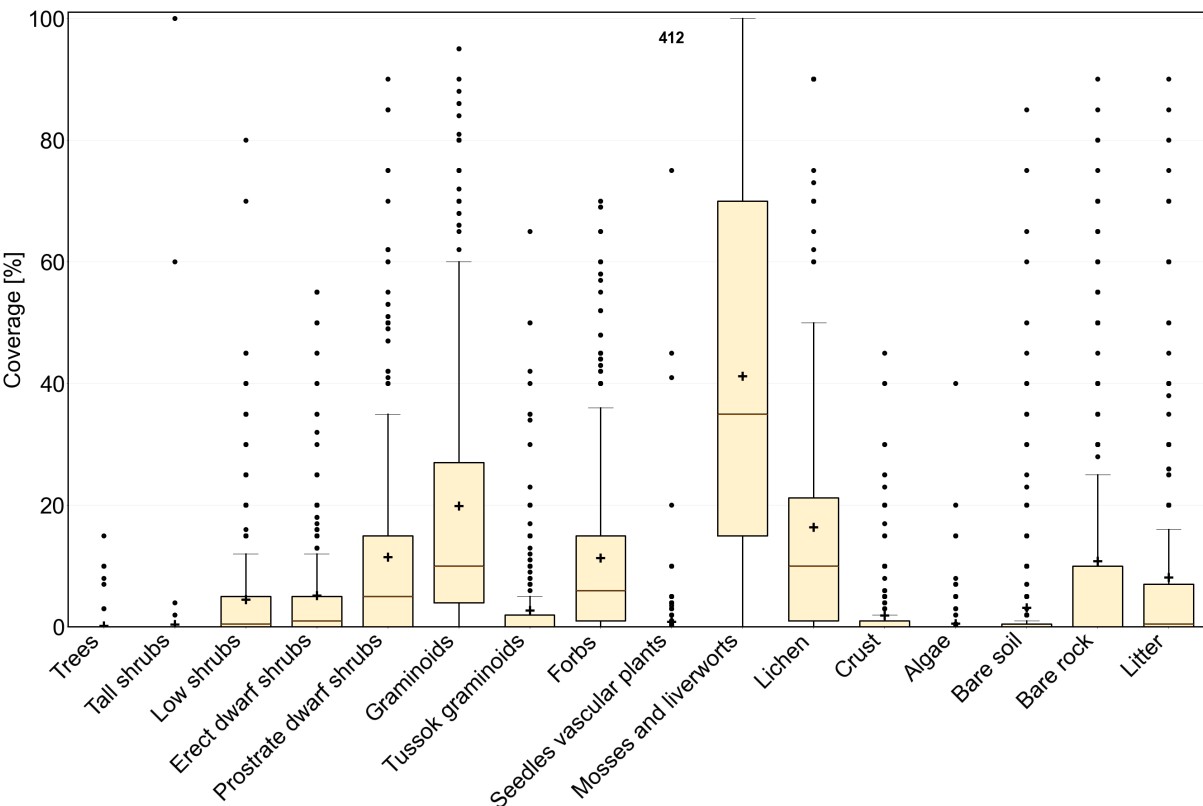

**Figure B13.** Class 6 vegetation properties based on AVA (Zemlianskii et al., 2023).



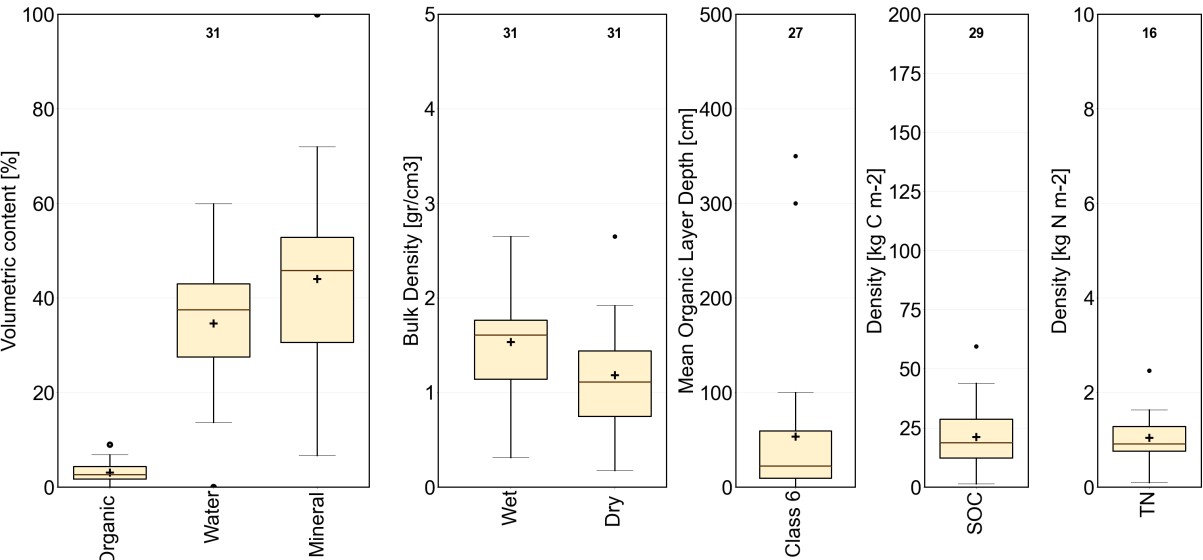

**Figure B14.** Class 6 soil properties based on Palmtag et al. (2022).



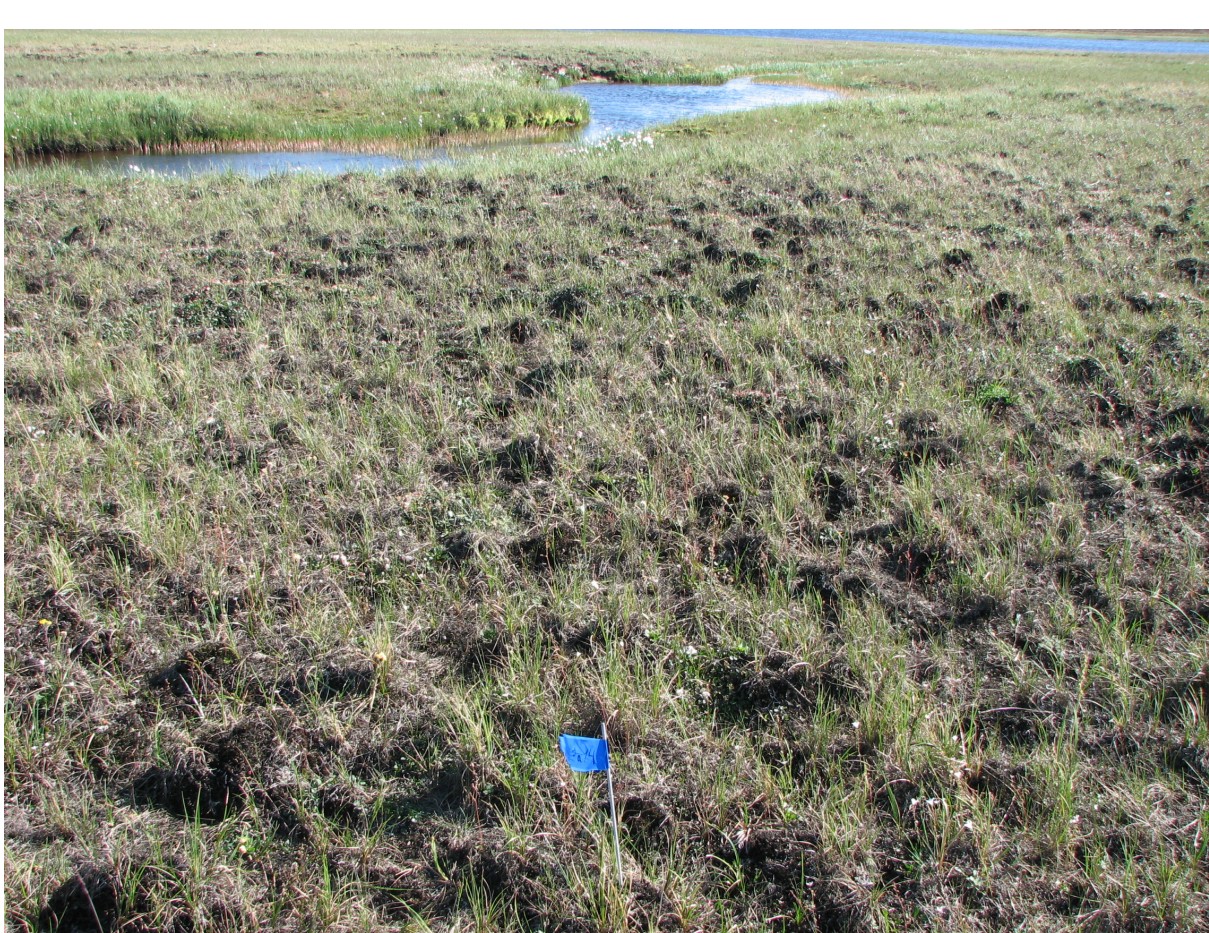

**Figure B15.** Example photograph class 6, foreground (Elena Troeva, 2017, Northern Yamal)



## B7    Class 7

**Description:** dry tundra, abundant lichen, prostrate shrubs; low to medium organic layer thickness, high mineral volumetric content.

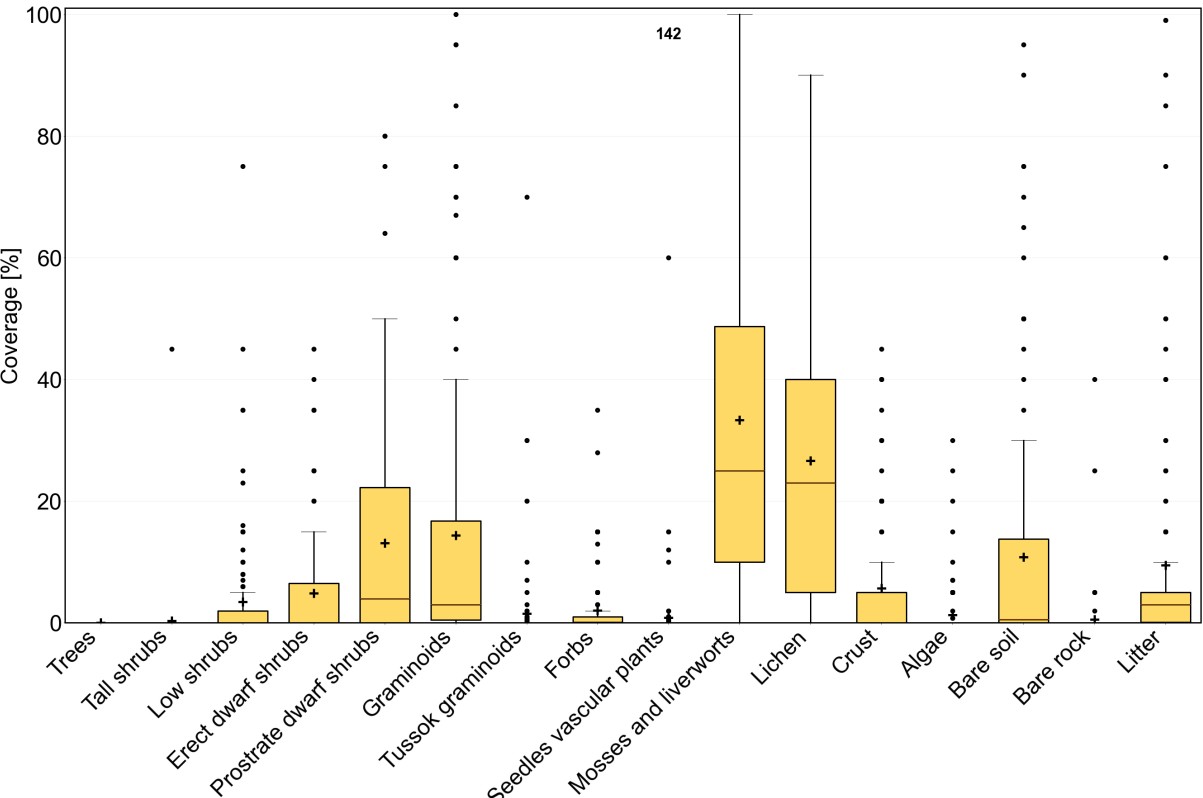

**Figure B16.** Class 7 vegetation properties based on AVA (Zemlianskii et al., 2023).




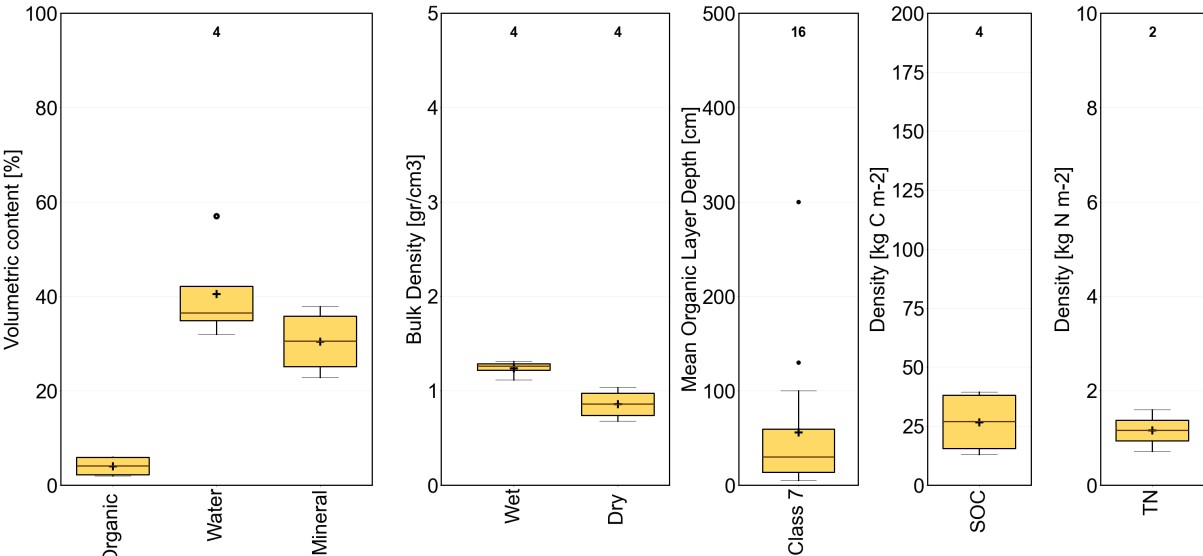

**Figure B17.** Class 7 soil properties based on Palmtag et al. (2022).

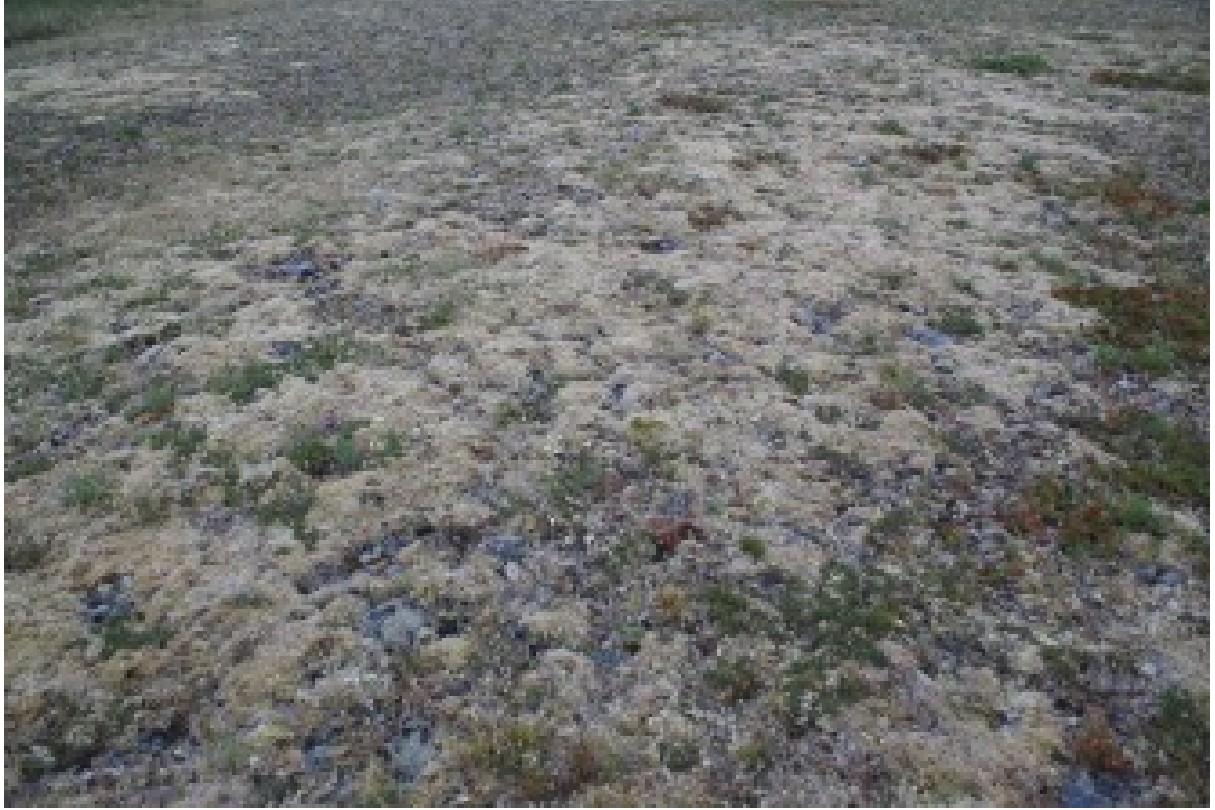

**Figure B18.** Example photograph class 7 (Annett Bartsch 2018, Polar Ural)



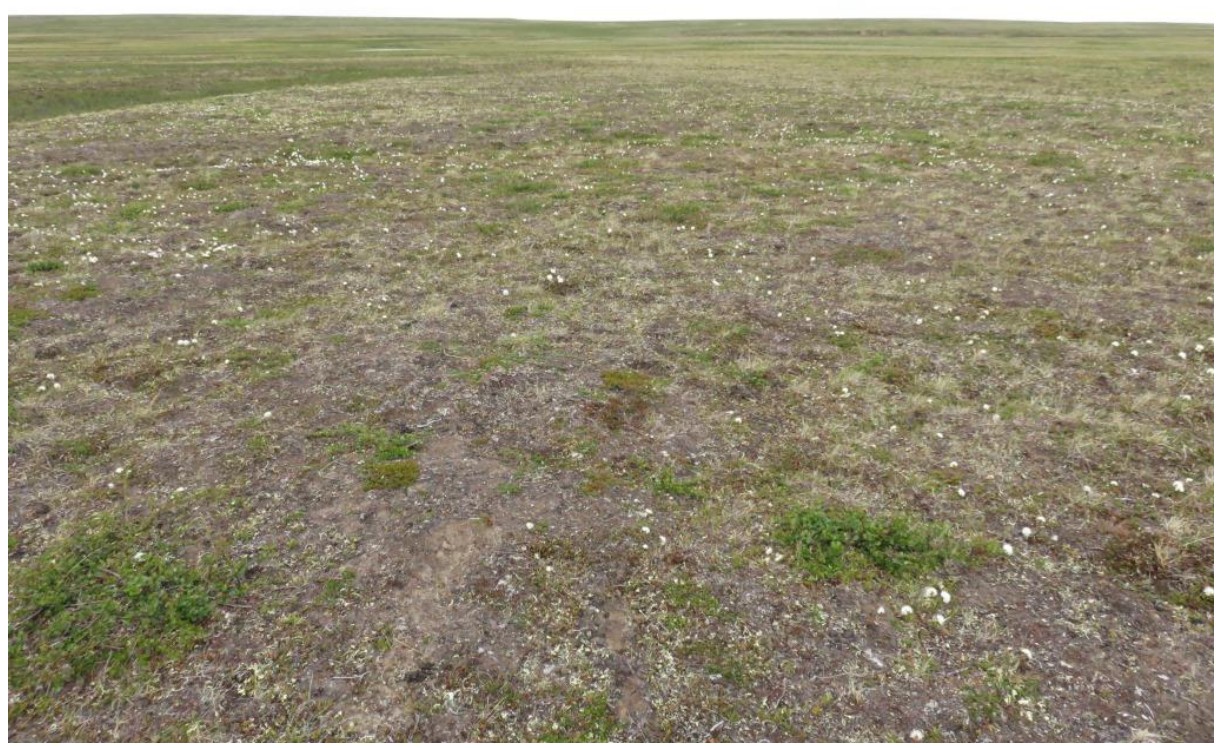

**Figure B19.** Example photograph class 7 (Olga Khitun, 2017, Central Gydan)




## B8 Class 8

**Description:** dry to aquatic tundra, dwarf shrubs (sparse tree cover along treeline, woodlands and open stands); medium organic layer thickness, medium mineral volumetric content.

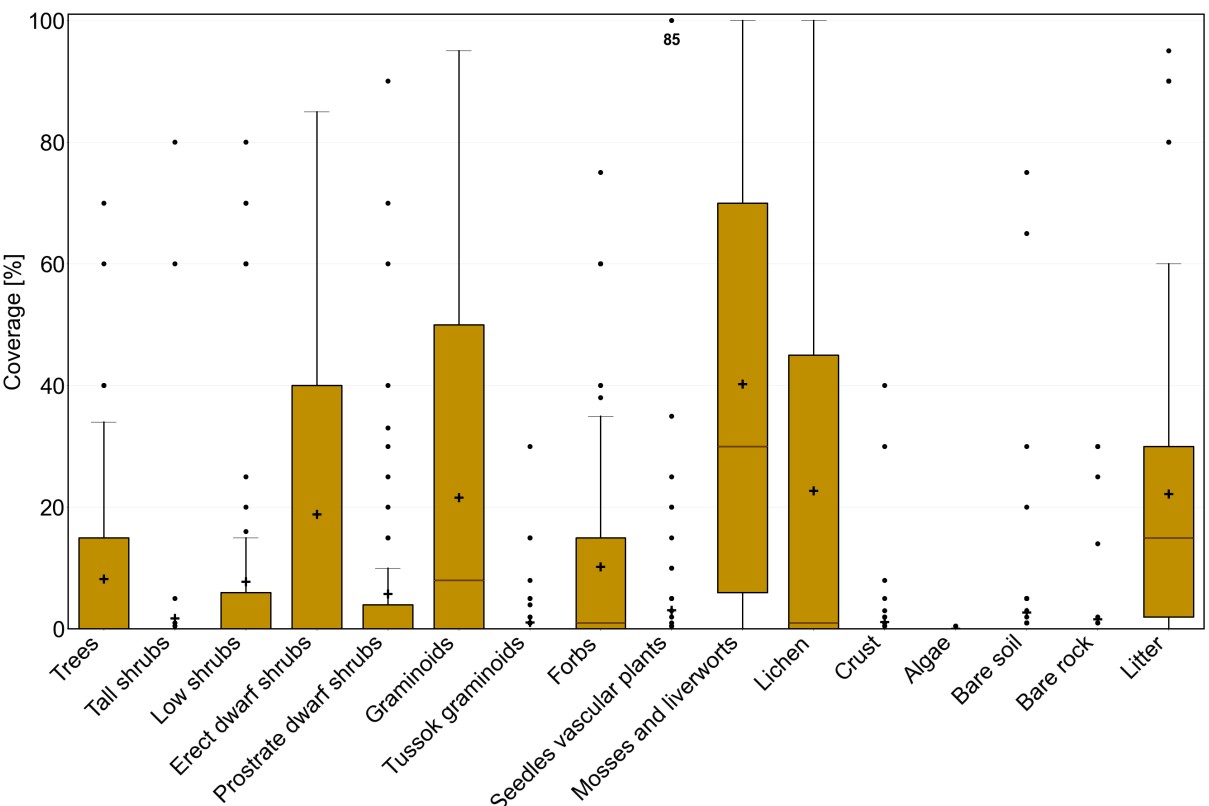

**Figure B20.** Class 8 vegetation properties based on AVA (Zemlianskii et al., 2023).





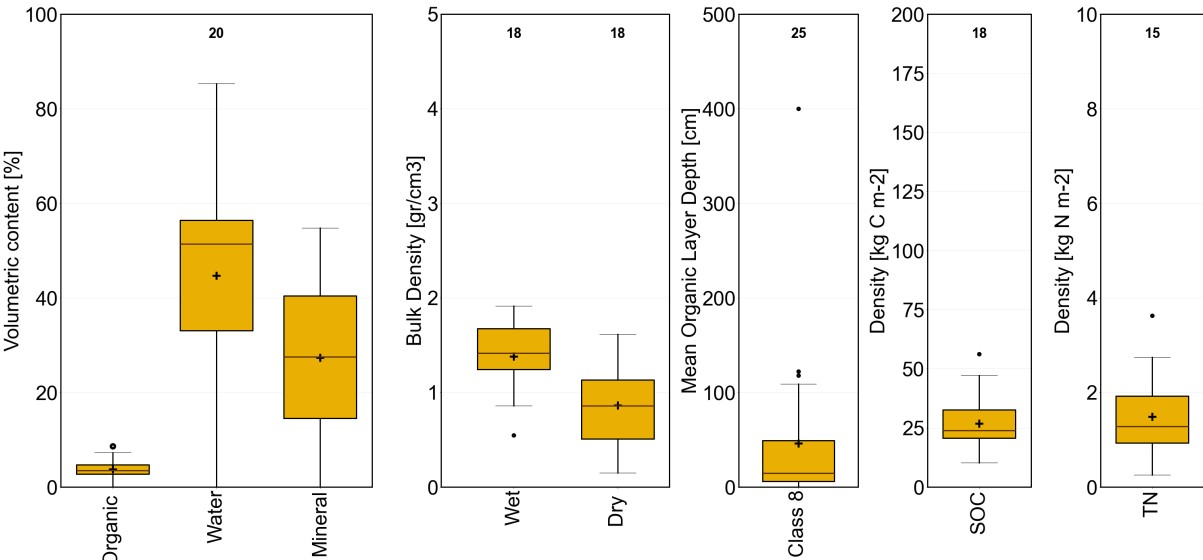

**Figure B21.** Class 8 soil properties based on Palmtag et al. (2022).



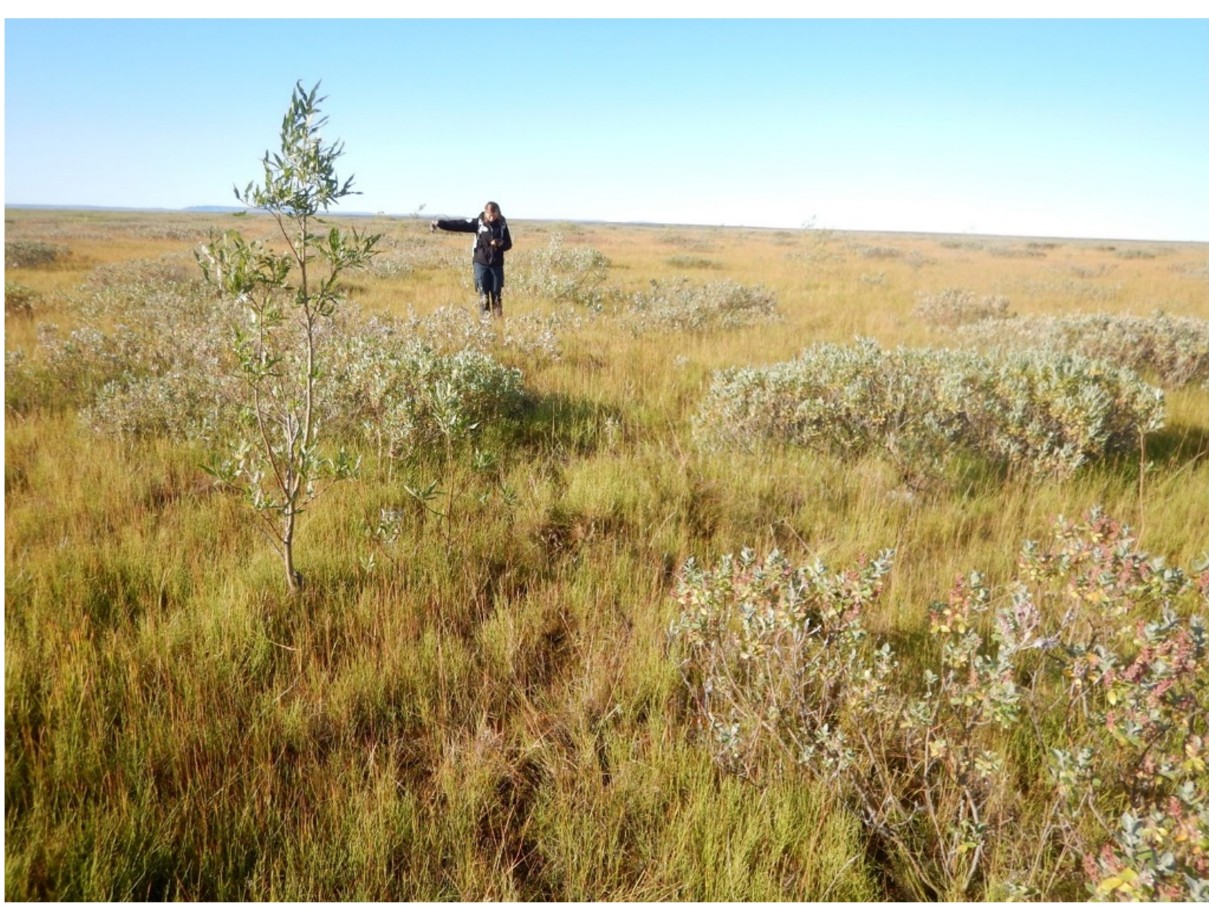

**Figure B22.** Example photograph class 8 (Birgit Heim, Lena delta)



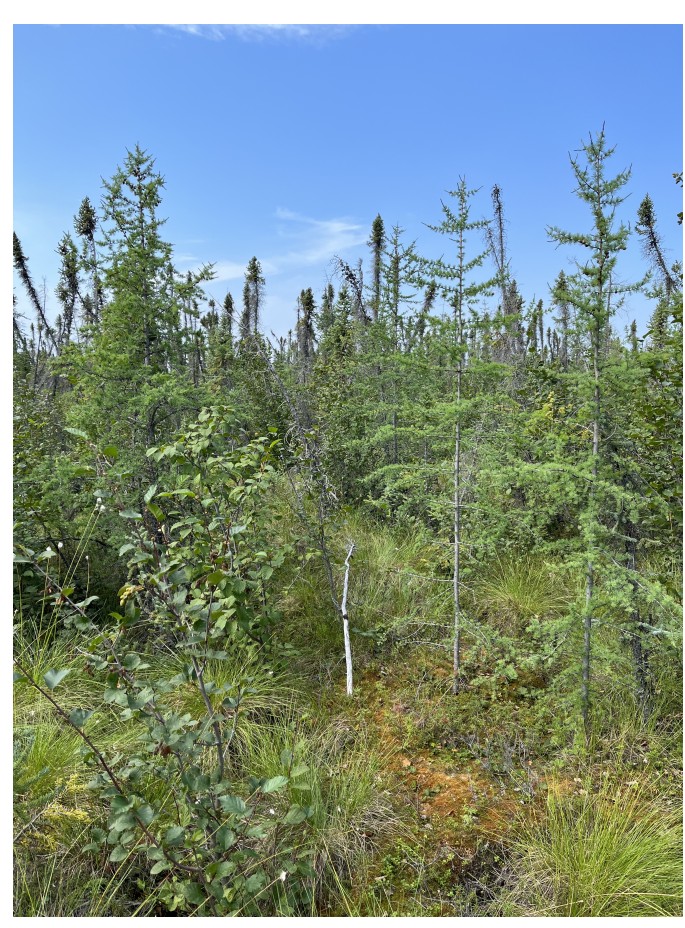

**Figure B23.** Example photograph class 8 (Annett Bartsch, 2023, Inuvik region)





## B9 Class 9

**Description:** dry to moist tundra, prostrate to low shrubs, tussocks; low to medium organic layer thickness, medium mineral volumetric content.

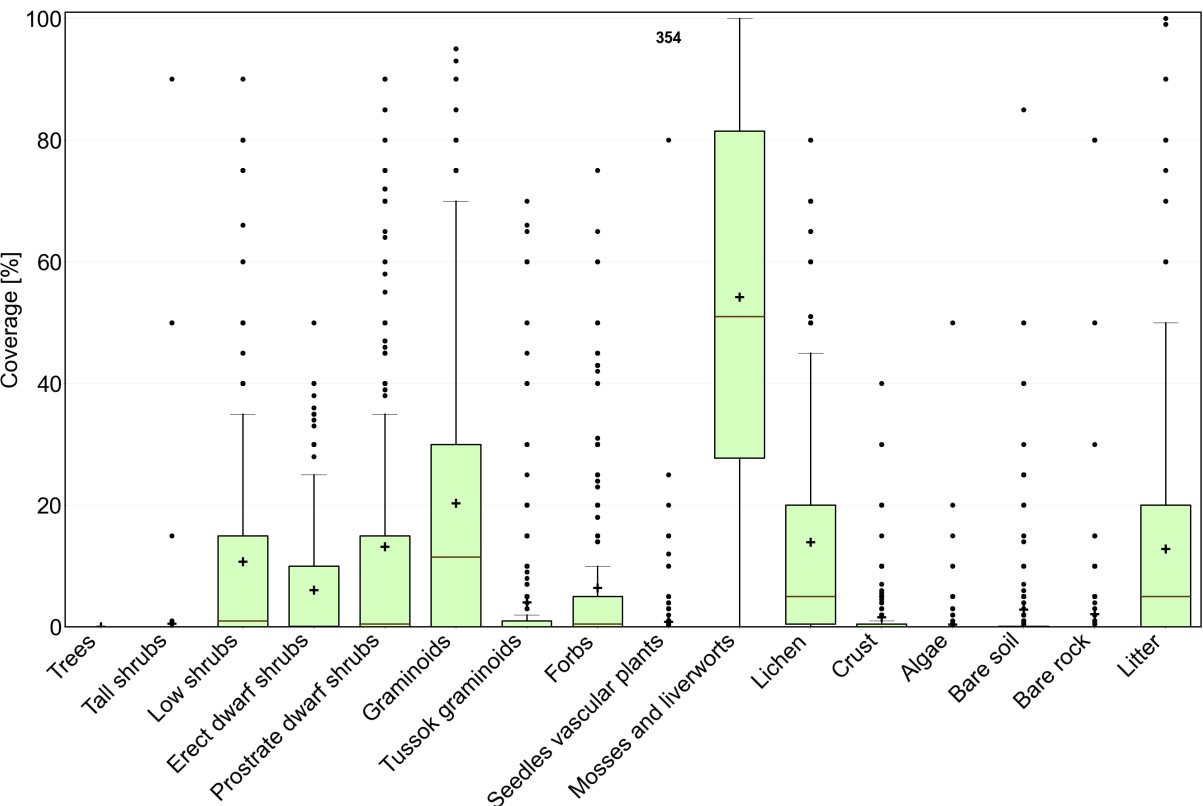

**Figure B24.** Class 9 vegetation properties based on AVA (Zemlianskii et al., 2023).





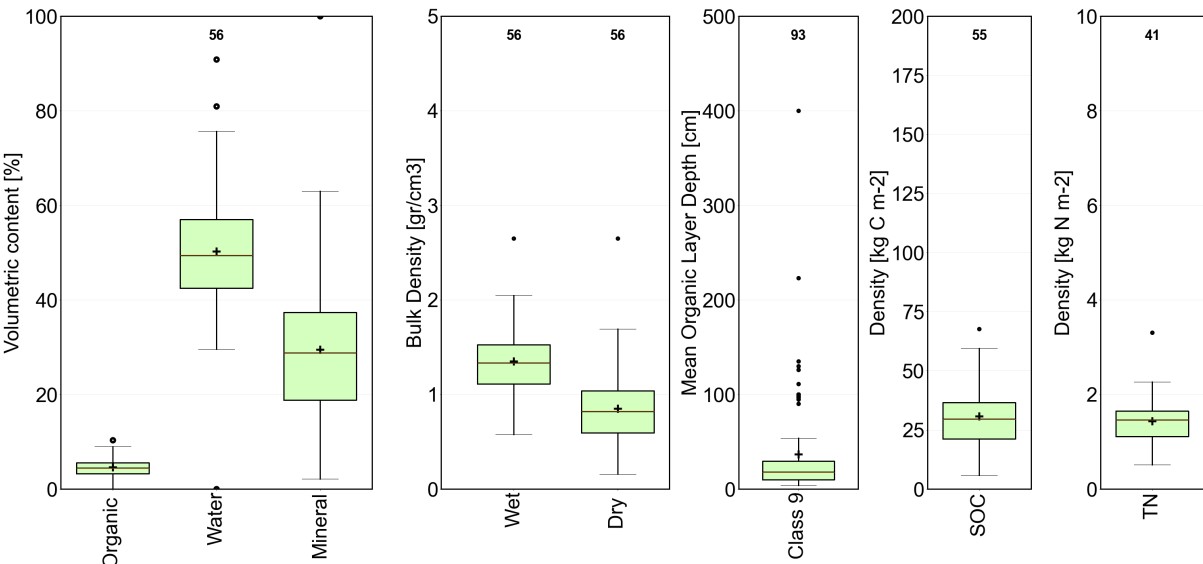

**Figure B25.** Class 9 soil properties based on Palmtag et al. (2022).





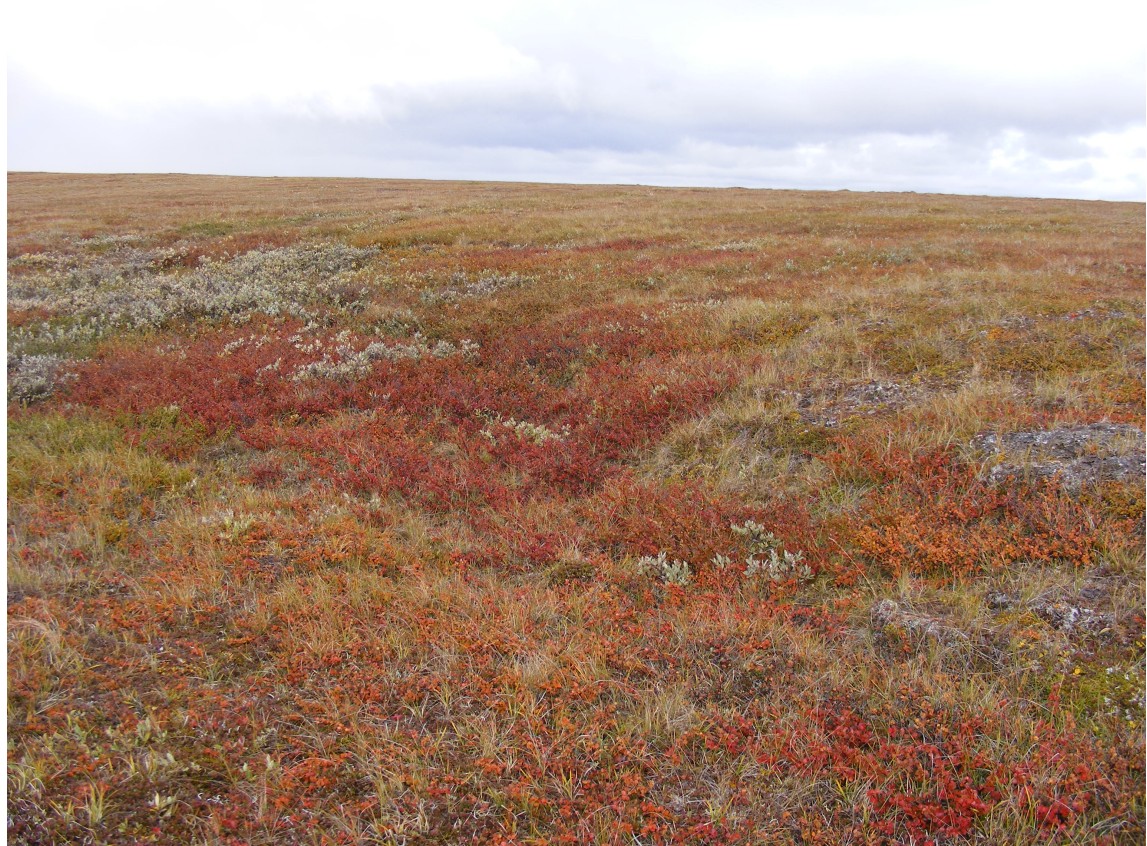

**Figure B26.** Example photograph class 9 (Marina Leibman, 29.08.2014, central Yamal)



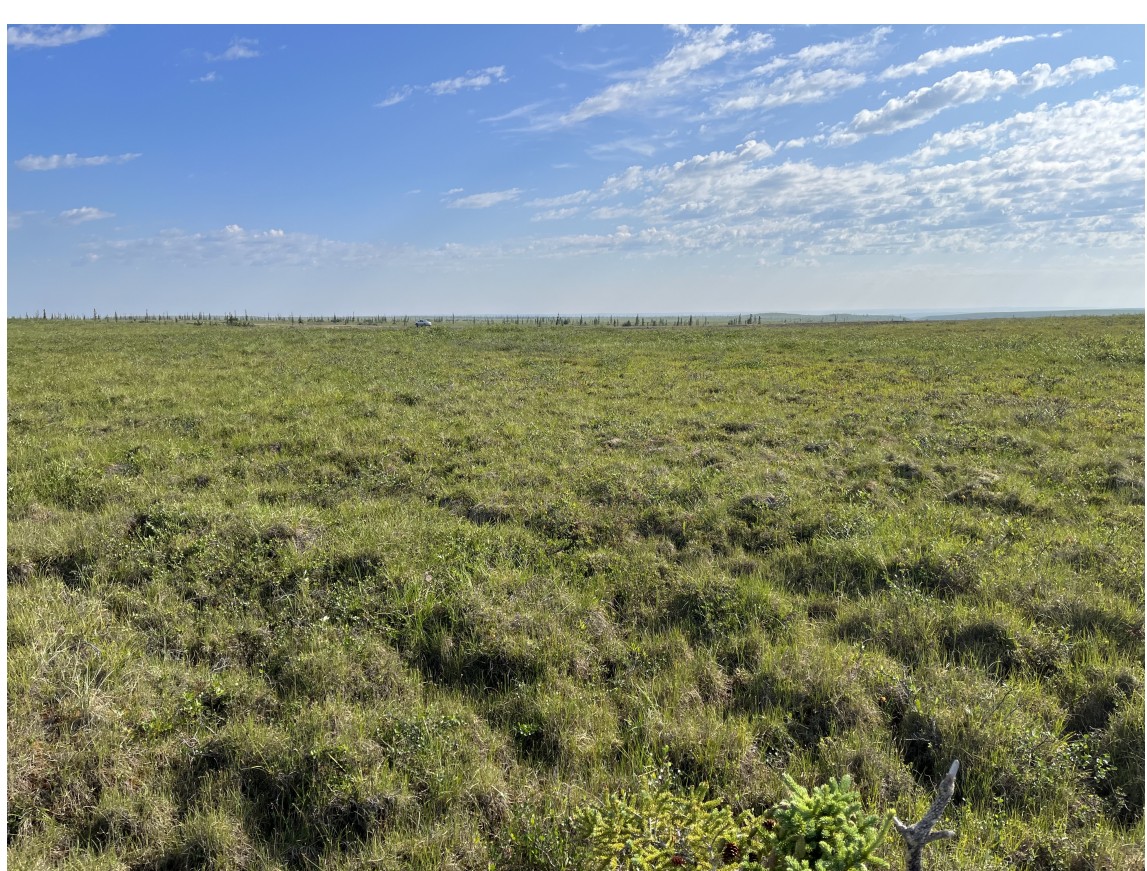

**Figure B27.** Example photograph class 9 (Annett Bartsch, 2023, Inuvik region)



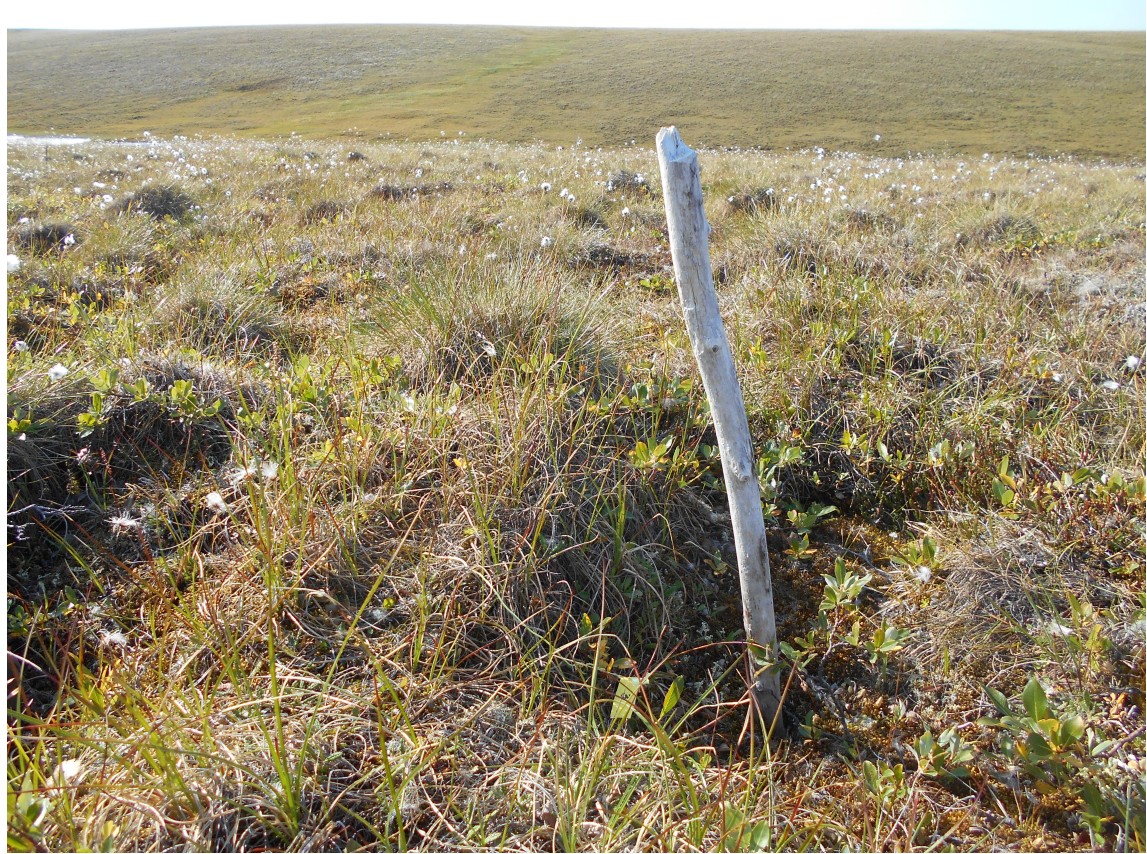

**Figure B28.** Example photograph class 9 (Birgit Heim, 2018, Lena Delta 3rd terrace)





## B10  Class 10

**Description:** moist tundra, abundant moss, prostrate to low shrubs, tussocks; low organic layer thickness, medium mineral volumetric content.

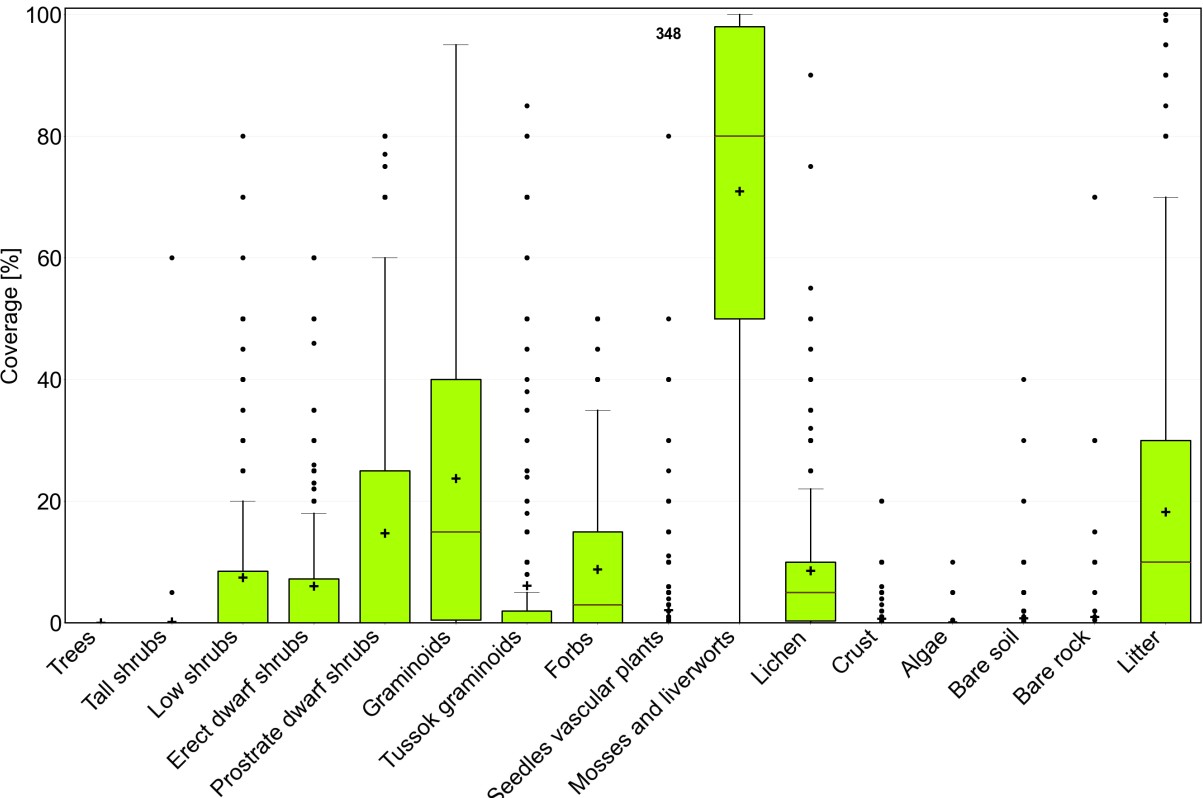

**Figure B29.** Class 10 vegetation properties based on AVA (Zemlianskii et al., 2023).



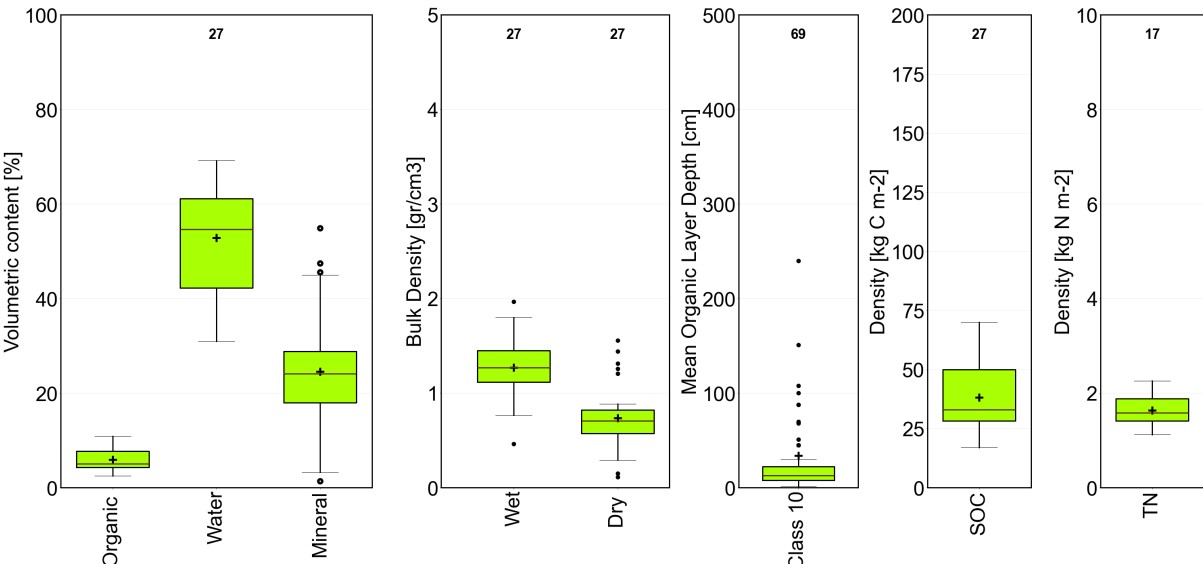

**Figure B30.** Class 10 soil properties based on Palmtag et al. (2022).



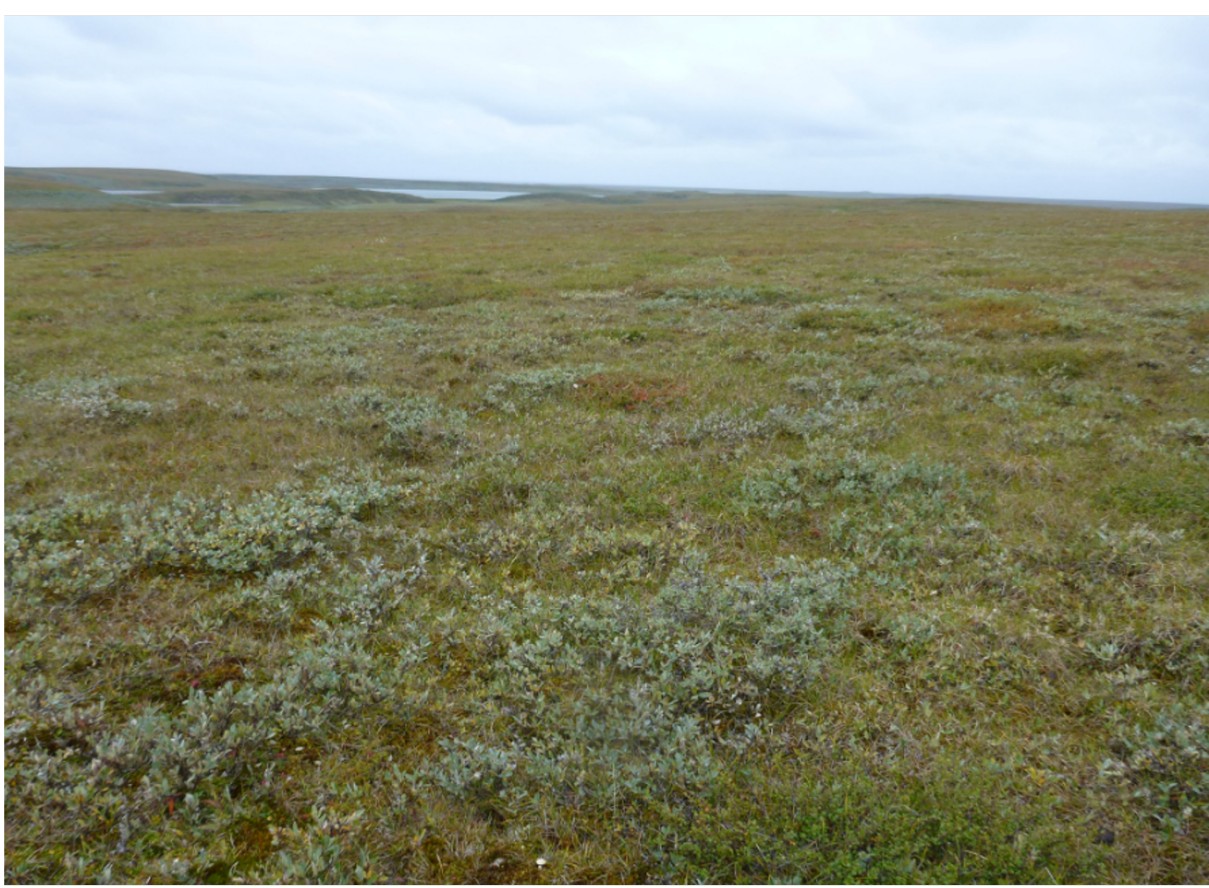

**Figure B31.** Example photograph class 10 (Marina Leibmann, central Yamal)



## B11 Class 11

**Description:** moist tundra, abundant moss, dwarf and low shrubs, tussocks; low organic layer thickness, medium mineral
volumetric content.

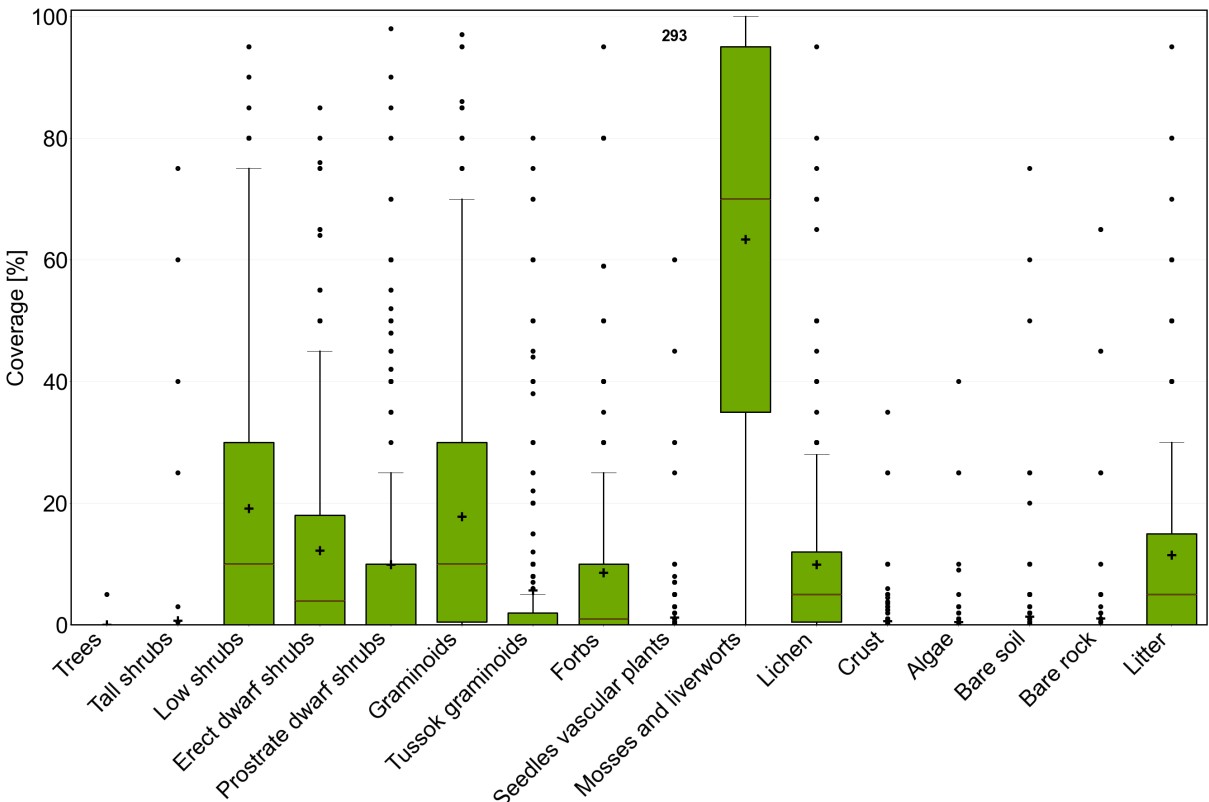

**Figure B32.** Class 11 vegetation properties based on AVA (Zemlianskii et al., 2023).



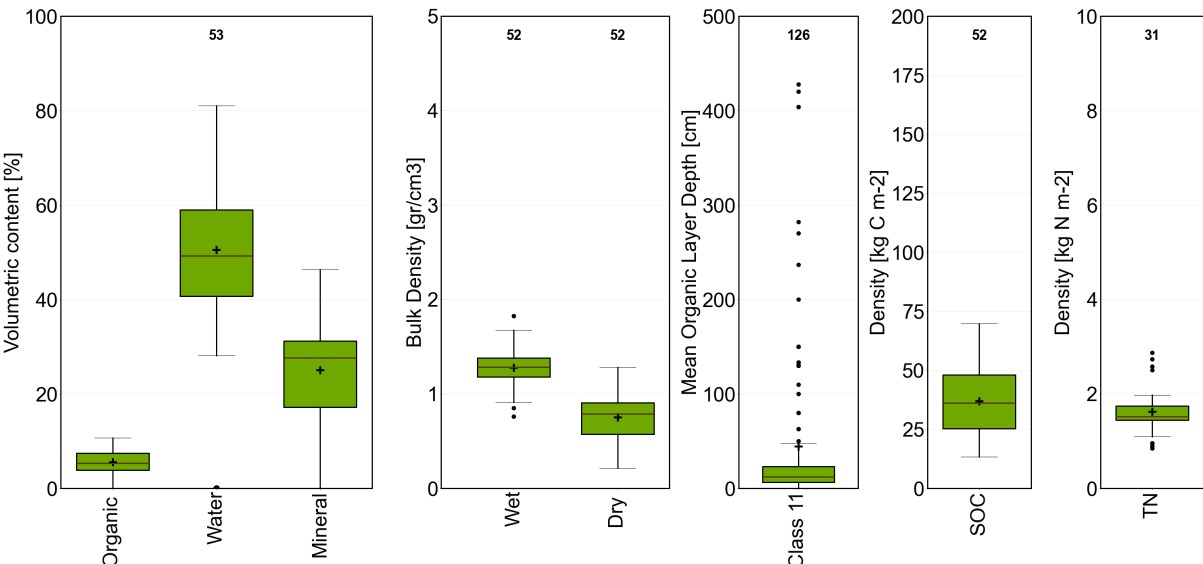

**Figure B33.** Class 11 soil properties based on Palmtag et al. (2022).



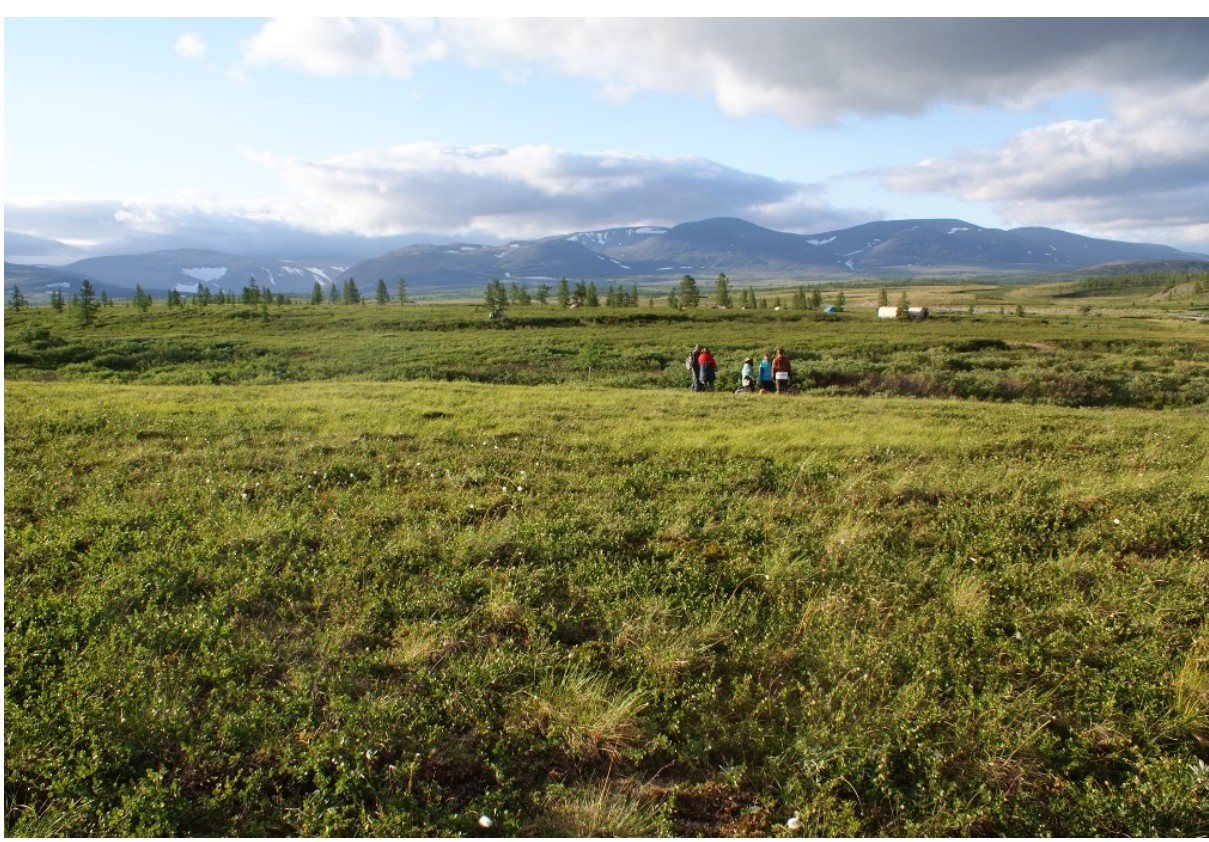

**Figure B34.** Example photograph class 11 (Annett Bartsch 2018, Polar Ural)



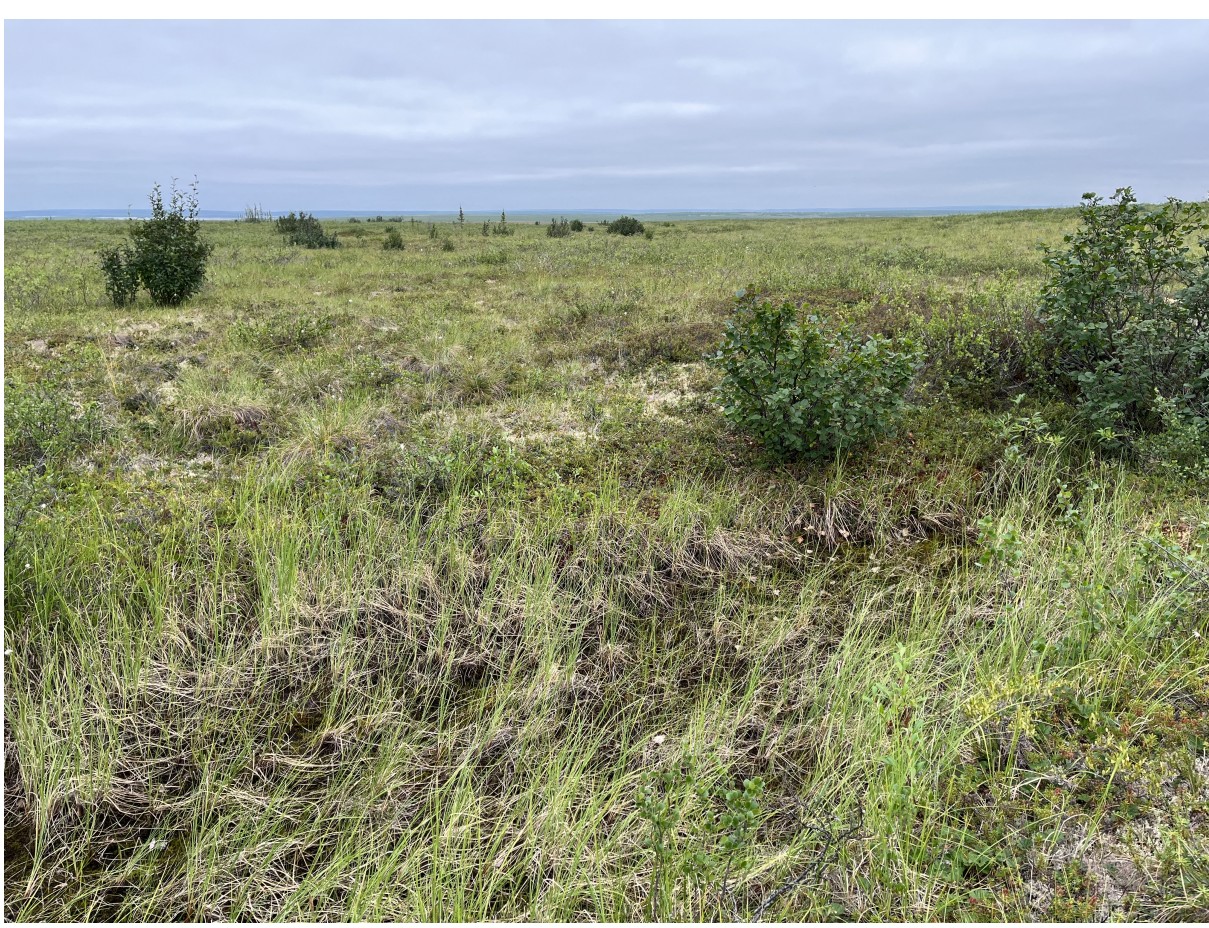

**Figure B35.** Example photograph class 11 (Annett Bartsch 2023, Inuvik region)





## B12   Class 12

**Description:** moist tundra, dense dwarf and low shrubs (sparse tree cover along treeline, woodlands with open stands); medium organic layer thickness, low mineral volumetric content.

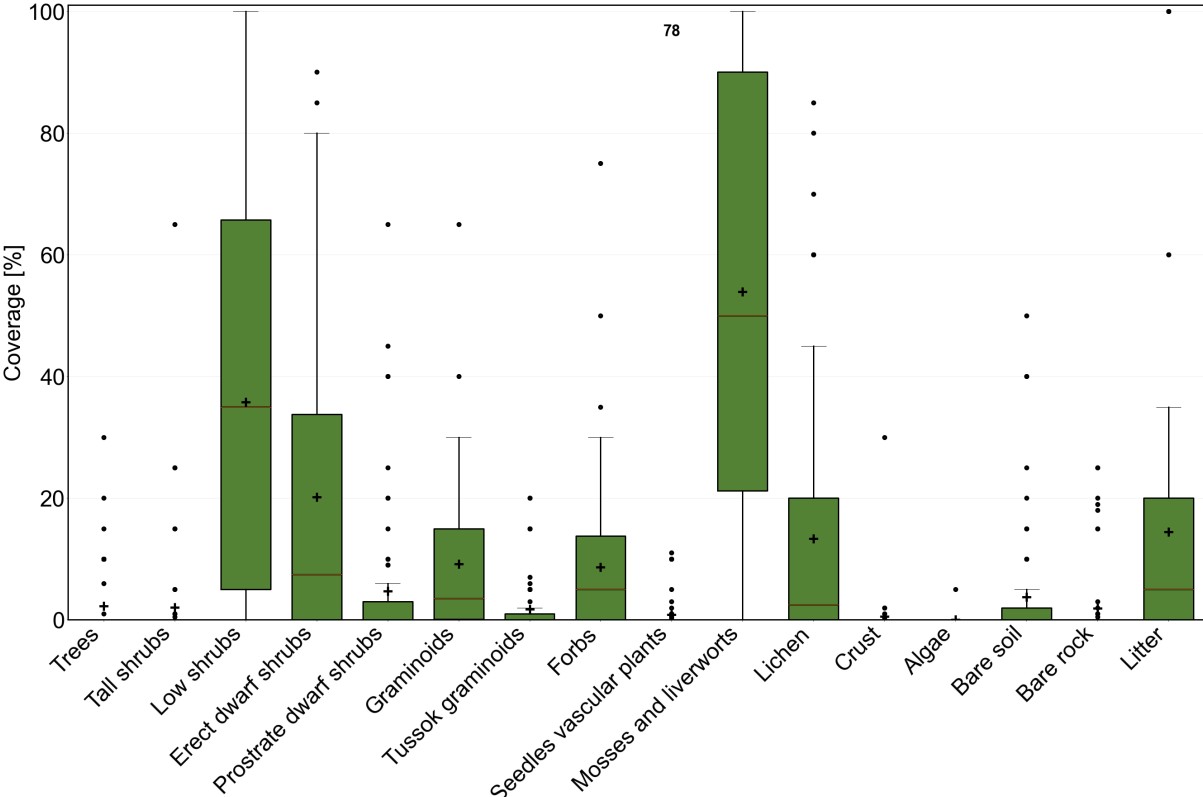

**Figure B36.** Class 12 vegetation properties based on AVA (Zemlianskii et al., 2023).



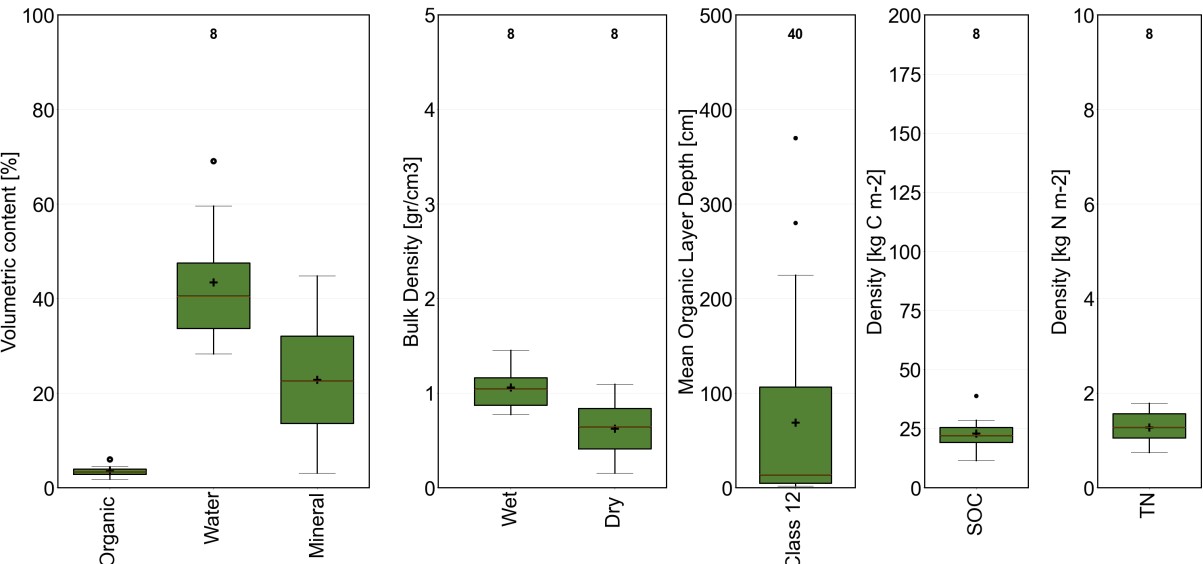

**Figure B37.** Class 12 soil properties based on Palmtag et al. (2022).



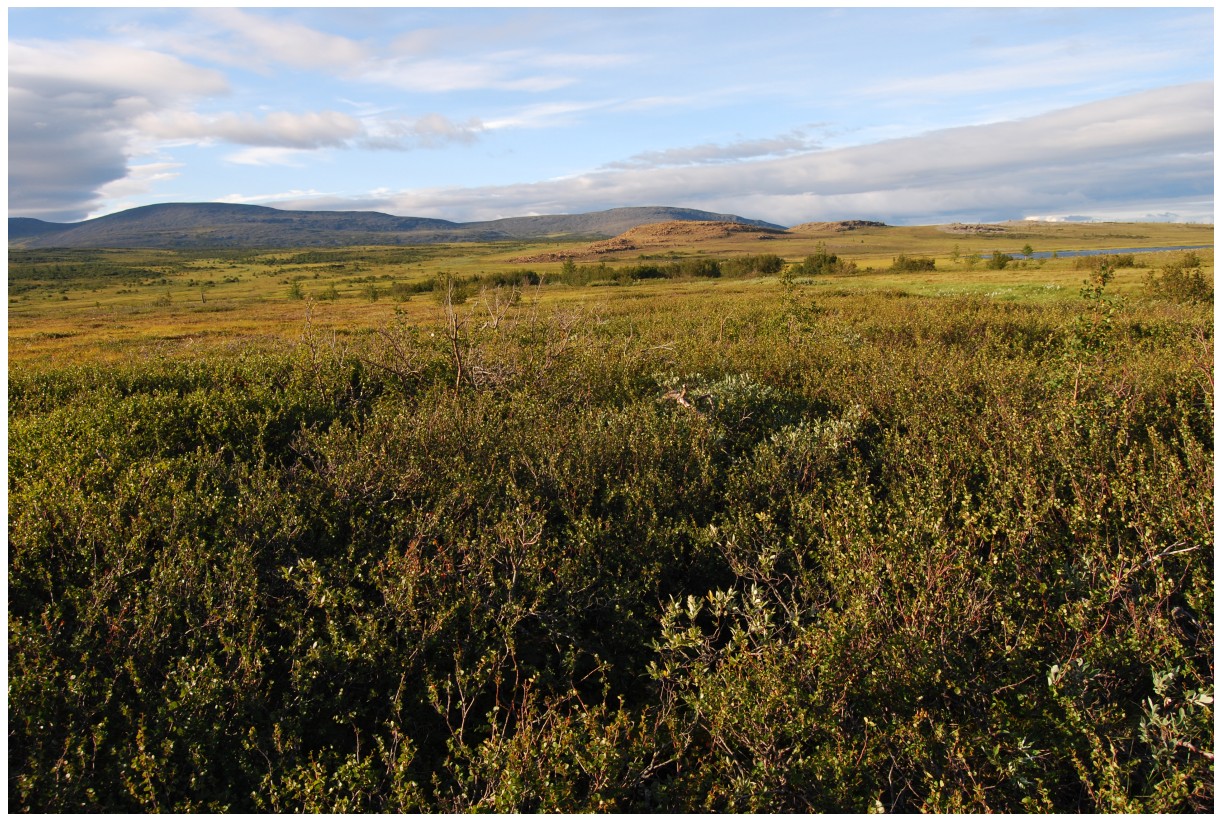

**Figure B38.** Example photograph class 12, foreground (Ksenia Ermokhina , 01.08. 2011, Polar Urals; mostly *Betula nana*, scattered *Salix* ssp.).

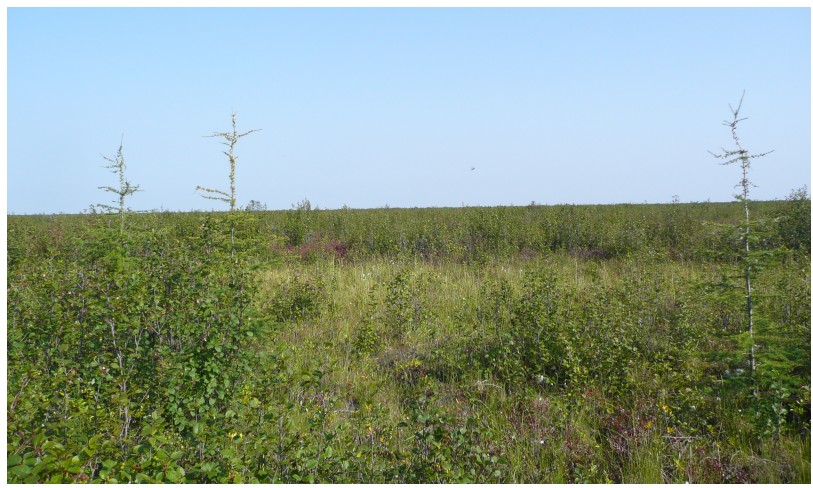

**Figure B39.** Example photograph class 12 (Mareike Wieczorek, 11.08.2012, Kolyma region; mostly *Betula nana* in proximity to patches of class #3.



## B13 Class 13

**Description:** moist to wet tundra, dense dwarf and low shrubs (sparse tree cover along treeline, woodlands with open stands).

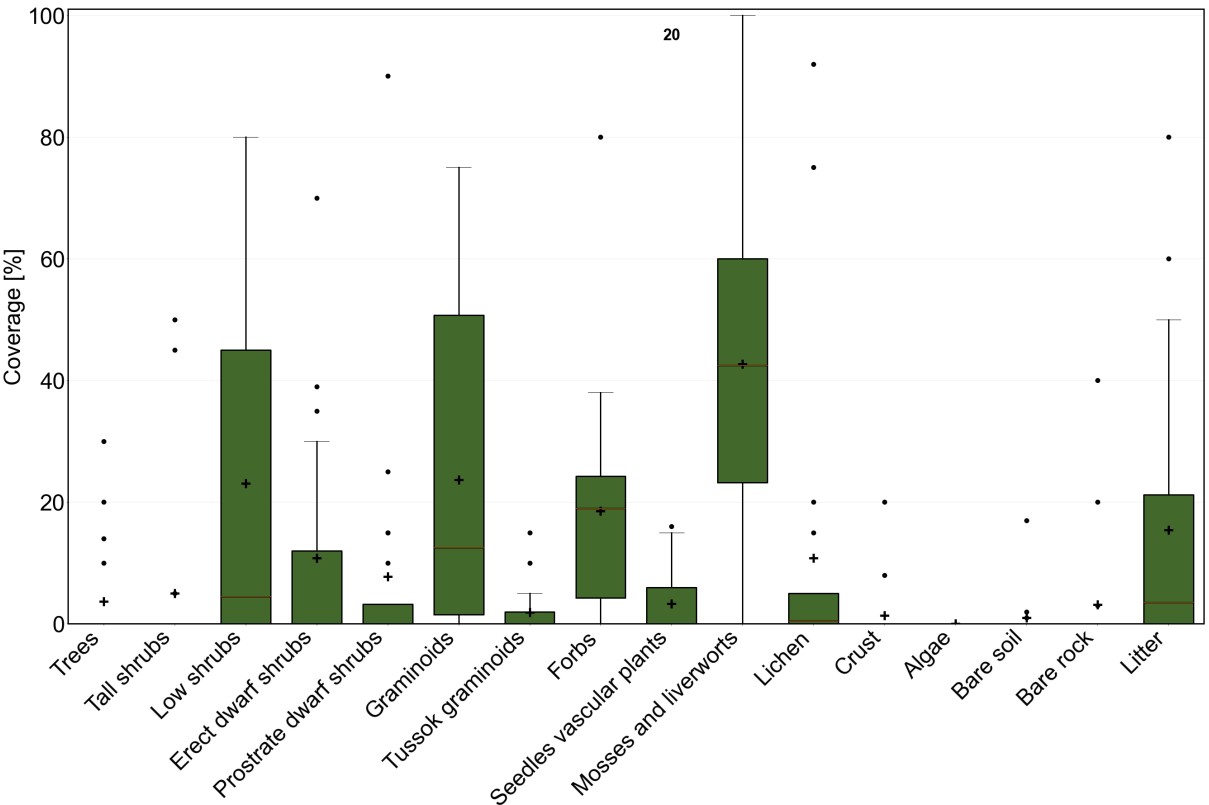

**Figure B40.** Class 13 vegetation properties based on AVA (Zemlianskii et al., 2023).



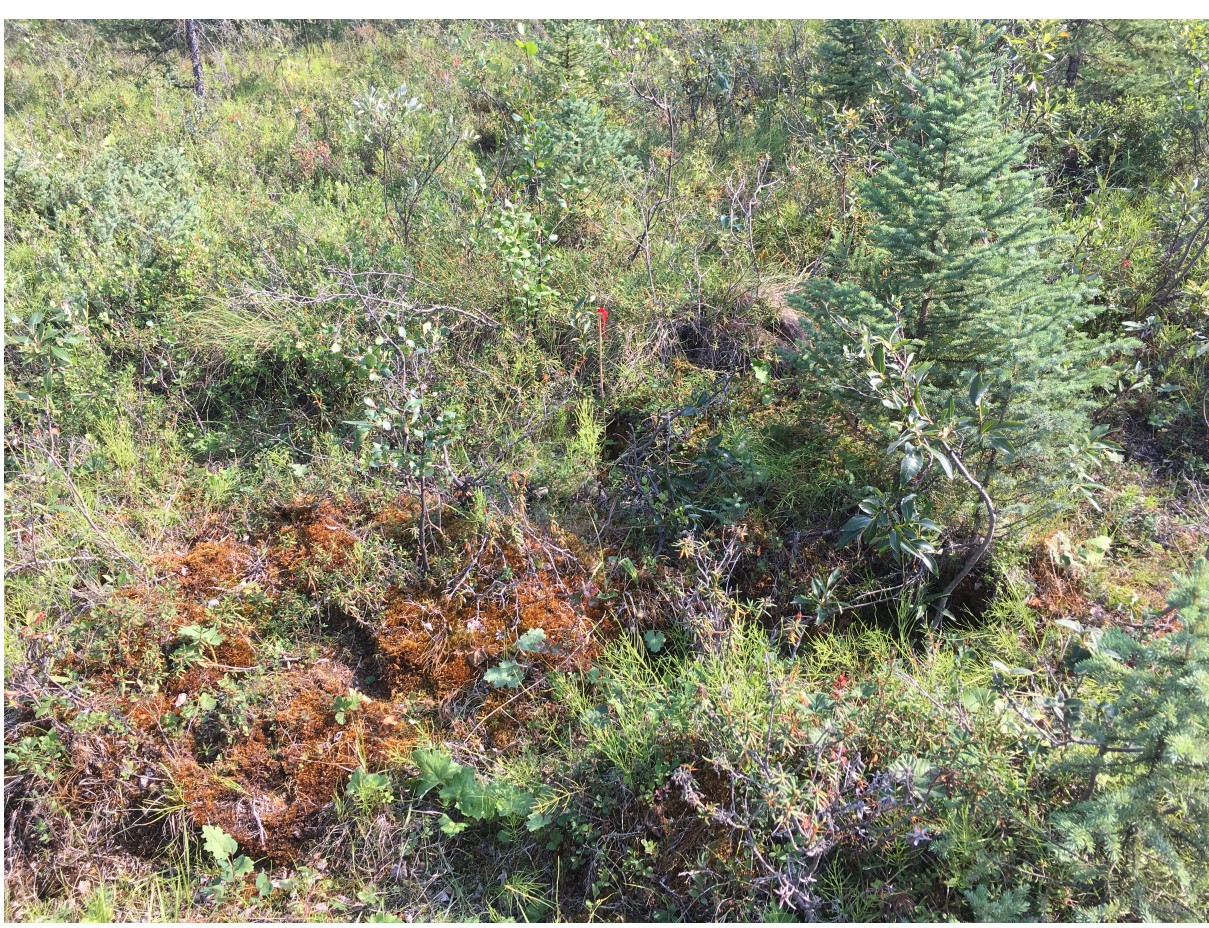

**Figure B41.** Example photograph class 13 (Clemens von Baeckmann, 25.07.2023, Inuvik region).





## B14 Class 14

**Description:** moist tundra, low shrubs; medium organic layer thickness, medium mineral volumetric content.

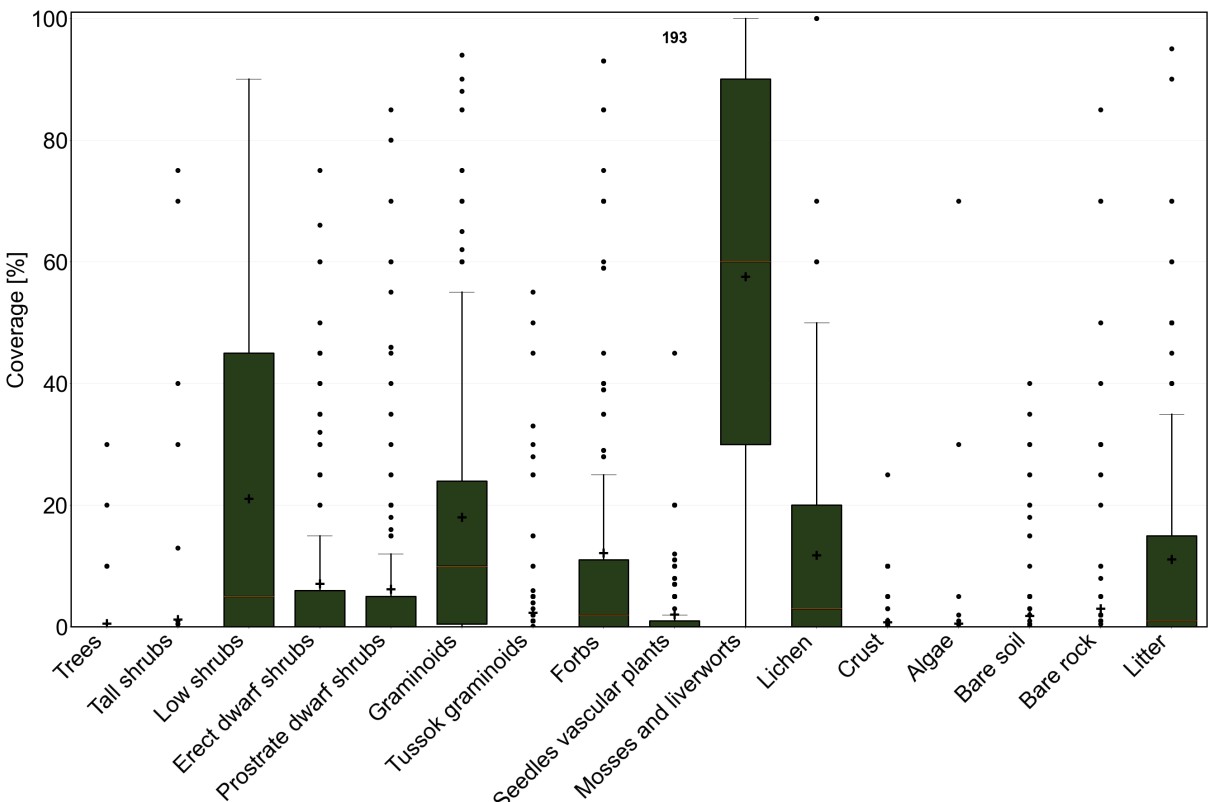

**Figure B42.** Class 14 vegetation properties based on AVA (Zemlianskii et al., 2023).



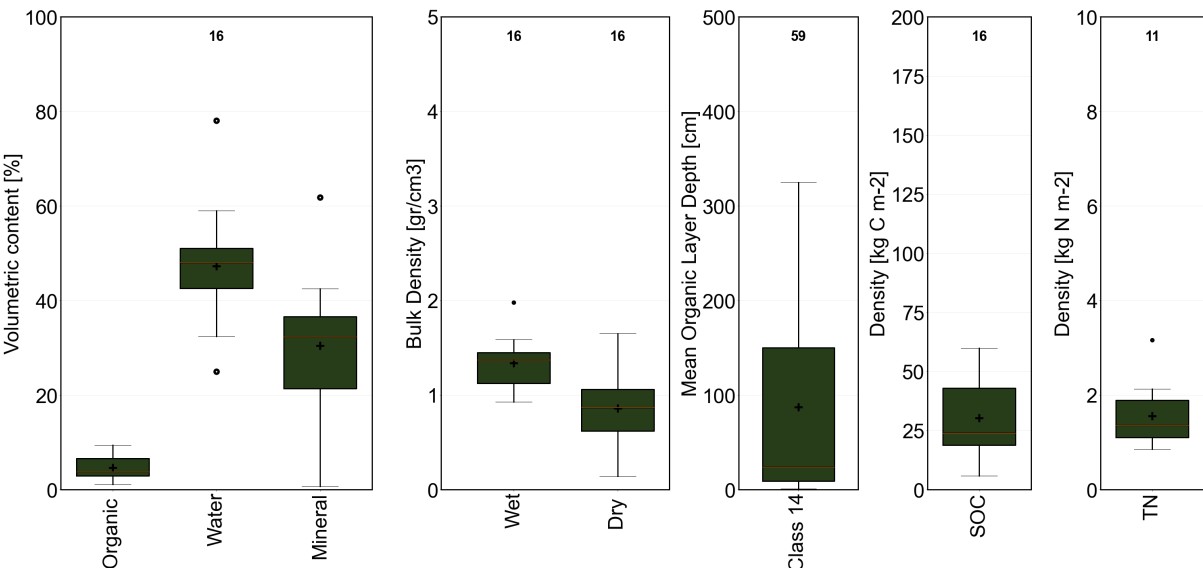

**Figure B43.** Class 14 soil properties based on Palmtag et al. (2022).



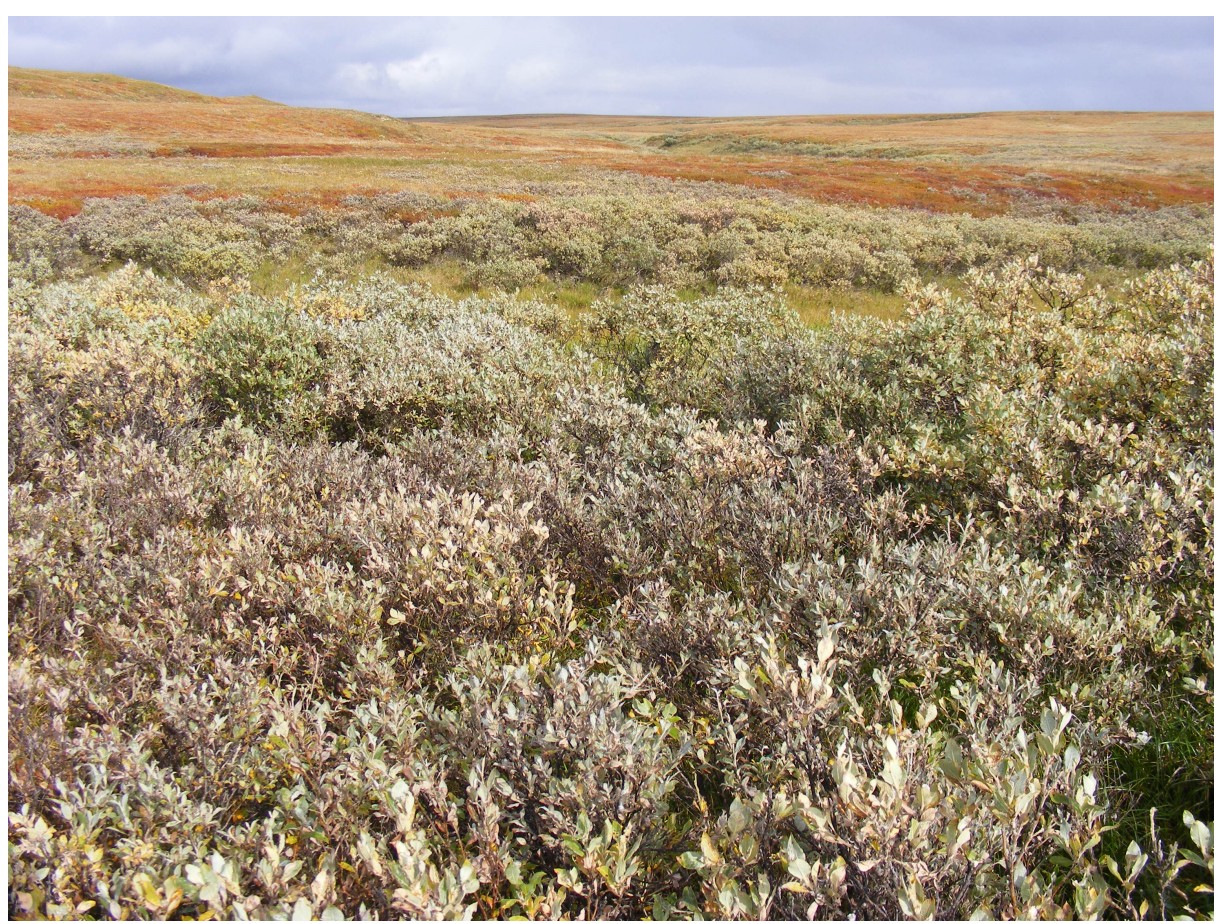

**Figure B44.** Example photograph class 14 (Marina Leibmann, 29.08.2014, central Yamal; mostly *Salix* ssp.)



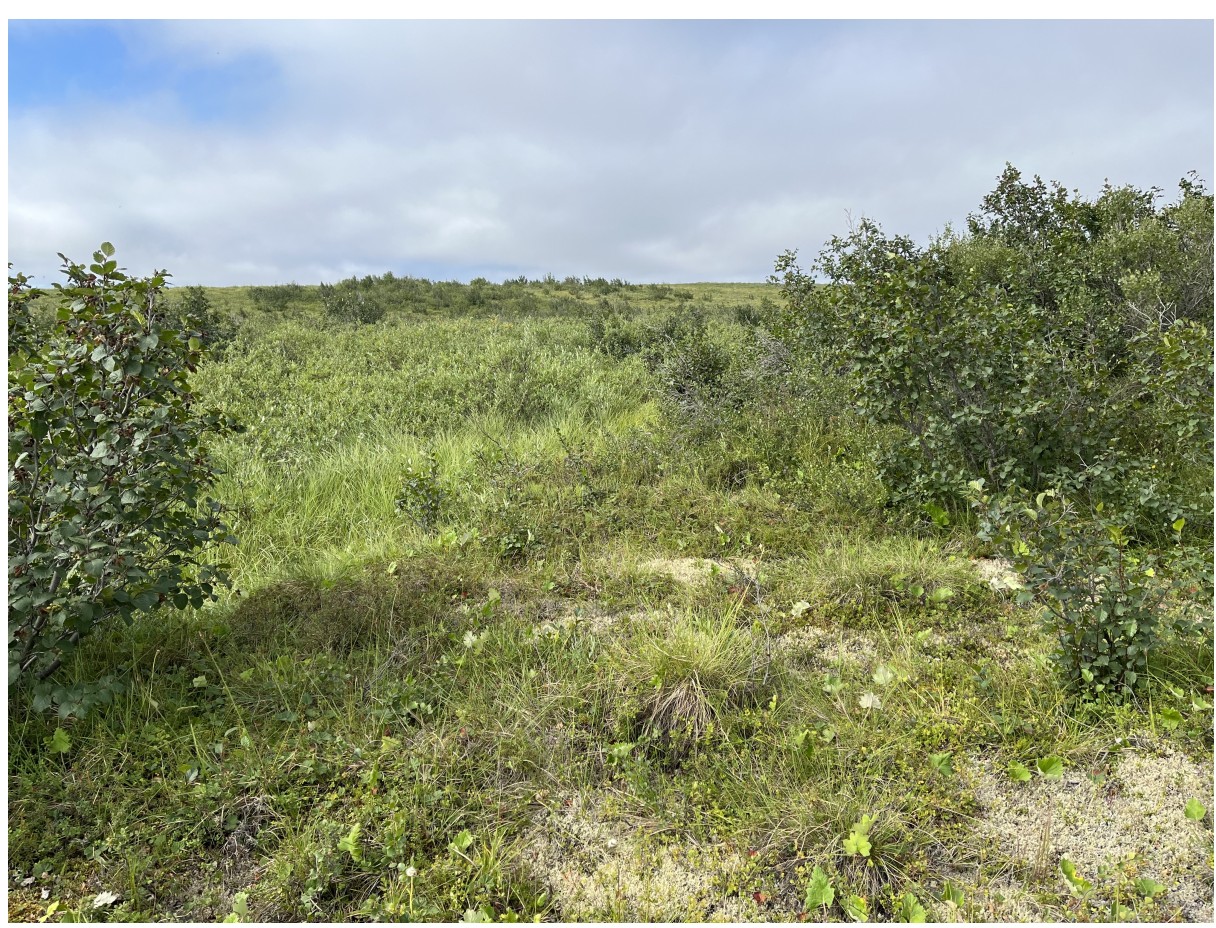

**Figure B45.** Example photograph class 14 (Annett Bartsch 24.07.2023, Inuvik region, backround *Salix* ssp., left and right *Alnus* ssp.)





## B15    Class 15

**Description:** moist to wet tundra, abundant lichen, in some cases partially barren (disturbed).

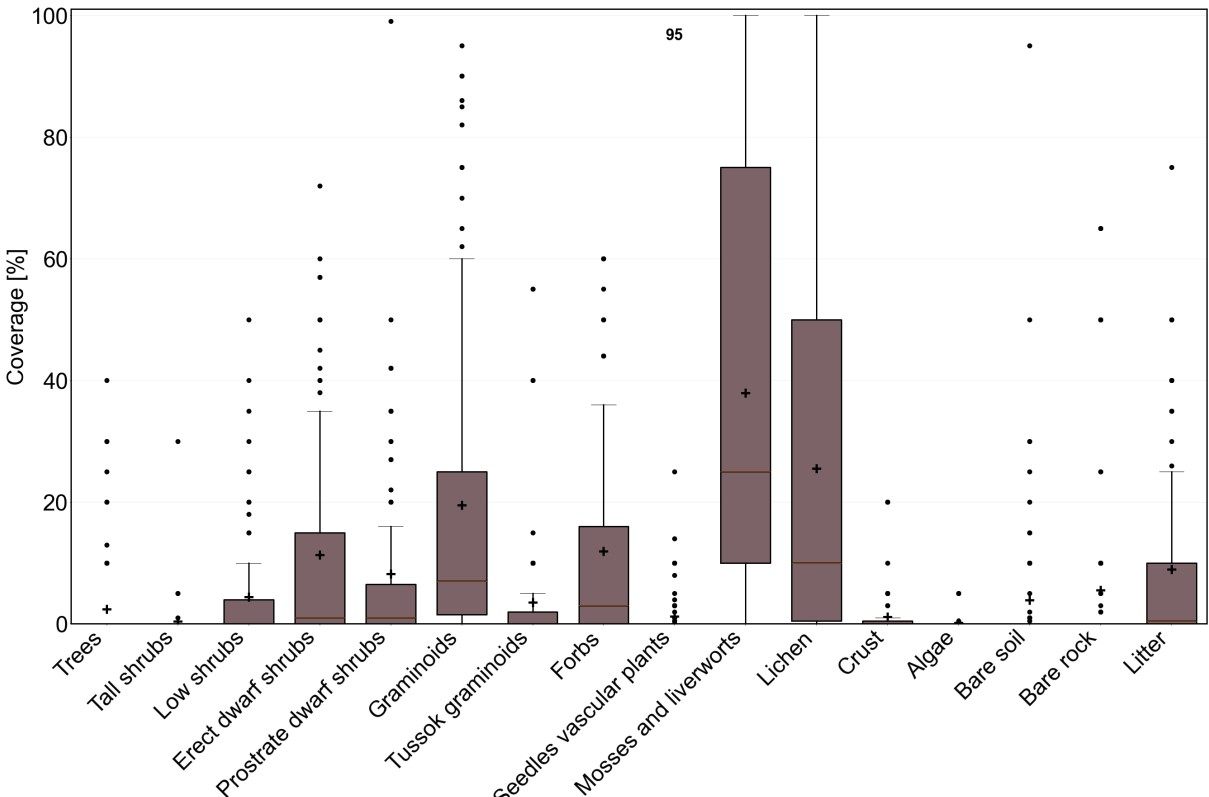

**Figure B46.** Class 15 vegetation properties based on AVA (Zemlianskii et al., 2023).



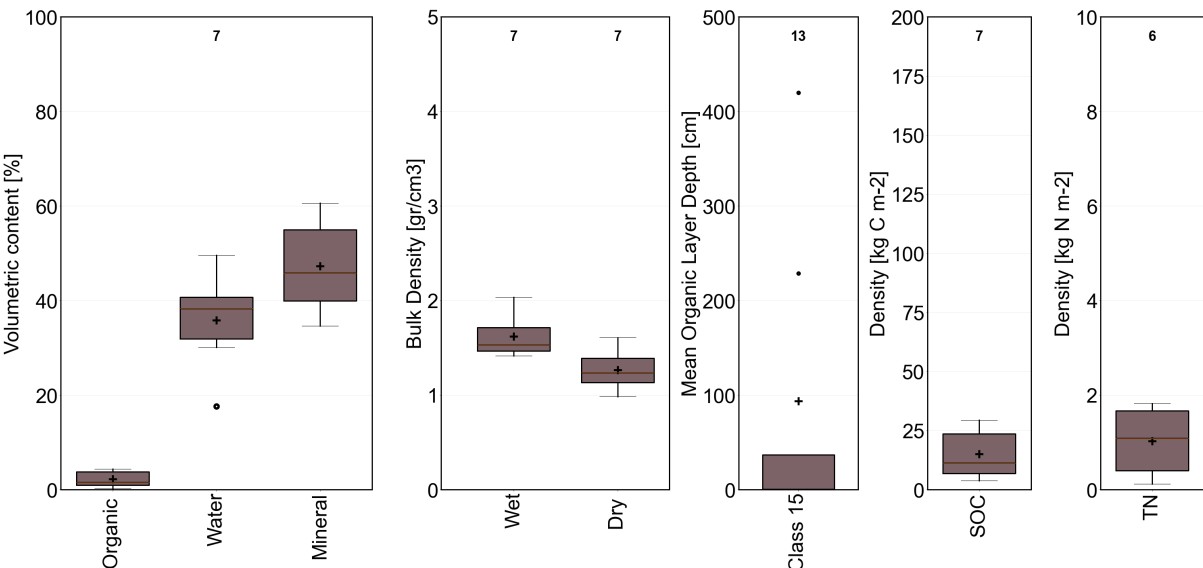

**Figure B47.** Class 15 soil properties based on Palmtag et al. (2022).



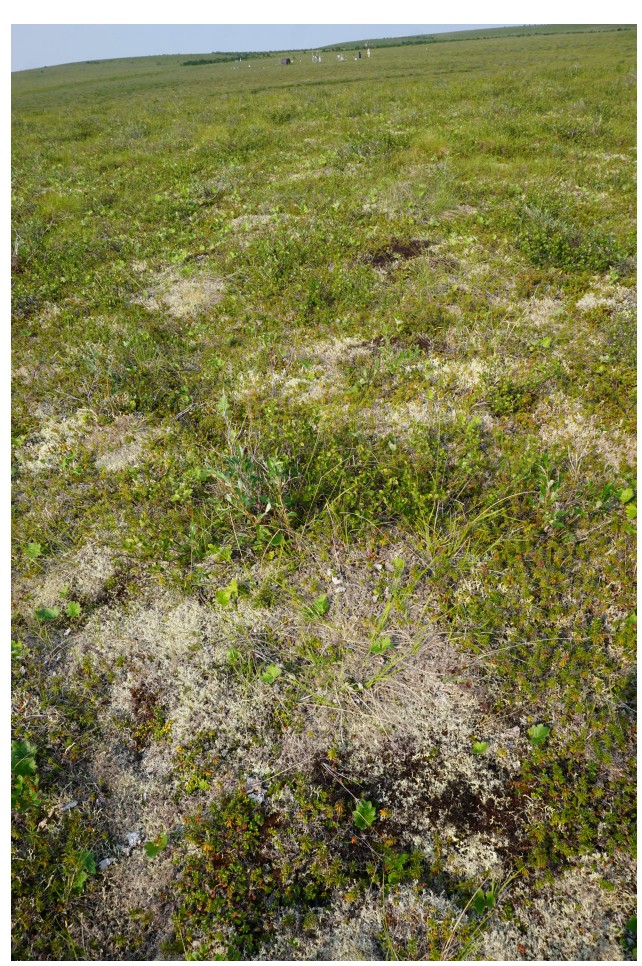

**Figure B48.** Example photograph class 15 (Veronika Döpfer, 17.07.2023, Inuvik region)



**B16    Class 16**

**Description:** moist tundra, abundant forbs, dwarf to tall shrubs; medium organic layer thickness.

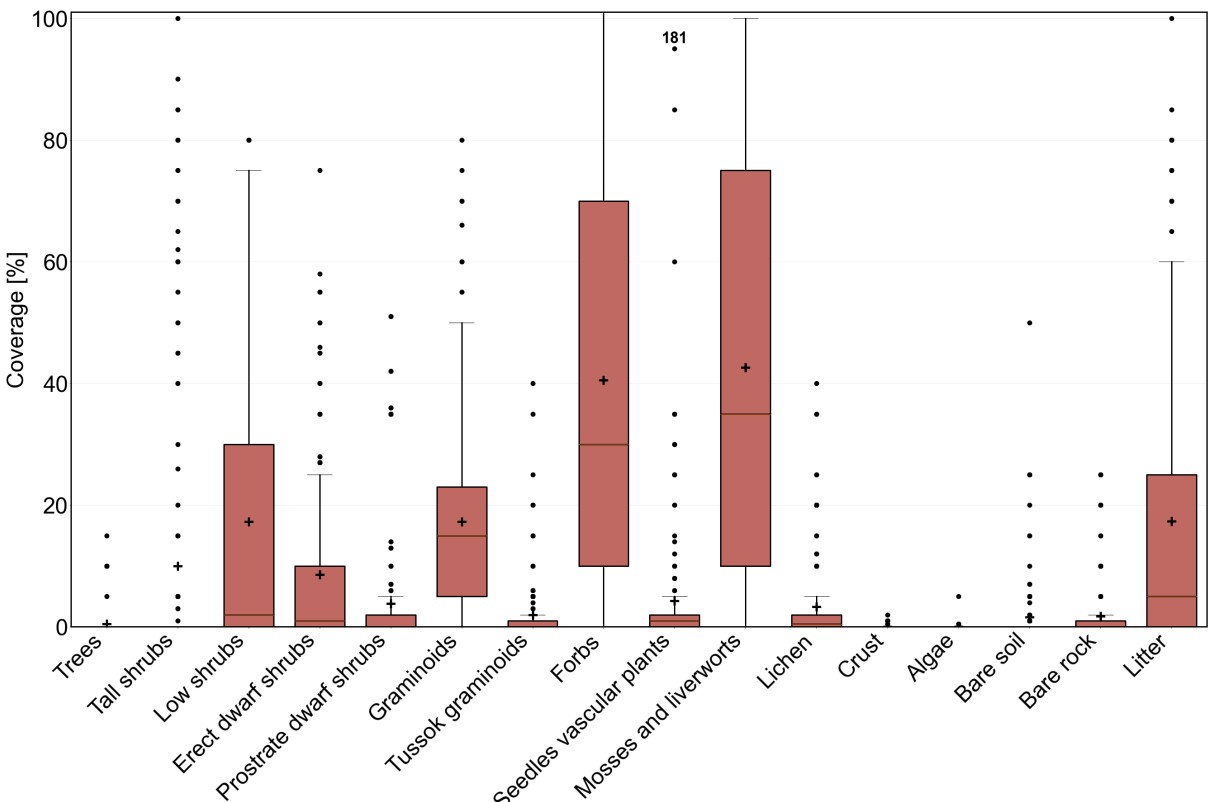

**Figure B49.** Class 16 vegetation properties based on AVA (Zemlianskii et al., 2023).



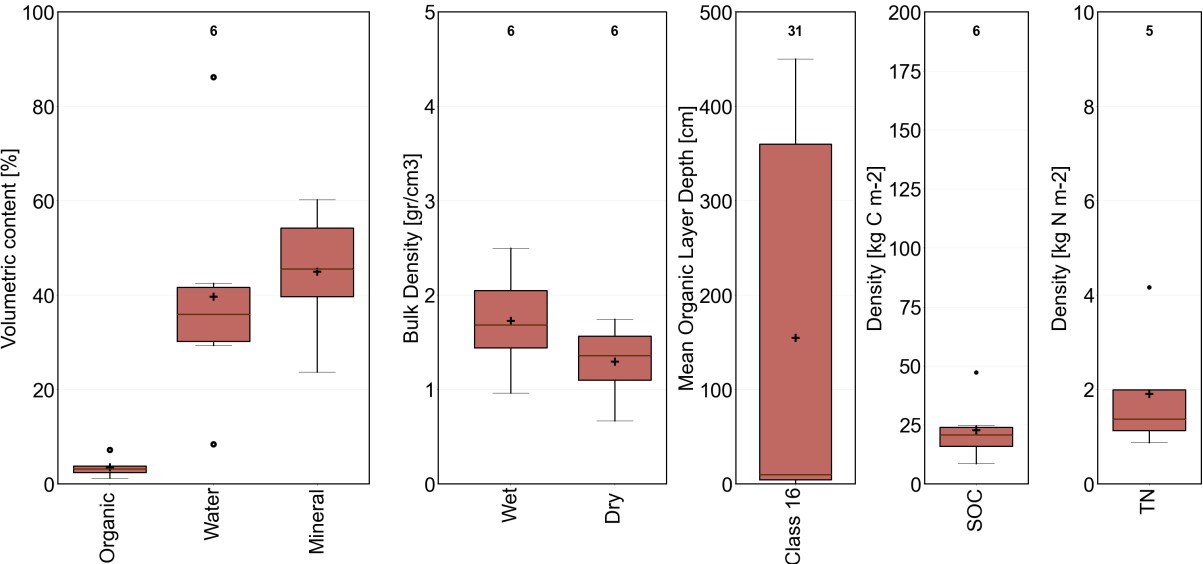

**Figure B50.** Class 16 soil properties based on Palmtag et al. (2022).

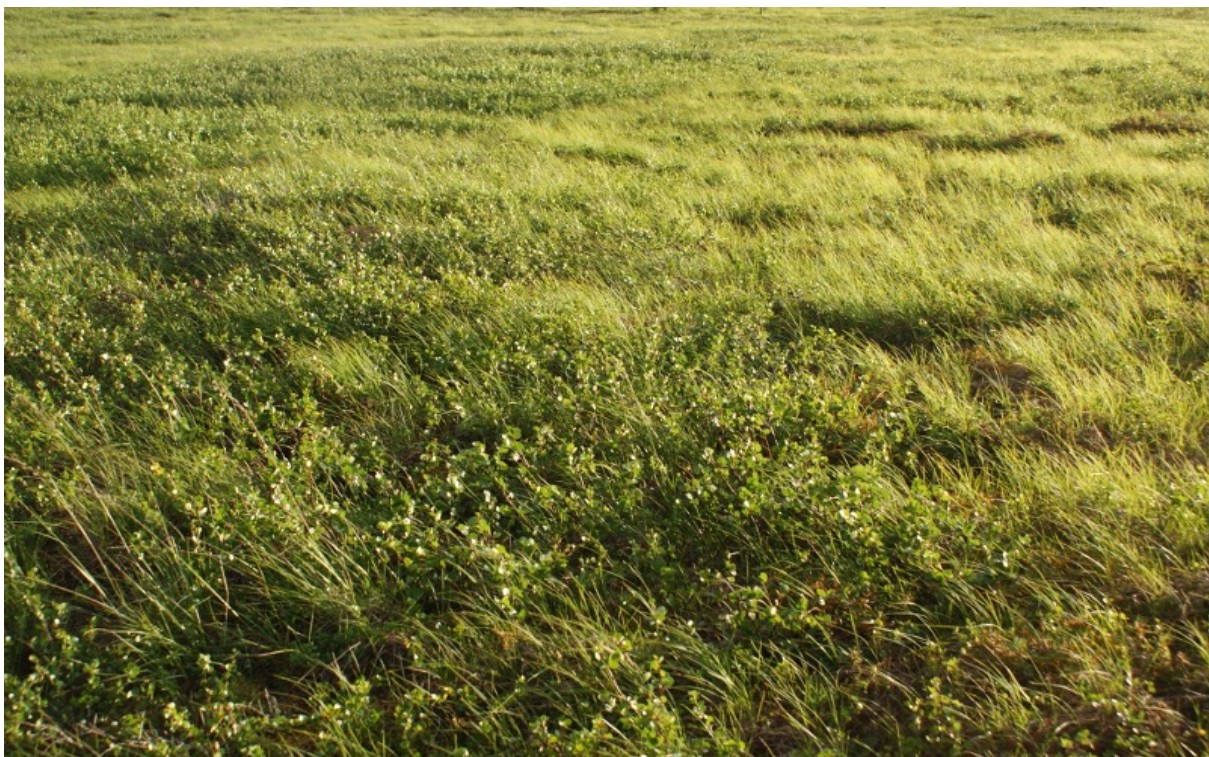

**Figure B51.** Example photograph class 16 (Annett Bartsch 2018, Polar Ural)



## B17    Class 17

**Description:** recently burned or flooded, partially barren.

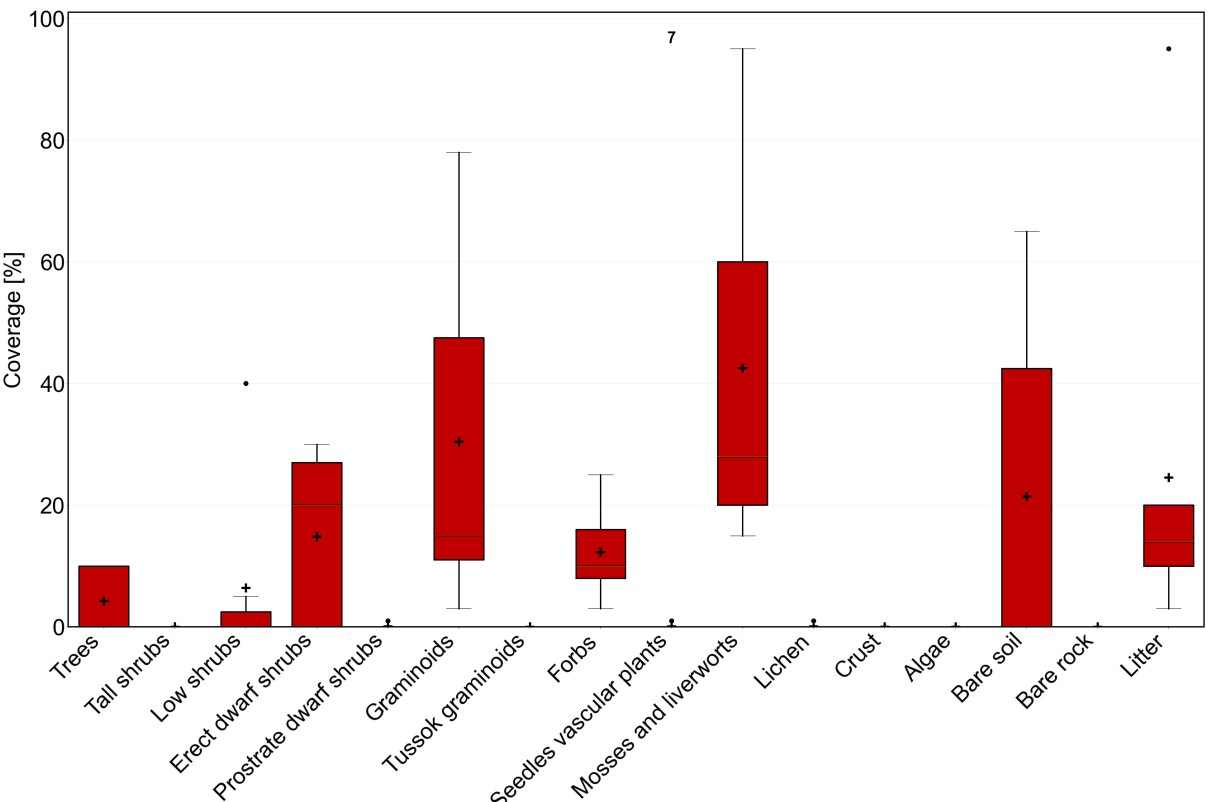

**Figure B52.** Class 17 vegetation properties based on AVA (Zemlianskii et al., 2023).





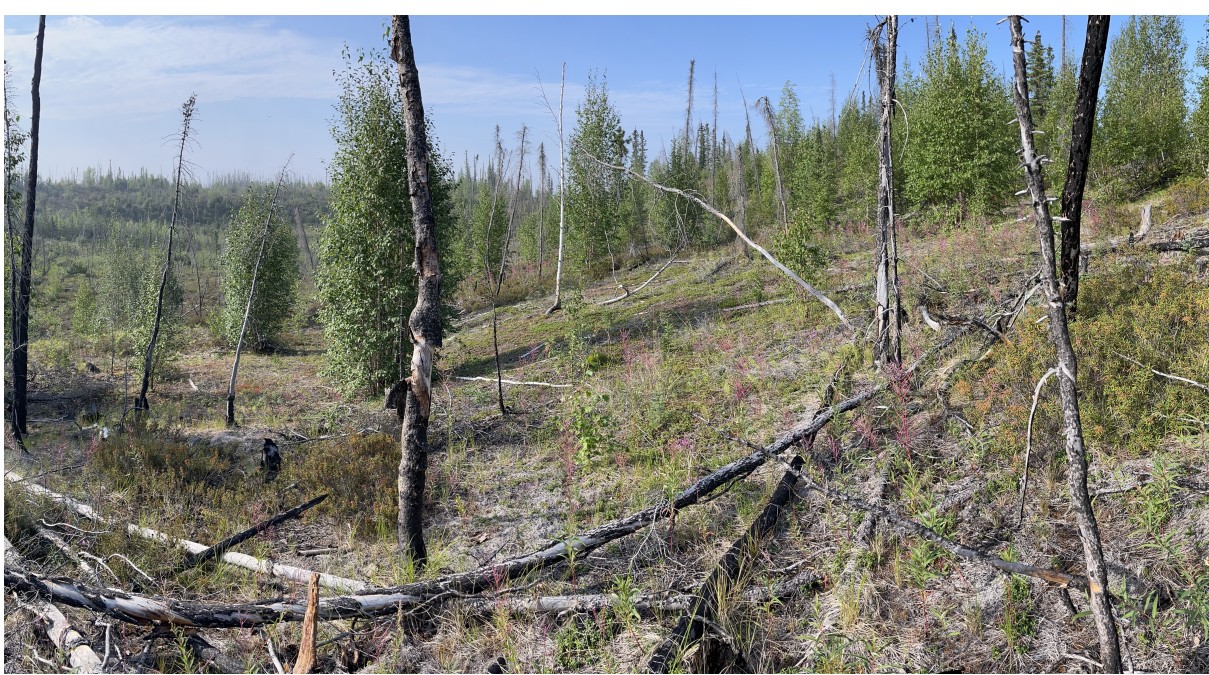

**Figure B53.** Example photograph class 17 (Annett Bartsch 27.07.2023, Inuvik)



## B18 Class 18

**Description:** Forest (deciduous) with dwarf to tall shrubs.

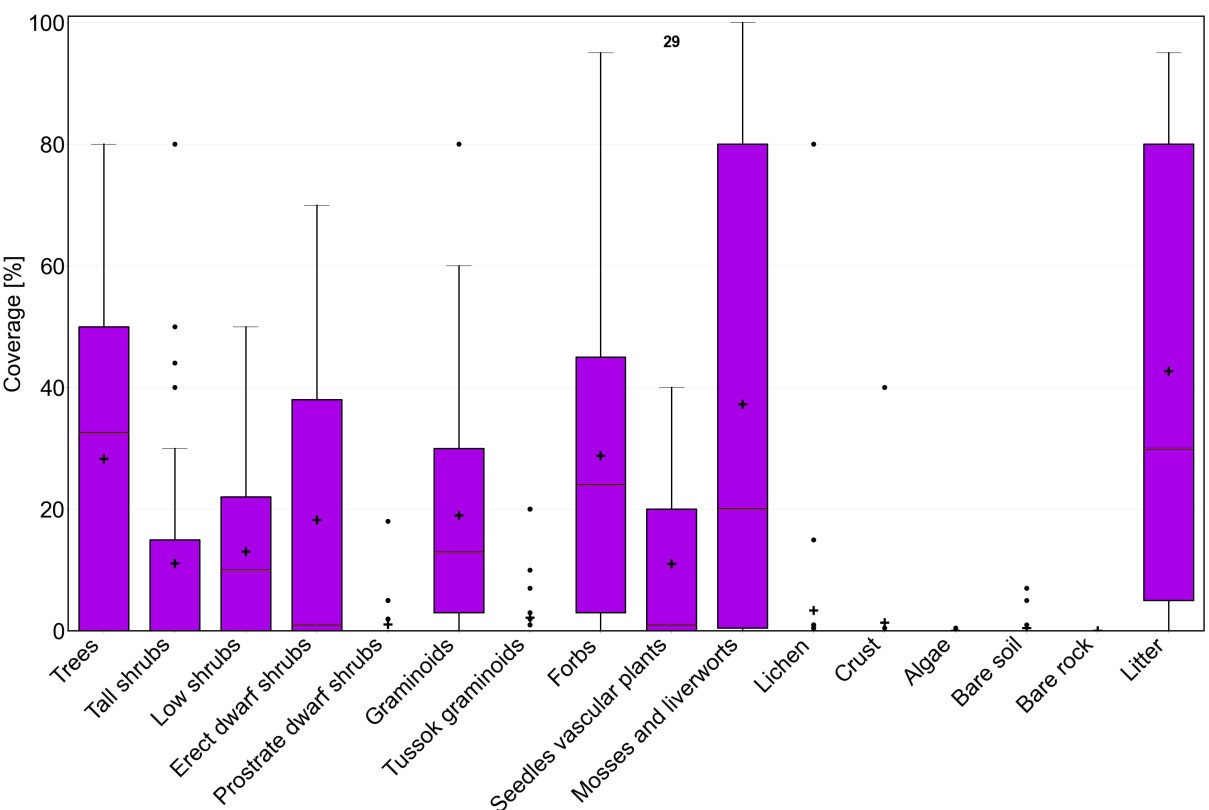

**Figure B54.** Class 18 vegetation properties based on AVA (Zemlianskii et al., 2023).



## B19 Class 19

**Description:** Forest (mixed) with dwarf to tall shrubs.

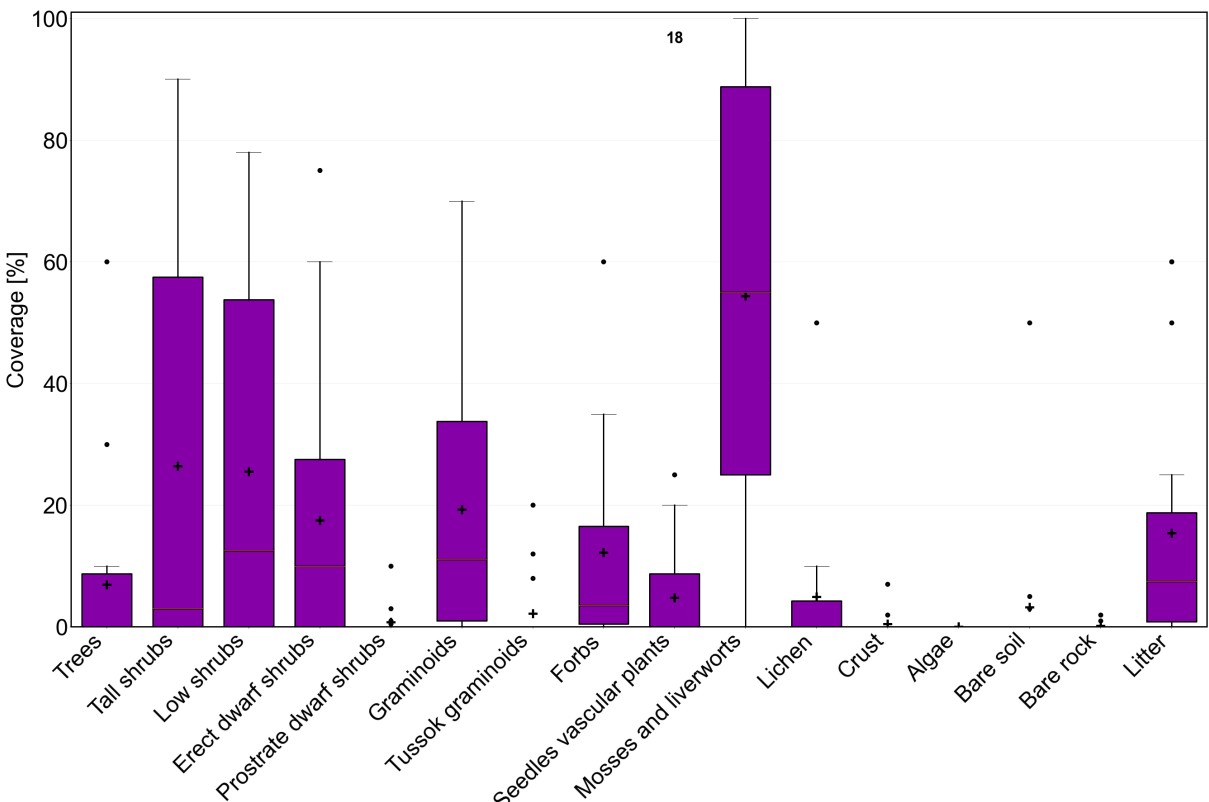

**Figure B55.** Class 19 vegetation properties based on AVA (Zemlianskii et al., 2023).



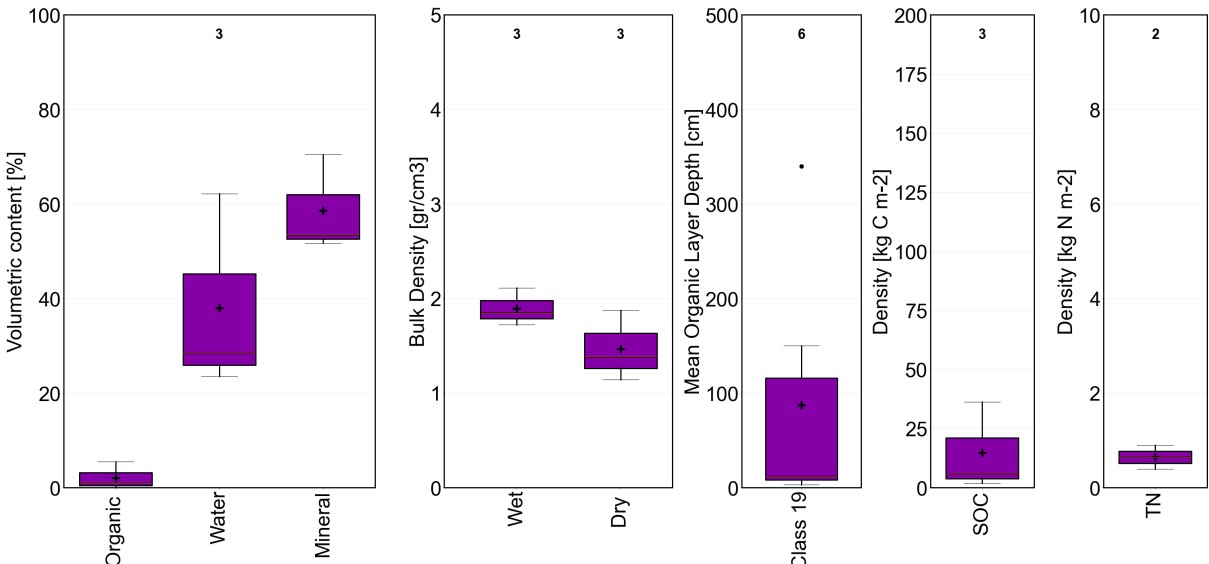

**Figure B56.** Class 19 soil properties based on Palmtag et al. (2022).





## B20 Class 20

**Description:** Forest (needle leave) with dwarf and low shrubs.

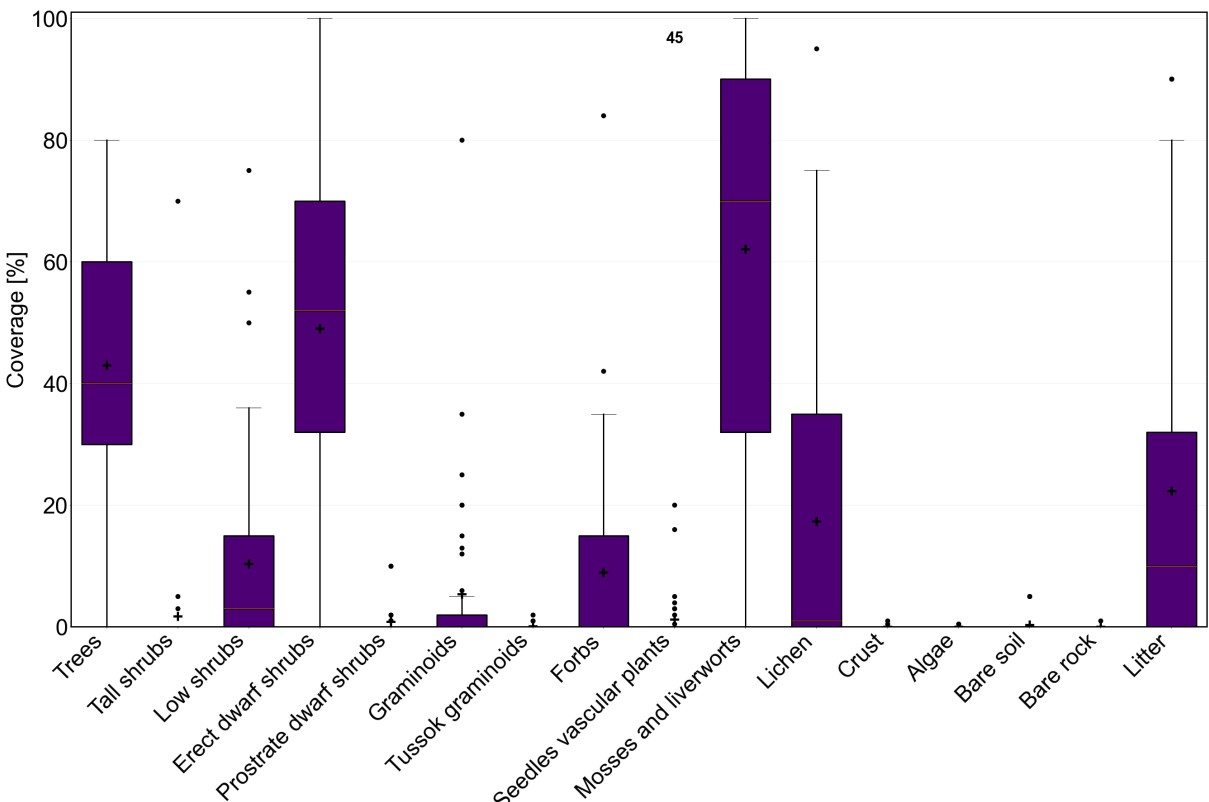

**Figure B57.** Class 20 vegetation properties based on AVA (Zemlianskii et al., 2023).



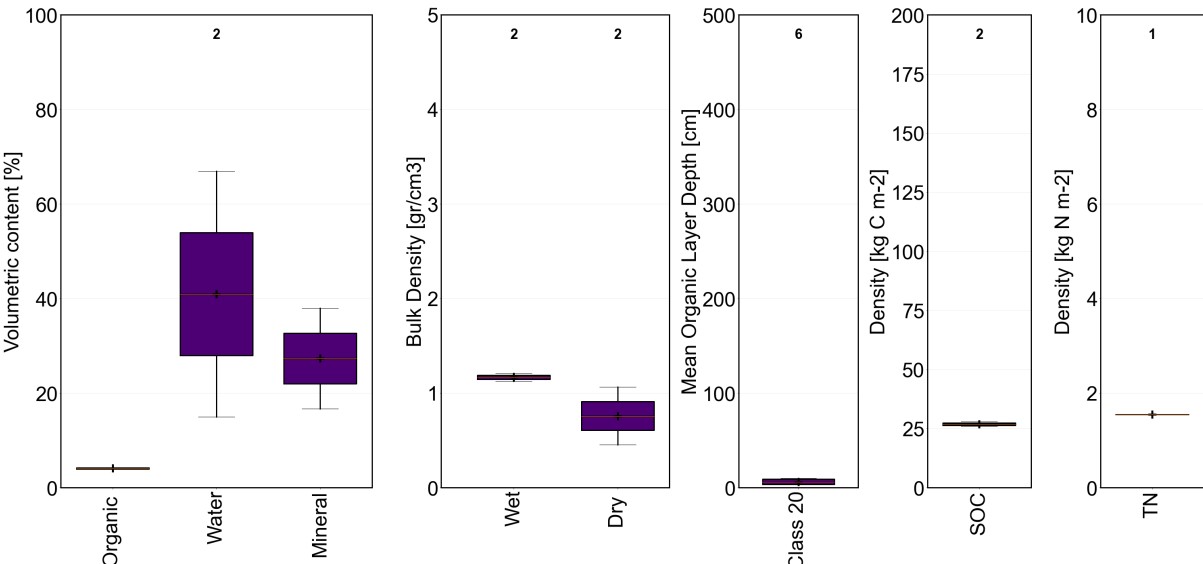

**Figure B58.** Class 20 soil properties based on Palmtag et al. (2022).





680  **B21  Class 21**

**Description:** Partially barren, dry.

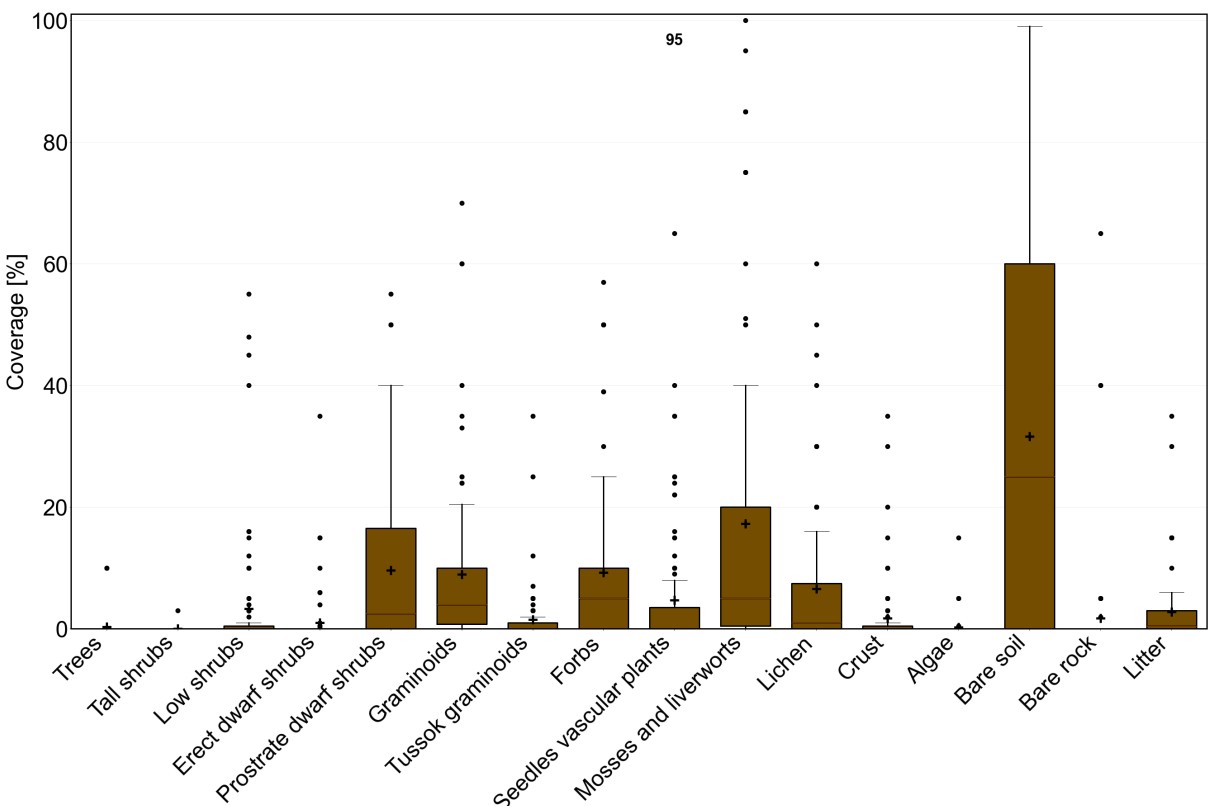

**Figure B59.** Class 21 vegetation properties based on AVA (Zemlianskii et al., 2023).



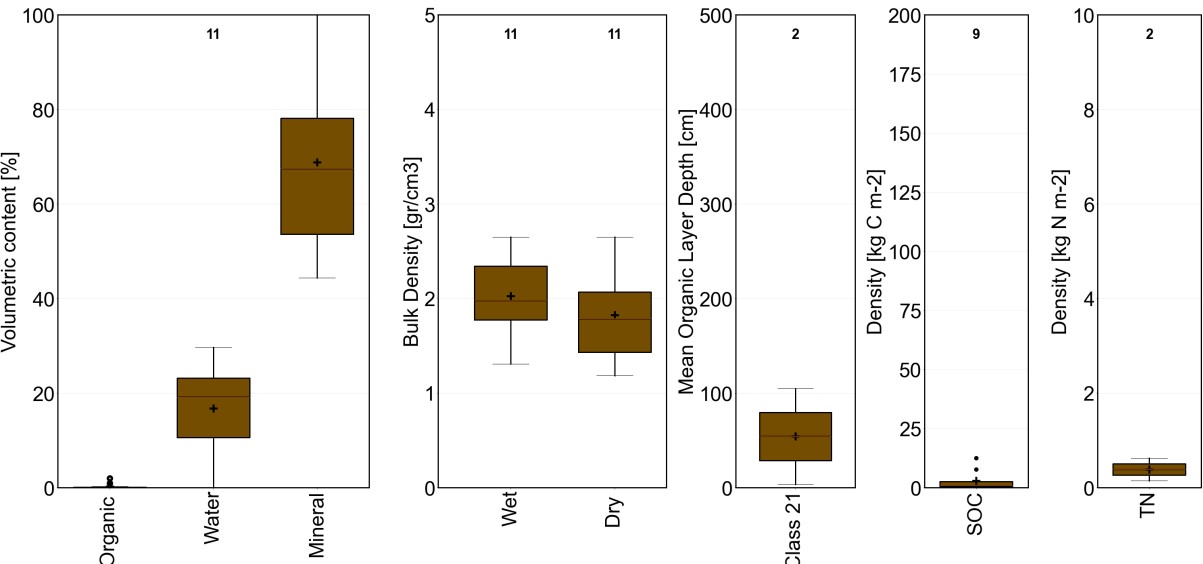

**Figure B60.** Class 21 soil properties based on Palmtag et al. (2022).



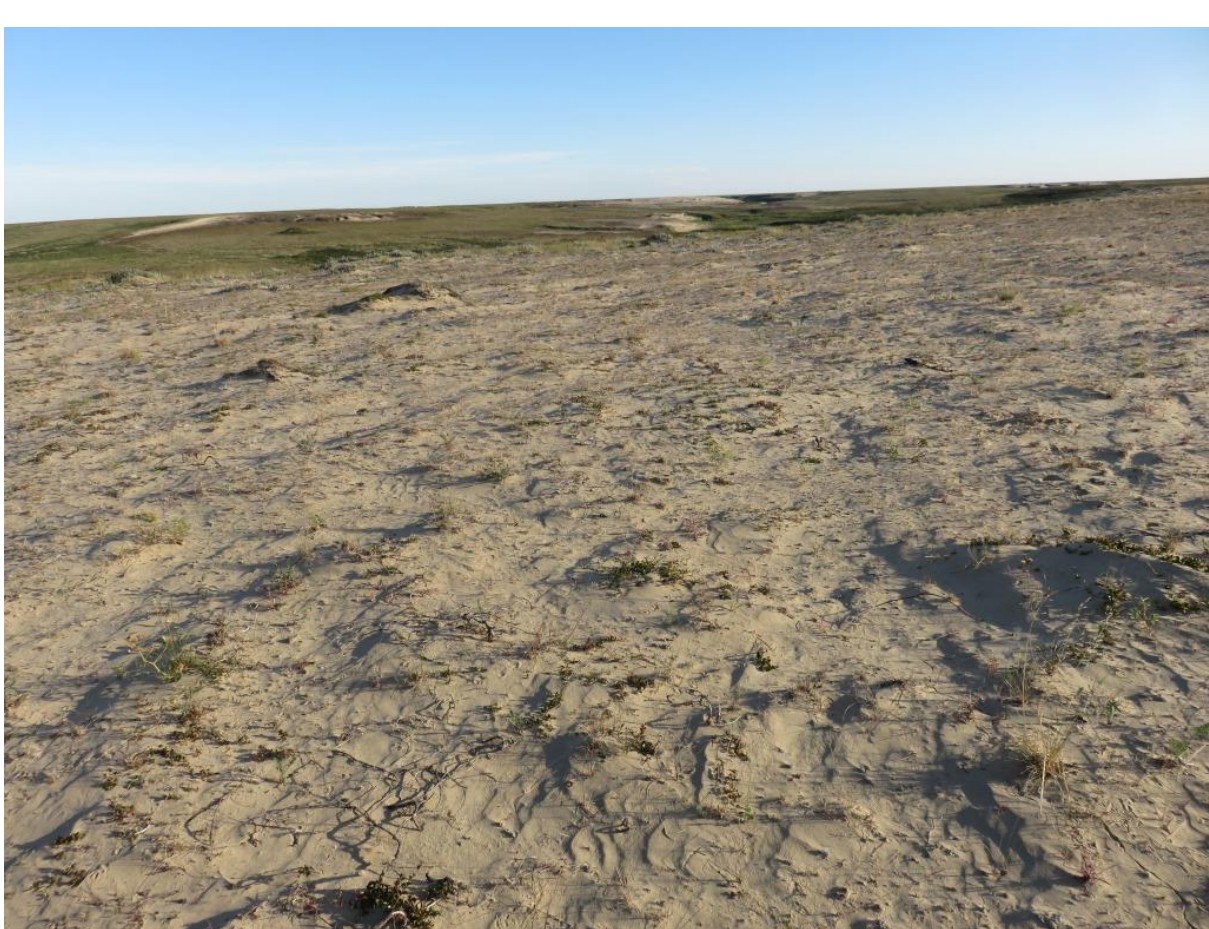

**Figure B61.** Example photograph class 21 (Olga Khitun, 2017, Tazovskiy Peninsula)



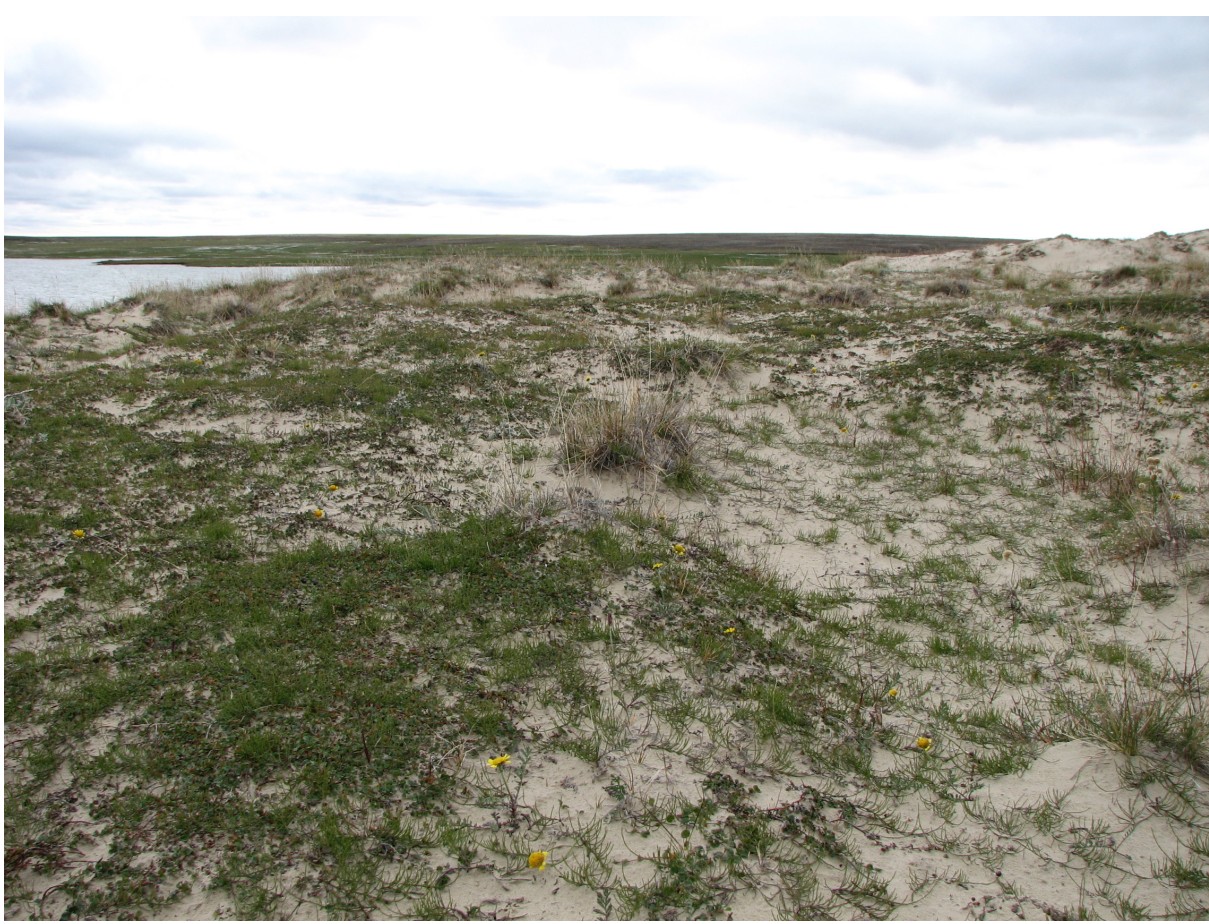

**Figure B62.** Example photograph class 21 (Elena Troeva, 2017, Northern Yamal)



## Appendix C: Comparison schemes for external landcover dataset



**Table C1.** Classes of CCI Landcover (Defourny et al., in preparation) and grouping for comparison (see Table 2).

| Group A | Description |
|---|---|
| other | 0 No Data |
| other | 10 Cropland, rainfed |
| other | 11 Herbaceous cover |
| other | 12 Tree or shrub cover |
| other | 20 Cropland, irrigated or post-flooding |
| other | 30 Mosaic cropland (>50%) / natural vegetation (tree, shrub, herbaceous cover) (<50%) |
| other | 40 Mosaic natural vegetation (tree, shrub, herbaceous cover) (>50%) / cropland (<50%) |
| forest | 50 Tree cover, broadleaved, evergreen, closed to open (>15%) |
| forest | 60 Tree cover, broadleaved, deciduous, closed to open (>15%) |
| forest | 61 Tree cover, broadleaved, deciduous, closed (>40%) |
| forest | 62 Tree cover, broadleaved, deciduous, open (15-40%) |
| forest | 70 Tree cover, needleleaved, evergreen, closed to open (>15%) |
| forest | 71 Tree cover, needleleaved, evergreen, closed (>40%) |
| forest | 72 Tree cover, needleleaved, evergreen, open (15-40%) |
| forest | 80 Tree cover, needleleaved, deciduous, closed to open (>15%) |
| forest | 81 Tree cover, needleleaved, deciduous, closed (>40%) |
| forest | 82 Tree cover, needleleaved, deciduous, open (15-40%) |
| forest | 90 Tree cover, mixed leaf type (broadleaved and needleleaved) |
| forest | 100 Mosaic tree and shrub (>50%) / herbaceous cover (<50%) |
| shrub tundra | 110 Mosaic herbaceous cover (>50%) / tree and shrub (<50%) |
| shrub tundra | 120 Shrubland |
| shrub tundra | 121 Evergreen shrubland |
| shrub tundra | 122 Deciduous shrubland |
| grassland | 130 Grassland |
| lichen/moss | 140 Lichens and mosses |
| barren | 150 Sparse vegetation (tree, shrub, herbaceous cover) (<15%) |
| barren | 151 Sparse tree (<15%) |
| barren | 152 Sparse shrub (<15%) |
| barren | 153 Sparse herbaceous cover (<15%) |
| wetland | 160 Tree cover, flooded, fresh or brakish water |
| wetland | 170 Tree cover, flooded, saline water |
| wetland | 180 Shrub or herbaceous cover, flooded, fresh/saline/brakish water |
| barren | 190 Urban areas |
| barren | 200 Bare areas |
| barren | 201 Consolidated bare areas |
| barren | 202 Unconsolidated bare areas |
| water | 210 Water bodies |
| snow/ice | 220 Permanent snow and ice |



**Table C2.** Classes of the CAVM (Raynolds et al., 2019) and grouping for comparison (see Table 2).

| Group A | CAVM Code | Unit |
|---|---|---|
| barren | B1 | Cryptogam, herb barren |
| grassland | B2a | Cryptogam, barren complex |
| shrubland | B2b | Cryptogam, barren, dwarf-shrub complex |
| other | B3 | Non-carbonate mountain complex |
| other | B4 | Carbonate mountain complex |
| grassland | G1 | Graminoid, forb, cryptogam tundra |
| grassland | G2 | Graminoid, prostrate dwarf-shrub, forb, moss tundra |
| shrub tundra | G3 | Non-tussock sedge, dwarf-shrub, moss tundra |
| shrub tundra | G4 | Tussock-sedge, dwarf-shrub, moss tundra |
| grassland | P1 | Prostrate dwarf-shrub, herb, lichen tundra |
| shrub tundra | P2 | Prostrate/hemi-prostrate dwarf-shrub, lichen tundra |
| shrub tundra | S1 | Erect dwarf-shrub, moss tundra |
| shrub tundra | S2 | Low-shrub, moss tundra |
| wetland | W1 | Sedge/grass, moss wetland complex |
| wetland | W2 | Sedge, moss, dwarf-shrub wetland complex |
| wetland | W3 | Sedge, moss, low-shrub wetland complex |
| water | FW | Fresh water |
| water | SW | Saline water |

**Table C3.** Classes of the CALC-2020 (Liu et al., 2023) and grouping for comparison (see Table 2).

| Group A | Class | Description |
|---|---|---|
| other | cropland | Arable land that is sowed or planted at least once within a 12-month period |
| forest | forest | Land covered by trees, with canopy coverage greater than 30% |
| grassland | graminoid tundra | Land covered by herbaceous vegetation with plant height typically ranging 5–15 cm |
| shrub tundra | shrub tundra | Land covered by shrubs of any stature with plant height typically ranging 20–50 cm |
| wetland | wetland | Land featured by aquatic plants and periodically saturated with or covered by water |
| water | open water | Inland open water bodies |
| lichen/moss | lichen/moss | Bedrock covered by cryptogam communities |
| barren | barren man-made | Impermeable land surface paved by man-made structures |
| barren | barren | less than 10% vegetation |
| ice/snow | ice/snow | Land covered with snow and ice all year round |