# Peer review of "Circumarctic landcover diversity considering wetness gradients"

_EGUsphere, 2023_

## Author Comment (AC5)

Supplement to response to **RC1**: 'Comment on egusphere-2023-2295', Martha K. Raynolds, 18 Dec 2023 (https://doi.org/10.5194/egusphere-2023-2295-RC1)

[Figure]

Figure 1: Comparison of CALU (10m, for legend of unit IDs see Bartsch et al. 2023) with CAWASAR wetness levels (500m, Widhalm et al. 2015) over the extent of the Alaskan North Slope as defined in Muller et al. (2018) (see also Figure 3).

[Figure]

Figure 2: Cross-comparison of CALU (10m) with the Lena Delta Landcover classification by Schneider et al. (2009), 30m (proportion of CAL Unit within class; for legend of unit IDs see Bartsch et al. 2023).

[Figure]

Figure 3: Cross-comparison of CALU (10m) with the Alaska landcover/ecosystems 2010 classification (30m) published in Muller et al. (2018) (proportion of CAL Unit within class; for legend of unit IDs see Bartsch et al. 2023).

[Figure]

Figure 4: Example for class 17 'recently burned or flooded' (shown in red). Burned areas in southwestern Alaska (outlines and years from Alaska Fire Database, https://www.frames.gov/catalog/10465). Dashed lines – boundaries of fires before 2019, bold line - fire extent 2019. (for legend see Bartsch et al. 2023)

[Figure]

Figure 5: Example for an AVA vegetation plot (Zemlianskii et al. 2023) located along a lake margin (square), falling into unit #2 (light blue, abundant macrophytes, shallow lake margin) instead of land area due to geolocation uncertainties. (for legend see Bartsch et al. 2023)

References

Bartsch, A., Efimova, A., Widhalm, B., Muri, X., von Baeckmann, C., Bergstedt, H., Ermokhina, K., Hugelius, G., Heim, B., and Leibmann, M.: Circumarctic landcover diversity considering wetness gradients, EGUsphere [preprint], https://doi.org/10.5194/egusphere-2023-2295, 2023.

Muller, S., D.A. Walker, and M.T. Jorgenson. 2018. Land Cover and Ecosystem Map Collection for Northern Alaska. ORNL DAAC, Oak Ridge, Tennessee, USA. https://doi.org/10.3334/ORNLDAAC/1359

Schneider, J., G. Grosse, and D. Wagner. 2009. "The Lena River Delta - Land Cover Classification of Tundra Environments Based on Landsat 7 ETM+ Data and Its Application for Upscaling of Methane Emissions." doi:10.1594/PANGAEA.759631. In Supplement to: "Land Cover Classification of Tundra Environments in the Arctic Lena Delta Based on Landsat 7 ETM+ Data and its Application for Upscaling of Methane Emissions." Remote Sensing of Environment 113 (2): 380–391. doi:10.1016/j.rse.2008.10.013.

Widhalm, B., Bartsch, A., and Heim, B.: A novel approach for the characterization of tundra wetland regions with C-band SAR satellite data, International Journal of Remote Sensing, 36, 5537–5556, 2015.

Zemlianskii, V., Ermokhina, K., Schaepman-Strub, G., Matveyeva, N., Troeva, E., Lavrinenko, I., Telyatnikov, M., Pospelov, I., Koroleva, N., Leonova, N., Khitun, O., Walker, D., Breen, A., Kadetov, N., Lavrinenko, O., Ivleva, T., Kholod, S., Petrzhik, N., Gunin, Y., Kurysheva, M., Lapina, A., Korolev, D., Kudr, E., and Plekhanova, E.: Russian Arctic Vegetation Archive—A new database of plant community composition and environmental conditions, Global Ecology and Biogeography, https://doi.org/10.1111/geb.13724, 2023.

---

## Author Response (AR1)

Dear editors,

We have revised the manuscript according to the general editor comments (removed reference 'in preparation' and added DOI for assets), the guest editor comment (rephrasing in case of similarity with input data description with a previous publication) and all reviewer comments.

**We would like to thank both reviewers for their very helpful and detailed revisions and comments.**

**Reviewer 1**

The authors present a circumarctic map of landcover derived from Sentinel data (both Sentinel 1 frozen-state radar data and Sentinel 2 multispectral) at 10-m nominal resolution. The legend has 23 units, including water, snow/ice and other/shadow. The map was created using an unsupervised classification, k-means approach, and descriptions of the units were summarized using ground data from over 3500 vegetation and soil samples in Western Siberia.

Mapping at this level of spatial resolution is a tremendous resource for Arctic research, and as the authors point out, could be especially useful for interpreting radar data. The examples presented in the paper (Figures 15, 16 and 19) show the level of heterogeneity that is captured at this scale. The mapping of water, other types that occur in small patches, and wetland heterogeneity is greatly improved with the smaller pixels. The measurements of heterogeneity in different parts of the Arctic, at the 1-km scale, are especially useful. Researchers often note the heterogeneity of arctic vegetation, and the difficulties that can lead to when trying to sample and extrapolate data.

The main problem with this study, is that it is difficult to understand the units in the map and how they would best be used in future analyses. Because the unit characteristics are summaries of ground data points, there is a lot of overlap in moisture, organic soil depth, and vegetation cover characteristics. The problem with this mapping approach is seen when one considers the difficulty of carrying out an accuracy assessment. It may not be possible to determine if a ground point is correctly mapped when distinctions (cut points) between the units are not well quantified or described.

- Reply: Note that the derived statistics represent to some extent geolocation uncertainties (in situ and satellite) and are partially limited in informational value due to the low number of samples. The soil and vegetation statistics are presented separately but should be interpreted together (as was done for the naming of the units). For example, volumetric water content statistics appear similar for many units, but their vegetation patterns differ.

- The overlap also exemplifies the heterogenous Arctic landscapes. It documents how far we can get with a 10 m nominal resolution. The 10 m cannot fully resolve landscape patterns, but provides a step forward. The statistics itself (standard deviation, box plots) could be used as an indicator for heterogeneity within the units.

Grouping the units into broader, more distinct categories can help, but eliminates much of the benefit of the fine spatial resolution. The title and abstract suggest that this mapping might be most useful for mapping moisture gradients (4 units: aquatic, wet, moist and dry). However, the legend categories are not clearly differentiated by soil moisture characteristics as seen in Figure 4, and one particularly problematic unit (#8) is characterized in Table 2 as aquatic to dry (but < 1% of map). Figure 6 shows that for the Western Siberia area, all map units had all four levels of

moisture, though some were dominated by one or two moisture levels. Unit #6, which is listed in Table 2 as dry is shown is Figure 6 as having mostly moist ground data.

- Reply: Note that the target moisture units are 'wet', 'medium' and 'dry' (see grouping B in Table 2). The mentioned 4 units are used in the vegetation in situ dataset (describing relevés, Figure 6). Figure 4 shows data from a soil dataset (point data). Both are considered for the descriptions. The reference to figure 6 is actually missing in line 288, and the text not directly referring to it what probably led partially to the confusion. We added the reference.

- We agree that the word 'represented' in the abstract, line 5, is misleading. The intention was to give a technical detail, the sampling of 10 m, and not the ability to capture the gradients fully at 10 m. Indeed, the spatial resolution of 10 m cannot fully resolve moisture gradients. Unit #8 represents landscapes with very high heterogeneity regarding wetness. However, as pointed out, this unit occurs only over 1% of the analyzed domain. A further challenge is the assignment of in situ points to certain pixels/units. Uncertainties exist for the determination of GPS coordinates in the field and also regarding geocoding of the satellite data since acquisitions with different viewing geometries (optical and SAR) are combined. The assignment of the four wetness units for the releves in the field is also partially subjective. These issues together result in assignments of 'wet' in situ points to dry units such as #6 and #7. But the proportion is comparably low. The sum of 'aquatic' and 'wet' in situ points is in the order of 20%. In case of the listing as wet in Table 2 it is >50%. In case of #6, in situ soil moisture data include also comparably dry sites (Figure 4, <20% volumetric content, comparably high number of samples). This led to the decision for assignment to group 'dry', what should have been explained in more detail. The comparison to the Alaska landcover map (now in supplement) shows that #6 primarily occurs in upland and alpine landscapes. Also the comparison with CAWASAR (Figure 1 below, item 3) confirms #6 as dry. However, we now also discuss the impact of the decision to label #6 as dry by adding a second wetness heterogeneity calculation (with #6 as moist).

- We rephrased the abstract:
  - Old: 'Patterns of lakes, wetlands, general soil moisture conditions and vegetation physiognomy are represented at 10 m nominal resolution.'
  - New: 'Patterns of lakes, wetlands, general soil moisture conditions and vegetation physiognomy are interpreted at 10 m nominal resolution.'
  - Old: 'The result goes beyond the capability of existing landcover maps which have deficiencies in spatial resolution, thematic content and accuracy.'
  - New: 'The result goes beyond the capability of existing landcover maps which have deficiencies in spatial resolution, thematic content and accuracy although landscape heterogeneity related to moisture gradients cannot be fully resolved at 10m.'

If the main purpose of the map is for wetlands/soil moisture, I would have expected it to be compared to the CAWASAR (CircumArctic Wetlands based on Advanced Aperture Radar) map (Obu et al., 2019). Instead, the map is compared to three land cover maps (CCI, CALC-2020, and Raster CAVM), using 6 grouped categories. One problem with this approach is that the other map legends have to be cross-walked to the 6 groups, and in the case of the Raster CAVM, that was not done correctly (see more detailed comments below).

- Reply: Please find our comments on the CAVM grouping in the detailed responses below. We now also provide a comparison to CAWASAR (originally published as Widhalm et al. 2015) as a supplement. There is specifically agreement regarding the CAWASAR

class 'dry' with the CALU group 'dry' (classes #6 and #21). The distinction between wet and moist is less reflected (scale issue?).

It could be that the meaning of the legend units would best be defined on a local or regional scale, rather than the circumpolar scale. The mapping seems to work quite well in the training and adjacent validation and example areas. However, the paper does not present maps of widely different parts of the Arctic, except in a small circumpolar map in Figure 11A. The extrapolation of the classification from the training area in Western Siberia to the circumpolar scale is one of the most questionable aspects of the method, so it needs to be shown to be effective, perhaps by comparing this CALU mapping with detailed local maps, where they are available for specific areas – perhaps from Long-Term Ecological Research (LTER) areas in the Arctic.

- Reply: We have now prepared comparisons with regional maps from Alaska and the Lena Delta in the supplement. However, the issue with such maps is that they are usually based on 30 m Landsat data. So most pixels are expected to be composed of different units except for comparably homogeneous regions such as the high Arctic with barren grounds and permanent snow/ice. We had therefore not included such comparisons in the main part of the manuscript. Note, that the used in situ soil probes represent sites across the Arctic, including Alaska, Canada and Greenland. The discussion has been extended (new subsection 'Description of units with in situ records') and refers to the examples in the supplement.

Comments on sections

Introduction

The introduction is very good, documenting the current state and limitations of Arctic land cover mapping, and the goals of this study. The references cited are appropriate and thorough.

Methods

Lines 121-123 – It is important to document the location of the in-situ (ground) data, so users of the map can evaluate the strengths/weaknesses of the map in their particular area. A map of ground data locations should be included in the Appendix.

- The map has been added

I suggest eliminating the word "transect" in favor of "region", as the training region and validation region are not linear transects (Figure 1 caption and map, and text). I got confused with these different terms. Line 221, 250 – change "transect" to "training area"

- Changed throughout the document and in Figure 1

Lines 221 -228 - Please describe the extrapolation of the prototype map to the entire circumpolar area in more detail. You say the "units remained largely the same". I would like to see this documented in some way in the Appendix. The use of this method, based on ground plots in only Western Siberia, to map the entire Arctic, is the weak point of this project. It is important to show that this did not prove to be a problem to convince people that this mapping is useful on a circumpolar scale.

- Reply: We agree that the phrasing "the units remained largely the same" is too vague. The largest change was change in the order. We now added a Sankey diagram in the

appendix to document this re-assignment. The method description on new unit assignment has been extended.

Lines 252-255 – The validation of the map relies on comparisons with other maps. You have done this by grouping the other map legends into 6 types. Reducing the number of legend units generally increases accuracy, so provides little information about the accuracy of the 21 CALU units.

- Reply:
- General remark: We have not defined target classes. We have now extended the description in the methodology also addressing comments by reviewer 2 to clarify this:
  - New: "The prototype is based on an unsupervised k-means classification. The resulting classes are units of similar reflectance of the shortwave incoming radiation (Sentinel-2) and radar backscatter at C-VV (Sentinel-1)."
- We use the in situ records to describe what the units represent. These assignments are an example of how to use the units. We have revised the manuscript in several places to clarify this. E.g. we changed in the column name in the unit/color table to "Example description"
- An actual accuracy assessment with the global to circumpolar maps is in general not possible due to the different thematic content and/or spatial resolution. We therefore only use them for cross-comparison. A cross comparison with multiple maps at the same time cannot be made without grouping. However, as suggested above, we have now made comparisons with 'ungrouped' regional maps (Lena Delta and Alaska). They are provided in the supplement.

There is also a problem in choosing how to group the legends of the other maps. In my opinion, as primary author of the Raster CAVM, the CAVM units are not grouped correctly in Appendix C-2. The Raster CAVM does not map most of the Arctic as shrub dominated. As a first response, I would say that types B3, B4, G1 and P1 should be in the barren group, B2a should be lichen/moss. G3 and G4 should be graminoid (but not grassland, as they are sedge-dominated). I do not know the CCI Landcover map, as it is unpublished, and cannot judge the groupings. The CALC-2020 map has few enough types that they seem to match the broad groupings well. I suggest that the authors work with the people who created the comparison maps to get the best possible crosswalk between the map units.

- Reply: We agree that the naming of the groups is confusing in the current version. The groups for comparison do not fully reflect botanical categories specifically regarding shrublands, the used 'shrub tundra' is misleading. We therefore propose to change the naming to 'dwarf to low shrub tundra' for clarification. Considering this change, regarding G3 and G4, which have dwarf shrub according to the CAVM legend, we would keep it in the 'dwarf to low shrub tundra' category. But we agree for B3 and B4 and have changed them now. A similar naming issue exists for 'barren'. Our 'barren' category excludes occurrence of graminoids. Thus B2a, G1 and P1, which have graminoids according to the CAVM legend would need to be kept separate. But instead of 'grassland' the category needs to be renamed 'Graminoids' (as is also suggested by you below) and the barren category needs to be clearly explained in the caption and text.

Results

The consistent use of colors in the figures and tables is helpful.

Table 2 - Change grassland to graminoid (and in appendix C and related figures). Grass-dominated areas are uncommon in the Arctic; sedge-dominated areas are common. At the 10-m resolution, grasslands would occur in revegetated disturbed areas, but this does not seem to be the definition of CALU #6, and would be rare, scattered pixels – probably not enough to make a unit on the circumpolar scale.

- Renamed throughout the manuscript

The low area for snow/ice indicates that you eliminated the Greenland Ice Cap and perhaps other year-round snow/ice. This should be described in the methods.

- We added' Only granules including ice free land areas have been considered.'

Section 4.4 - The text says "Burned areas from events before 2014 are represented through other classes (vegetation recovery).", however the map in Figure 10 (right) shows areas with fires in 1997 mapped as unit #17. Please clarify.

- Reply: Figure 10 shows a special case where fires occurred several times over the same area. The most recent one overlapped several old ones, but the extent was not clear with the chosen type of visualization (with year label in polygon centre). We now use a different line type for the most recent fire, what then clearly shows that the disturbed class occurs only within this boundary

"The disturbance unit #17 occurs within recently burned areas in approximately 72 % of all cases." How do you know this if you only have fire data from Alaska?

- Reply: We agree that the phrasing is misleading.
- New (original line 329): ' … of all cases documented for Alaska.'

Section 4.6 – The start of this section should include a summary of what is shown in Figures 15 and 16.

- New: 'Subsets for regions with dwarf to low shrubs with patches of wetlands within the prototype extent (see Figure 1) are compared in Figures 15 and 16. The prototype, CCI landcover, the CAVM and CALC-2000 are compared with CALU.'

Section 4.6.1 - Please add to the text a summary of how much the overall proportions of mapping units changed, and a comparison matrix (like Table 4) in the Appendix.

- We have now added a Sankey diagram

Section 4.6.3 – As described in my comments above under Methods, I do not think the comparison shown in Figure 17 and 18 are helpful due to the problem with the grouping. The maps shown in Figures 15 and 16, of areas in southern Western Siberia, are reasonable comparisons, but are all in Western Siberia.

- Reply: The choice for Western Siberia was determined by the extent of the prototype. But we agree that also other regions should be included. We have now added an example from Alaska and from Canada.

Section 4.6.4 – Lines 370-375 and Figure 18 – the comparison between CALC-2020 and CAVM is not very helpful here, as readers are trying to understand CALU, not these other two maps. I can understand the utility of this comparison for the authors, but I suggest moving Figure 18 to the Appendix, and focusing the text in this section on the comparison between CALC-2020 and CALU.

- the comparison has been now removed

Discussion

The Discussion Section needs work. It does not address the major issues in this study. It should discuss the effectiveness of the classification method used, in expanding from a training area to the circumpolar area. It should describe the effects, benefits, drawbacks of using a legend derived from statistically modeled groups (this study) vs. a legend of units defined by plant community composition or moisture units or soil organic content.

The existing text should be rewritten with separate paragraphs summarizing the CALU's effectiveness in mapping diversity, wetlands, and soil organic content. Comparisons to other maps should be minimal (moved to the Results 4.6)

- We now restructured and extended the discussion

Conclusion – no comments

General edit suggestions for the entire article

*Be consistent in your acronym – you use CAL unit, CALU, CALU ID. Choose one and use it consistently in the text, tables and figures

- we have now changed 'CAL units' to 'units (CALU)' where it was used as it is addressing specific units in all these cases
- CALU ID refers to specific unit numbers, so we kept it

*Be consistent in how you name your map units - unit #6 or unit 6. Or CAL unit 6. I think the # sign is helpful, but if you chose to use it, then don't use # elsewhere for numbering other things besides map units.

- Revised as requested

*Use a space between numbers and units, e.g. "30 m" (this is done inconsistently in the paper)

- Revised as requested

*Use the past tense for the methods and results sections. All of these actions have been completed, so should not be discussed in the present tense.

- Revised as requested

*'Has been' and 'have been' are not commonly used, as they suggest an action that started in the past, but continues in the present. Change them all to 'was' and 'were', respectively

- Revised as requested

*Delete the word "eventually", which suggests something that happens at the end of a very long process. Perhaps replaced by "then", which suggests a step in the process. But probably you can just delete the word.

- Revised as requested

Specific edits – These are mostly English wording changes, where I found the wording confusing. These suggested edits should be checked carefully to make sure to preserve the meaning desired by the authors.

Line 11 – change "considering 1kmx1km units " to "at a 1-km scale"

- Revised as requested

Line 13 – change "what is potentially " to "which is"

- Revised as requested

Line 33 – change "1kmx1km" to "1 km x 1 km"

- Revised as requested

Line 53 – change "(e.g. for the Arctic Reschke" to "(e.g. for the Arctic, Reschke"

- Revised as requested

Line 70 – delete ", however,"

- Revised as requested

Line 71 – delete "before"

- Revised as requested

Lines 85-91 – the mixed use of present and past tense in the last paragraph on the introduction makes it difficult to distinguish between studies done in the past and the current study. Please clarify and see general note above recommending using the past tense for completed studies (including this one).

- Revised as requested

Revised as requested:
Line 110 – change "depends on the used band" to "depends on the band"

Line 112 – change "have been used" to "were used"

Line 117 – change "analyses" to "analysis"

Line 123 – change "have been used" to "were used"

Line 124 – change "355 pedons were available for #1. The field soil sampling took place between 2006 and 2019" to "The field soil sampling took place between 2006 and 2019, and produces 355 pedons available for this analysis"

Line 125 – change "788 non forest samples were available for #2. The data was extracted from several different sources, including" to "Surface organic horizon depth were available for 788 non forest samples, extracted from several different sources, including". The first time I read this, I thought the #2 referred to subset 2 in Figure 1, which was confusing!

Lines 126-128 How does the % carbon distinguish between surface and buried organic layers? Ah, you are likely distinguishing between organic vs. non-organic layers. Maybe re-write this way "Soil horizons were defined as organic when their organic carbon content was ≥17% (equivalent to ca. 30% organic matter content) (Hugelius et al., 2020)."

Line 128 – delete this sentence. It is said more clearly in line 143

Line 129 – change "Arctic Vegetation Archive" to "Russian Arctic Vegetation Archive"

Line 134 – change "The AVA data used in this study was obtained during fieldworks" to "The AVA data used in this study were obtained during fieldwork"

Line 135 – clarify the "analysis extent". The training region, the evaluation region, the entire Arctic?

- Added (area within CAVM boundary)

Revised as requested:

Line 136 – change "assignment unit descriptions" to "assignment of unit descriptions"

Line 163 – change "analyses" to "analysis"; change "wild fire affected areas" to "wildfire-affected areas"

Line 172 – change "the Copernicus in DEM Product Handbook overall" to "Copernicus in the DEM Product Handbook, overall"

Line 174 – "In areas where these were not available the DEM is built on elevation contours" How were those elevation contours derived? Probably too much detail and you can just delete this sentence. Those interested in the details can look up Melvaer (2014).

Line 175 – change "has been derived" to "was derived"

Line 176 – change "The data was available" to "The data were available"

Line 180 – change "have been compared" to "were compared"

Lines 184-185 – change "In total six subzones are distinguished for the Arctic of which five can be found along the validation and calibration transect" to "In total five subzones are distinguished for the Arctic, of which four can be found along the validation and calibration transects"

Line 207 – change "have not been" to "were not"

Line 212 – change "of lack of cloud free" to "with limited cloud-free".

Line 212 - What does "Up to eight granules have been considered" mean? Considered for what? Do you mean "Up to eight Sentinel-2 granules (acquisition dates) were selected for any particular location"?

Line 230-231 – change "Not all shadow areas can be however identified due to similarities in reflectance over water bodies and wetlands." To "Not all shadow areas could be identified, due to similarities in reflectance with water bodies and wetlands."

Line 235 – change "areasusing" to "areas using"

Line 248 – change "as CALU" to "the same resolution as CALU"

Line 249 – "Classes have been grouped for comparison due to the large differences in thematic content" Explain this sentence more clearly. Which classes were grouped? For comparison with what? Which classes had large differences in thematic content? What types of differences?

From looking at the supplemental material and Table 2, I would say "A simplified set of nine units was developed that allowed cross-comparison between the CALU and the three external maps. Table 2 shows the grouping for the CALU, and Appendix C shows the groupings for the external maps.

Line 257 – change "is eventually" to "was"

Line 259 – change "rater" to "raster"

Line 268 – change "what could be achieved" to "which was achieved"

Line 274 – change "in the used Copernicus DEM" to "in the Copernicus DEM"

Line 276 – explain how you identified "anomalous meteorological/hydrological conditions"

- Reply: We used re-analyses data (as used for Sentinel-1 scene selection, line 216) to assess if a year was unusually dry or wet in summer.
- added: '(evaluated with re-analyses data)'

Line 277 - explain what is "too late or too early in the season"

- Reply: It refers to days at the beginning and end of the scene selection period (mid July to mid August) and varying seasonality/phenology between years.
- New: 'Low quality was usually caused by acquisitions close to mid-July and mid-August in years with deviations in phenology (early or late spring).'

Line 285 – change "occurrence patches." to "patchy occurrence at a scale of < 1 km2."

- Revised as suggested

Line 286 – change "40% are assigned" to "40% of the circumarctic area were assigned"

- Revised as suggested

Line 291 – delete "only"

- Revised as suggested

Line 298 – please describe, 48% of what? 48% overall average cover? Occurs in 48% of plots?

- Reply: ‚coverage' added

Line 299 – change "Lichens occur on average 12% and are reaching more than 20% only" to "Lichens had on average only 12% cover, and had over 20% cover only"

- Revised as suggested

Line 300  - change "average is 41%." To "average cover was 41%."

- Revised as suggested

Line 303 – change "with 11%." to "with 11% cover."

- Revised as suggested

Line 304 – change "which is are not" to "which were not"

- Revised as suggested

Lines 304-305 – you say that CALU #2 was not represented in the AVA, yet it is shown in Table 2 as having 51 samples.

- Reply: The used AVA dataset includes a number of plots which have been taken along lake margins, also at the boundary to unit #2. Due to geolocation uncertainties in the in situ data as well as satellite retrievals, some are located in unit #2 pixels.

- A typical example is now included in the appendix under the class description.

- We modified the sentence and added further details: 'Unit #2 represents shallow water along lakes and seashores which are not well represented in the AVA or soil records (see example in Appendix Figure B4). Vegetation and soil records (in case of organic layer thickness) are taken on land along the lake margins. Full pedon descriptions were not available.'

-

Line 306 – change "needle leave trees" to "needle-leaf trees"

- Revised as suggested

Line 307 – change "is common for the" to "commonly occur in the"

- Revised as suggested

Line 308 – change "Areas" to "These areas"

- Revised as suggested

Line 314-315 – tell us which CALU are you referring to as "permanent wetland class" (#3, I think) and "seasonal wetland class" (#4) and explain what measure has contrasting low and high standard deviation (it's not water volumetric water or mineral content based on Figures 4 and 5)

- Reply: Thanks for spotting. This is a remainder from an old version of the manuscript. The paragraph has been revised

Line 316 – change "(Table A2 with" to "(Table A2) with"

Line 316 – you say only 6 pedon samples were available for CALU #16, but doesn't Fig. 8 show that 21 were available?

- Reply: This refers to sites with full soil description for which only 6 are available (see figure B50). Figure 8 shows the sites with soil organic layer thickness what is available from more locations. There is actually a ')' missing after 'A2' on line 316 what may have led to the misunderstanding. It is now added.

Line 318 – change "SOC and TN" to "Soil organic carbon (SOC) and total nitrogen (TN)" change "the 'Barren' unit (Table A2)." to "the 'Barren' unit (CALU #21, Table A2)."

- Revised as suggested

Lines 325-326 – change "The assignment to the disturbance unit #17 does occur up to four years" to "Pixels were assigned to the disturbance unit #17 for up to four years". The text says this, however the map in Figure 10 (right) shows areas with fires in 1997 mapped as unit #17. Please clarify.

- Reply: See our clarification above (number 12). The fire 2019 overlapped several preceding fires.

Line 343 – change "do not only relate" to "are not only related"

- Revised as suggested

Line 347 – change "can be also confirmed" to "can also be confirmed"

- Revised as suggested

Line 349 – "The new disturbance classes". I thought there was only one disturbance unit, #17

- See the explanations above for prototype versus new version

Lines 356-357 – "Figures 15 and 16). 63% are shrub tundra and about 9% grassland and 16% lichen/moss (Table 11)." Maybe change to "Figures 15 and 16), with 63% shrub tundra, about 9% grassland and 16% lichen/moss CALU grouped units." There is no Table 11. Would you like to include more tables in the Appendix?

- Reply: Thanks for spotting. This is a remainder of an earlier version of the manuscript. It should refer to Table 4 and the shrub fraction is 60% instead of 63% (and sum 85%).

Lines 380-381 – change "Especially wetlands (in addition to lakes from lakes (Matthews" to "Wetlands (in addition to lakes (Matthews", change "dry and wet" to "moisture"

- Revised as suggested

Line 387 – change "CALU provides potentially allows for" to "CALU potentially allows for"

- Revised as suggested

Line 387-389 – This last sentence makes a separate, important point that should be expanded into a full paragraph, perhaps moved to lines 404-410.

- Reply: This is indeed an important point. We added a dedicated subsection "Application potential".

Line 390 – change "based Landsat derived landcover maps" to "of Landsat-derived global landcover maps"

- This would change the meaning. Not Landsat derived global maps have been assessed, Landcover CCI was assessed with regional Landsat maps (from North Slope and Lena Delta). We suggest to the following phrasing:
  - Based -> which used; and adding "regional"

Lines 392-393 – "over the top meter" – does this mean on the surface of the ground or soils to 1 m depth? Please clarify.

- Reply: It refers to soils to 1m depth

Line 394 – change "The agreement for the" to "The total proportion of"

- Reply: this would change the meaning. We refer here to classification accuracy. Landcover CCI only captures about 20% of wetlands.
- Rephrasing: The agreement between the wetland classes of Landcover CCI and CALU was very low what confirms the assessment based on in situ observations by Palmtag et al. (2022).

Line 409 – change "used tools" to "tools used"

- Revised as suggested

Line 411 – change "Although that" to "Although", "data was" to "data were"

- Revised as suggested

Line 419 – change "identified ice/snow" to "identified as ice/snow"

- Revised as suggested

Line 423 – change "analyses does not" to "analyses do not"

- Revised as suggested

Line 439 – What does "(1%)" mean. Please expand

- Now removed

Figure 2 caption – add "(number of granules)"

- Added

Figure 3 caption – change to "Number of scenes used, by Sentinel granule, within the CAVM boundary."

- Changed

Figures 4, 5, 8 captions – add description of horizontal axis, including definition of CAL Unit, the meaning of the numbers at the top of the figure, and the meaning of the boxes, lines, plus and circle symbols

- adjusted

Figure 6 caption – change to "Wetness categories for Circumarctic Landcover Units (CALU) based on Arctic Vegetation Archive plots, Western Siberia (only units with at least 10 data points)."

- Revised as suggested

Figure 7 – this figure is not necessary. The data would be better presented by adding as "SD" column to Table 3. Possibly the same for Figure 12.

- Reply: We moved it to a separate table providing Std, mean and median. The mean value was deleted from Table 3.

Figure 10 – put both scale bars in the same unit, km

- Revised (different example used to allow unit match)

Figure 11 caption – add name of map and original resolution.

- Revised as suggested

Figure 13 caption – change "Circum Arctic vegetation map" to "Circumpolar Arctic Vegetation Map". This figure is helpful in comparing the CALU with the CAVM.

Figure 14 – missing legend. I cannot tell what the different shades of gray mean. Once the legend

is in there, the caption will need to explain it. I originally took the comma to mean a decimal-place marker. A separate legend would be clearer.

- Reply: The legend is included and explained in the caption ('sum of groups ...'). The confusion comes probably from the comma which separates the legend item from the actual value. However, with the addition of the calculation of #6 as 'moist', we have changed the chart from circular to bar chart. A further change is that 'no data' (which was represented as 0) is now removed from the calculation of proportion as it is the same for both. This results in higher percentage than before for #6 as 'dry'.

Figures 15 &16 captions – Many readers will turn to these figures to see a summary of your results. You should include the full name of each map in the comparison and its spatial resolution and refer to Table 1.

- changed

Table 1 caption – change "CALC" to "CALC-2020" as that is how it is referred to in the rest of the paper; include "LCP (Land Cover Prototype??)"; change "Circumarctic Vegetation Map" to "Circumpolar Arctic Vegetation Map"

Changed

Table 2 caption – change to "Classes of Circumarctic Landcover Units (CALU) and Gr.A) grouping for comparison, Gr.B)"  Change "Appendix A" to "the Appendices".
- The caption was rephrased also considering further details

Table 2 – Add "seasonal" to the description of CALU #4, as you describe it that way in the text.
- Added

Table 3 caption – Please change the caption to clearly state what you are showing in the table. I think you mean "Average cover of different lifeforms for each Circumarctic Landcover (CAL) Unit based on Arctic Vegetation Archive plots, Western Siberia.".
- Revised as suggested

Table 4 – define CAL and CCI in the table heading. Similar tables for the other two maps (CAVM and CALC-2020) could be presented in the Appendix.

**Reviewer 2**

This paper presents very interesting circumarctic land cover mapping product. The patchiness and heterogeneity of tundra landscapes is well known, but not at all well mapped issue. This product has several advantages when compared to products available earlier. Only one recent (CALC-2020) has similar spatial resolution (10 m), but this one seems to be thematically better suited for different environmental etc. analyses and modelling. In any case, it could be still argued, is now used thematic division relevant enough, some classes does not look very sensible, and how well they in the end are mapped.

- Reply: With our study we present how far it is possible to go on Arctic scale with current remote sensing capabilities. The thematic division represents the separability of Arctic landscapes using satellite data as it is based on unsupervised clustering (k-means). Target classes have not been predefined. The division also represents the potential of 10m nominal spatial resolution which is clearly not sufficient to fully address landscape heterogeneity in many cases. A further challenge is the interpretation of the

clusters/units which relies on extensive consistent in situ observations. We agree that both needs to be better explained and discussed. The text has been extended.

It is also still clear, that this kind of ambitious circumarctic land cover mapping is still very much "work in progress". In any case, I see this is important work to publish, and I congratulate Authors to get this huge task done to this step.

Unfortunately, due to some flu etc. issues, I had no time to give very comprehensive review in given time, but some more detailed comments here in any case. However, other reviewer already did a good job giving a lot of relevant comments.

Used classification algorithm is not described well at all (rows 117-9). It seems that it is described in more detail in refereed paper, but I would added here some more information. For example, what was criteria to select just those 25 classes?

- Reply: Since the approach is based on k-means, no target classes have been predefined. The resulting classes are units of similar reflectance of the shortwave incoming radiation and radar backscatter at C-VV. We have now added a detailed description for the new class assignment.

Then, was there any way to handle/smooth the mixed-pixel effect, like some fuzzy method using nearest cells, etc – and if not, I would discuss would it have been sensible to try to use some?

- Reply: The prototype was developed at 20m nominal spatial resolution as also the 20m-bands of Sentinel-2 have been used to better exploit the spectral capabilities. In this study a super-resolution approach adjusted to Arctic environments was now applied to bring these bands to 10m. The approach is based on convolutional neural networks (see line 205). The documentation of the units with in situ data clearly shows the limitation of the nominal 10m. The inclusion of the 20m bands widens the spectral resolution but impacts actual spatial resolution
- We extended the discussion (line 379) with a detailed presentation which specific units have changed between the prototype and the new version

Used field vegetation data coming from Russian Arctic Vegetation data Archive, covers only European (mostly) Russian and west Siberian region. I could argue that in other locations in the Arctic, there are vegetation communities not matching well communities found in this region. This data set was harmonized, and it was also from same area, where earlier version of land cover classification using similar method was done, so its use was in that sense understandable. However, when the goal is make really relevant circumarctic product, in the next phase more different data sets other part from the arctic should be also included. This issue is shortly mentioned in rows 410-11, but I would, like to see some more discussion, how this regional "bias" might have affected the product. This then also might lead to some extent different thematic content of the product.

- Reply: We extended the discussion regarding this issue (also under new section application potential). However, note that the soil data represent sites across the entire Arctic. We added a map of their distribution for clarification.

Some thematic classes looks to some extent odd, like classes 6 and 8, where both wet and dry habitats are included. This is problem, when thinking, for example, methane dynamics. The logic behind selected thematic classes could be explained and justified still better. And some adjustments would be good to do, when next version of this product (hopefully) is produced.

However, I well understand huge challenges to try to formulate such general classes, that could cover all circumarctic vegetation communities and other land cover types, and could still be mapped from remote sensed data.

- Reply: As noted above, the 10m are insufficient to fully resolve Arctic landcover heterogeneity. This is specifically the case for units #6 and #8, which are each areas spectrally in one class but represent a range of moisture conditions. What needs to be also noted in this context is the assignment of the names based on the in situ data statistics. There are uncertainties in geolocation of both the satellite product and the in situ data. The representation of the in situ description for the 10x10m is also an issue. This mismatch may also lead to the fuzzy/broad naming of the units in heterogeneous areas.

I was to some extent surprised to notice that organic layer thickness was so low in those classes that looks like paludified, like classes 3, 4, 5 (and 6?), and how it was so high in class 15 (see Figure 8)? This could be at least discussed.

- The representative nature of organic layer depth records and soil probes in general indeed needs to be discussed. Data are usually collected at specific type of environments, often river flood plains (e.g. Lena delta, Kytalyk) and drained lake basins (e.g. Barrow). The time for paludification has been limited there. Units 3, 4, 5 and 15 represent gradients at such locations. Unit 5 and 15 sites also show a high mineral content. The number of records from unit 15 is comparably low and the STD very high (spread from 0 to more than 4 m). Probes from other settings are underrepresented.
- We have now extended the discussion:
  - Probes for organic layer thickness come also from selected environments, mostly river floodplains (e.g. Lena Delta, Kytalyk) or drained lake basins (e.g. Alaskan North slope). Data for wet unit types (#3-5) come largely from these type of sites. Paludification is therefore comparably low. OLD is thus rather low for these units. Unit #15 had rather high OLD (Figure 7), but only 25 samples with a high spread and high mineral volumetric content. Most sites are also located within floodplains. In addition to use of more in situ data and region specific retrieval, a stratified random selection of soil probes representing a diversity of settings might be eventually required to allow upscaling of soil properties.

Add colour legend to Figure 10. For figures 15, 16 (and uppermost in 17?) colour legend information is said to be found in Table 2. It might work like that, but does not look very clear to me.

reply: Thanks for spotting. The legend information can be also found in Table 2 in this case. A reference has been now added.

---

## Author Response (AR2)

Dear editor,

We have implemented all suggested grammar and spelling changes and a reference as requested. In addition, regional names (West Siberia, Polar Urals) have been made consistent.

Figure 9 has been adjusted for color-blindness. In other cases the readability is given, mostly due to the provision of unit IDs in addition.
The color has been removed from Table 2 to comply with the journal rules. A copy of the color version has been added as figure to the appendix instead. The color tables of appendix C have been converted to figures. References to these Tables/Figures have been updated in the text accordingly.
Color and caption updates have been made to figure 19 and a further supporting figure added to the supplement (see detailed reviewer response below).

Response to reviewer comments:

There are two things you said you included in the supplements (Appendices) that I cannot find:
• In your response document, you say "We now also provide a comparison to CAWASAR (originally published as Widhalm et al. 2015) as a supplement." I do not see this comparison in the Appendix.
• In your response document, you say "We have now prepared comparisons with regional maps from Alaska and the Lena Delta in the supplement". I do not see this in the Appendix.

>Response: There is an Appendix and a separate Supplement (was new after first revision). The information not found by the reviewer is part of the supplement which was uploaded as a separate file.

Minor editorial comments (except grammar and spelling, addition of reference)

Line 185 – similar to what?
>Response: added ' as anticipated in this study'

Line 334-335 – I found this sentence confusing and suggest this possible rewrite. "AVA records, which include wetness descriptions, include the categories wet and aquatic in more than 50 % of the samples in units #3, 4 and 8 (Figure 6).
- Adapted to: AVA records, which provide wetness descriptions, included the categories wet and aquatic in more than 50 % of the samples in units …

Lines 354-357 – At first (and second reading), I didn't understand these sentences, what the point was that you were trying to make, or what the groups were in Figure 13. Since I found this so confusing, a sentence explaining this in the Fig. 13 caption or in the text would be helpful. While this may seem like overexplaining, something like the following would be helpful: "Wetness group 1 sum is the proportion of 1 x 1 km areas that had only one wetness category, group 2 had two wetness categories, and group 3 had all three."
- Added to caption 'Values for wetness group sum '1' are the proportion of 1 x 1 km areas that had only one wetness category, group sum '2' had two wetness categories, and group sum '3' had all three.'

Line 423 – change "fires are abundant and smoke is transported to the tundra area." to "fires

are abundant in the boreal forest and smoke is transported to adjacent tundra areas."

Lines 423-425 – Is there a northern water-cover layer that you would recommend users apply to fix this problem? Or would you recommend users apply regional water-cover screens to improve the map?

- Added: The snow/ice class can be, however, re-labeled as water in regions without glaciers (excluding high Arctic islands, Greenland and mountain ranges)

Figure 19 caption – on my computer screen, there is no gray or light green in the CAL-2020 maps. I see a brown and a tan color, so do not know which is the lichen/moss and which is the graminoid.

- Response: This is indeed a wrong version of the color scheme. Figures are now replaced.
- The caption has been extended in addition. The information on wetscape type according to Olefeld et al. 2021 has been included and the Wetscape comparison added to the supplement. One sentence was added in the main text.

Line 508 – I would add that the CALU covers 97% of the Arctic (or say only 3% missing).

- Added: CALU covers only 97% of the Arctic (excluding ice sheets).

Table A3 – In caption, change "characteristics" to "percent cover" or something similar. The column "Samples" has incorrect data – a copy of the "Mean". Change "Tussok graminioids" to "Tussock graminoids"

- Thanks for spotting. The sample column is now removed as it is not applicable in this case (template for soil table was initially used).

Figure D1 – Please include CALU unit #s

- updated